# HIV-1-induced nuclear invaginations mediated by VAP-A, ORP3, and Rab7 complex explain infection of activated T cells

Mark F. Santos [1], Germana Rappa[1], Jana Karbanová [2], Patrizia Diana[3], Girolamo Cirrincione [3], Daniela Carbone [3], David Manna[4], Feryal Aalam[1], David Wang[1], Cheryl Vanier [1,5], Denis Corbeil [2] ✉ & Aurelio Lorico[1] ✉

The mechanism of human immunodeficiency virus 1 (HIV-1) nuclear entry, required for productive infection, is not fully understood. Here, we report that in HeLa cells and activated CD4[+] T cells infected with HIV-1 pseudotyped with VSV-G and native Env protein, respectively, Rab7[+] late endosomes containing endocytosed HIV-1 promote the formation of nuclear envelope invaginations (NEIs) by a molecular mechanism involving the VOR complex, composed of the outer nuclear membrane protein VAP-A, hyperphosphorylated ORP3 and Rab7. Silencing VAP-A or ORP3 and drug-mediated impairment of Rab7 binding to ORP3-VAP-A inhibited the nuclear transfer of the HIV-1 components and productive infection. In HIV-1-resistant quiescent CD4[+] T cells, ORP3 was not hyperphosphorylated and neither VOR complex nor NEIs were formed. This new cellular pathway and its molecular players are potential therapeutic targets, perhaps shared by other viruses that require nuclear entry to complete their life cycle.

Although current therapies have made HIV-1-induced acquired immune deficiency syndrome (AIDS) a manageable chronic disease, it remains a global health priority. The unavailability of a cure and the development of drug resistance indicate that a deep understanding of all steps of the HIV-1 life cycle is required to develop better therapeutic strategies. The early phase of the retroviral life cycle requires cell entry and access to the nuclear compartment for integration of viral components into host DNA. While early studies supported pH-independent cellular entry of HIV-1 by direct fusion with the plasma membrane[1–3], accumulating evidence implicates the fusion of viruses with endosomal membranes upon cellular uptake[4–8]. For example, pseudotyping HIV-1 by the vesicular stomatitis virus G (VSV-G) protein targets HIV-1 entry to the endocytic pathway[9,10]. By applying *trans* dominant-negative mutants of dynamin and Eps15, it was found that dynamin-dependent, clathrin-mediated endocytosis can lead to productive HIV-1 entry into CD4[+] HeLa cells[11]. Although the cellular entry of HIV-1 by endocytosis has been viewed and/or considered as a dead-end pathway leading to degradation of viral components in the lysosomal compartment[12,13] (reviewed in ref. 14), it might nonetheless dictate the pathway through which the large pre-integration nucleoprotein complex (PIC) reaches and enters the nucleus. Indeed, the subsequent spatiotemporal intracellular steps, which can be cell type-dependent, are unclear or under debate for HIV-1 and other enveloped viruses that replicate in the nucleus[15–18]. Viral capsid uncoating and nuclear import of the viral genome are critical for productive infection[19]. Host factors are potentially involved in nuclear entry of the PIC, including importins[20,21] and components of the nuclear pore complexes (NPCs)[22–27]. Recently, transport of intact cone-shaped HIV-1 capsids through NPCs[28,29] and capsid uncoating at the NPCs[30,31] or inside the nucleus[32,33] were reported. Since inhibition of HIV-1 nuclear import would block productive infection, this step of the HIV viral life cycle is a potential drug target.

[1]Touro University Nevada College of Osteopathic Medicine, Henderson, NV, USA. [2]Biotechnology Center (BIOTEC) and Center for Molecular and Cellular Bioengineering, Technische Universität Dresden, Dresden, Germany. [3]Department of Biological, Chemical, and Pharmaceutical Sciences and Technologies (STEBICEF), University of Palermo, Palermo, Italy. [4]Touro College of Osteopathic Medicine, Middletown, New York, NY, USA. [5]Present address: Imgen Research, LLC, 5495 South Rainbow #201, Las Vegas, NV, USA. ✉e-mail: denis.corbeil@tu-dresden.de; alorico@touro.edu

We have recently described a novel intracellular pathway by which proteins and nucleic acids from endocytosed extracellular membrane vesicles (EVs) reach the nuclear compartment and alter the gene expression profile or cell fate[34,35]. Entry of endocytosed EVs into the nucleoplasmic reticulum (NR), specifically in type II nuclear envelope invaginations (NEIs)[36], and subsequent nuclear transfer of EV cargo via NPCs require docking of late endosomes to the outer nuclear membrane (ONM)[34]. The docking is mediated by the VOR protein complex, comprised of ONM-localized vesicle-associated membrane protein (VAMP)-associated protein A (VAP-A), cytoplasmic oxysterol-binding protein (OSBP)-related protein-3 (ORP3), and late endosome-associated small GTPase Rab7[37]. This pathway can be intercepted by the FDA-approved antifungal drug itraconazole (ICZ)[38] that acts on ORP3, a member of the OSBP family. ORP3 is involved in exchange of lipids, notably sterols found at membrane contact zones between organelles[39–41]. The homology between exosomal pathways and the viral cycle, as postulated by the Trojan exosome hypothesis[42–44], led us to investigate whether the EV nuclear entry pathway applies to viruses that require nuclear access for their life cycle. Here we show that HIV-1 pseudotyped with VSV-G or native Env uses the same pathway as EVs after internalization into HeLa cells or primary CD4+ T cells to deliver its content into the nucleus of host cells, a process that can be intercepted by chemical drugs.

## Results

### VSV-G-pseudotyped HIV-1 is packaged into late endosomes and induces nuclear envelope invaginations prior to entering the HeLa cell nucleus

To investigate whether viruses use a new intracellular pathway recently described to be involved in the nuclear transfer of EV cargo[34], we employed VSV-G-pseudotyped HIV-1 Gag-iGFP (hereafter HIV-Gag-iGFP) and HeLa cells as our first model system. Like EVs, it is well established that VSV-G pseudotyped HIV-1 uses endocytosis as a mechanism of cellular entry[9,10]. Throughout the study, we used a multiplicity of infection (MOI) of 2, unless otherwise specified, and RetroNectin, a recombinant fibronectin fragment, as the primary agent to infect the cells of interest[45–47]. As a control, we used "bald" virus, lacking VSV-G (see below). The intracellular trafficking of viral particles was monitored by confocal laser scanning microscopy (CLSM) after immunolabeling of HIV-1 integrase (IN) with the IN-2 antibody or by Gag-iGFP trapped in viral particles. The very low fluorescence of iGFP could be observed, as demonstrated using viral particles adhered to coverslips, by applying the gamma function of Fiji software. The analysis revealed a weak GFP signal (mean fluorescence per particle is $3.78 \pm 1.67$ ($n = 52$) and $8.49 \pm 2.17$ ($n = 43$) relative fluorescence units (RFU) for VSV-G-pseudotyped and "bald" viruses, respectively), especially compared with CD9-GFP+ EVs ($45.59 \pm 4.11$ RFU, $n = 30$) produced by engineered melanoma cells (Supplementary Fig. 1a, c, respectively). Thus, Gag-iGFP was detected upon signal enhancement (Supplementary Fig. 1a).

After 1 h of infection, IN-2 immunoreactivity (IN-2) was detected in the cytoplasm and NR, specifically in type II NEIs, as evidenced by immunolabeling of VAP-A, a marker of ONM and endoplasmic reticulum (ER) (Fig. 1a). It should be noted that the entire nuclear compartment was scanned to observe the NEIs, as they are limited to certain x-y planes. Thus, IN-2 within the NEIs can appear at any z-level of the nucleus, a phenomenon that varies from cell to cell. After 1 h infection, $34.5 \pm 7.0\%$ of cells were positive for IN-2 (40 cells per experiment, $n = 3$). No IN-2 was observed without infection (Fig. 1a, Control) or when primary antibody was omitted (Supplementary Fig. 2a). Although the Gag-iGFP associated with viral particles was barely detectable (Supplementary Fig. 2b), after image post-processing (Supplementary Fig. 2b), the enhanced Gag-iGFP signal colocalized with IN-2 in cytoplasm and NEIs (Pearson's R correlation coefficients: $0.85 \pm 0.07$ and $0.72 \pm 0.10$, respectively, $n = 10$), suggesting that the intact virion moves to the cytoplasmic core of NEIs within 1 h of infection.

After reaching the cytoplasm and NEIs, IN-2 began to be detected in the nucleoplasm (Fig. 1b). Between 1 and 5 h after infection, IN-2 increased in the nuclear compartment while decreasing in the cytoplasm (Fig. 1c, d). After 3 h of infection, $\approx 40.0 \pm 9.0\%$ of cells contained IN-2 (40 cells per experiment, $n = 3$). Interestingly, the number of type II NEIs per 100 cells increased significantly, while type I NEIs (containing only the inner nuclear membrane (INM)), detected by anti-SUN domain-containing protein 2 (SUN2), remained unchanged (Fig. 1e, f). The number of type II NEIs also increased at the single-cell level (Supplementary Fig. 3a), further suggesting that viruses modulate nuclear architecture (see below). Induction of NEIs was not observed when "bald" viruses were applied (Supplementary Fig. 3b, see below). Indirectly, these data suggest that EVs ($\approx 2.2 \times 10^8$ particles/ml) present in both the "bald" and VSV-G-pseudotyped virus preparations did not contribute to the formation of NEIs. Nonetheless, a concentrated preparation of melanoma cell-derived EVs ($1 \times 10^9$ particles/ml; their characterization is presented in ref. 38) also induced type II NEIs (Supplementary Fig. 3b), in agreement with our previous study[34].

To better understand the relevance of the mechanism regulating HIV-1 IN entry into the NR and nucleoplasm, we investigated whether HIV-1 enters HeLa cells by endocytosis by pretreating them for 30 min with dynasore (DNS), a cell-permeable dynamin inhibitor that prevents the scission of clathrin-coated pits from plasma membrane[48], before 3-h infection. In dimethyl sulfoxide (DMSO) control, IN-2 appeared in cytoplasm, NEIs, and nucleoplasm, while no cellular IN-2 was detected when endocytosis was blocked (Fig. 1g, h), consistent with previous reports[6]. Instead, IN-2 accumulated outside the cells and near the cell surface (Fig. 1g, right panel, asterisks). To get more insight into the intracellular trafficking of HIV-1 IN, we induced the expression of either Rab5- or Rab7-red fluorescent protein (RFP), as markers of early and late endosomes, respectively, using a baculovirus-based system. Cells were then infected for 1 h, and the subcellular localization of IN-2 was determined every 15 min by CLSM (Fig. 1i). IN-2 colocalized with Rab5-RFP after 15–30 min either at the cell periphery or the perinuclear region (Fig. 1j, l and Supplementary Movie 1). At 45–60 min, IN-2 increasingly colocalized with Rab7-RFP in the perinuclear region and often asymmetrically around the nucleus (Fig. 1k, l and Supplementary Movie 2). Colocalization of IN-2 with Rab5-RFP or Rab7-RFP shifted over 1-h monitoring, as shown by Pearson's R correlation coefficients (Fig. 1m). No IN-2 was observed before HIV-Gag-iGFP infection (Fig. 1j, k, control). The HIV-1 IN and Rab7 colocalization was also observed when Rab7 was immunolabeled, with both proteins found in the NR, as seen in transverse and longitudinal sections of NEIs (Fig. 2a and insets a1 and a2). These data exclude the effect of Rab7-RFP overexpression. After 1 h of infection, 48% of NEIs contained Rab7 and IN-2, while 27% contained only Rab7 and 25% had neither ($n = 50$ cells). The presence of p24 antigen, a component of the HIV-1 particle capsid, in Rab7+ late endosomes was also observed (Pearson's R correlation coefficients: $0.81 \pm 0.11$, $n = 10$ cells) (Supplementary Fig. 4a, b and Supplementary Movie 3). Beside Rab7, CD63, a marker of intralumenal vesicles (i.e., precursors of exosomes) associated with late endosomes/multivesicular bodies, was found to colocalize with IN-2 in NEIs (Fig. 2b and inset b1). In contrast, Lamp1, a lysosome marker, did not (Fig. 2c and insets c1 and c2). Only one out of 50 infected cells analyzed showed Lamp1 in NR. Double labeling for Rab7 and Lamp1 in noninfected cells confirmed the exclusion of Lamp1, but not Rab7, in NEIs (Fig. 2d, and inset d1). Although excluded from the NEI, Lamp1 nevertheless partially colocalized with Rab7 in the cytoplasmic compartment (Fig. 2d and inset d2), as described by others[49,50]. Similar observations were made in HIV-Gag-iGFP- infected cells (Fig. 2e, insets e1 and e2, and Supplementary Movie 4). In infected

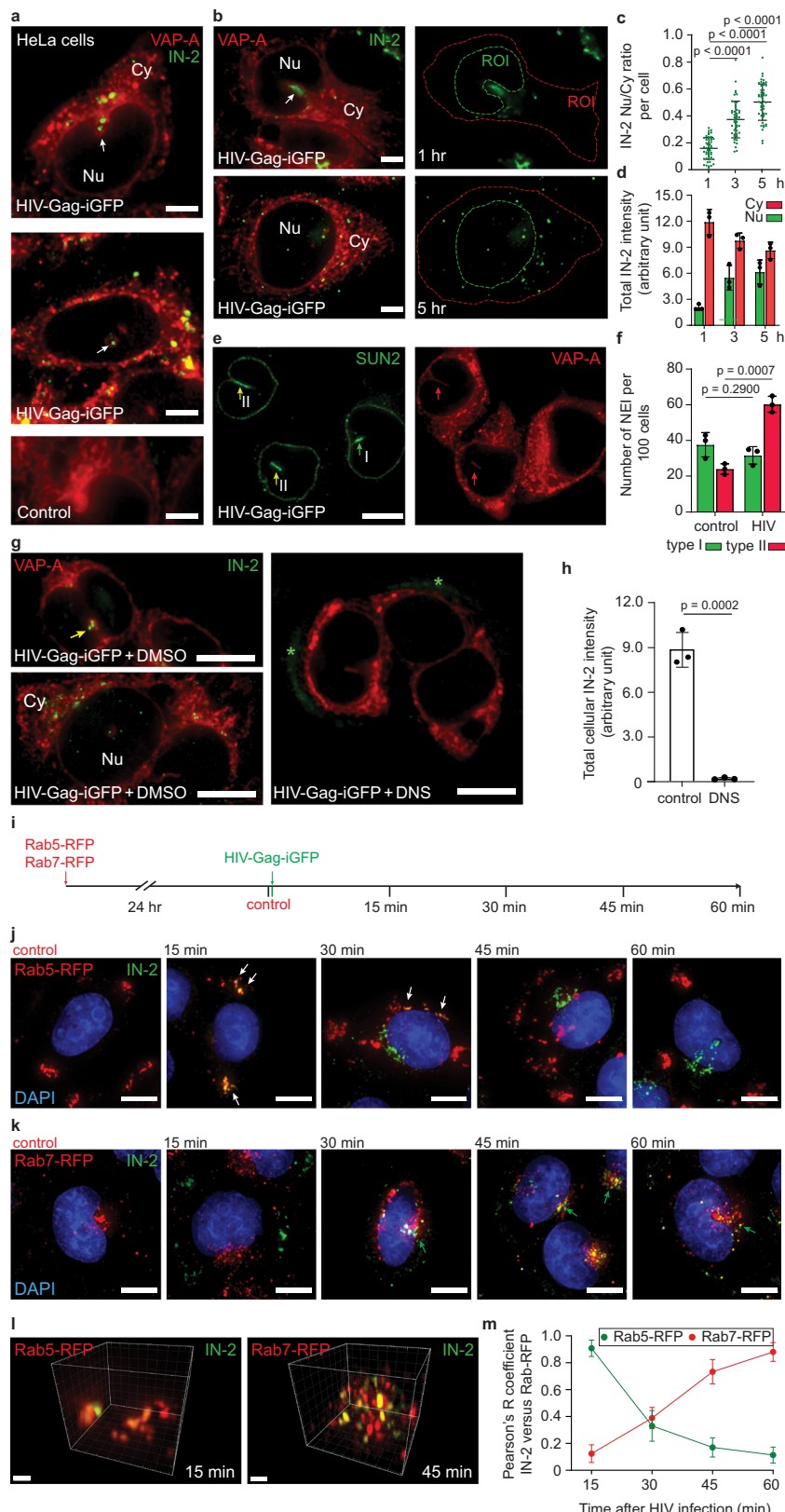

cells about 70% of NEIs contained only Rab7 and almost none contained both Rab7 and Lamp1 (Fig. 2f). Indirectly, these data suggest that the intracellular pathway used by a fraction of virus-laden Rab7⁺ late endosomes to reach the NEI is distinct, at least in part, from that of late endosomes that fuse with lysosomes and lead to the degradation of viral components.

Almost no IN-2 was observed in the cytoplasmic compartment of cells infected with "bald" virus for 1 h (Fig. 2g, inset g1). In the latter case, an accumulation of IN-2 was observed outside the cells, similar to the DNS treatment. Only a minute fraction of IN-2 from "bald" virus was able to enter some cells in a non-specific manner (see below), which is consistent with other studies[51,52]. Overall, these experiments confirm

**Fig. 1 | VSV-G-pseudotyped HIV-1 is endocytosed and induces type II NEIs.**
**a** HeLa cells were 1-h HIV-Gag-iGFP infected prior to VAP-A and HIV-1 IN (IN-2) immunolabeling. As controls, noninfected cells or no HIV-1 IN antibody were used (Supplementary Fig. 2a). Samples were observed by CLSM and single sections are displayed. Longitudinal (top and bottom panels) or transverse (middle) sections of NEI (arrow) are shown. **b–d** Cells were 1-, 3-, or 5-h infected prior to VAP-A and IN-2 immunolabeling. Nucleoplasmic (Nu) and cytoplasmic (Cy) IN-2 were quantified as defined by regions of interest (ROI, **b**) and their ratio per cell from one representative experiment (**c**) or total IN-2 intensity per cell are indicated (**d** 50 cells per experiment, $n = 3$). **e**, **f** Cells were 3-h infected (or not) prior to VAP-A and SUN2 immunolabeling. Type I (VAP-A$^-$SUN2$^+$) and II (VAP-A$^+$SUN2$^+$) NEIs (**e** green and yellow/red arrows, respectively) were quantified (**f** 100 cells per experiment, $n = 3$). **g**, **h** Cells were 30-min pretreated with DMSO or DNS (80 μM) prior to 3-h HIV-Gag-iGFP infection followed by VAP-A and HIV-1 IN immunolabeling (**g**). Note that IN-2 accumulated in NEI (yellow arrow) and/or dispersed into cytoplasm and nucleoplasm in the control, whereas it remained outside the DNS-treated cells (asterisk). Total IN-2 intensity is presented (h, >50 cells per experiment, $n = 3$).
**i–m** Colocalization of IN-2 with Rab proteins. Experimental methodology (**i**). Cells were baculovirus infected to express Rab5/Rab7-RFP and then HIV-Gag-iGFP infected. At different times, cells were processed for ICC for HIV-1 IN and counterstained with DAPI. Noninfected cells expressing Rab5/7-RFP were analyzed (control). Composite images revealed colocalizations of IN-2 and Rab5-RFP (**j** white arrow) or Rab7-RFP (**k** green arrow) at early and late time points, which are rendered in a 3D image (**l**) and quantified using Pearson's R correlation coefficient (**m** $n = 25$ cells per time point). We used a MOI of 2. In all cases, the means ± S.D. and, where appropriate, the individual values for each experiment are shown. P values are indicated. Scale bars, 2 (**l**), 5 (**a**, **b**), 10 (**e**, **g**, **j**, **k**) μm.

that HIV-Gag-iGFP entering HeLa cells by endocytosis sequentially reaches early and late endosomes to concentrate in the perinuclear region prior to entering NEIs and nucleoplasm.

## VOR complex integrity is essential for productive infection

To determine the molecular mechanism for viral entry into NEIs, we investigated whether HIV-1 IN reached NEIs when VAP-A, its homolog VAP-B, or ORP3 were silenced by short hairpin RNA (shRNA). It was previously demonstrated that VAP-B, in contrast to VAP-A, does not interact with ORP3 and Rab[37]. Gene silencing was carried out by transfection and stable HeLa cell lines were established. As shown by immunoblotting (IB), downregulation of VAP-A to $15.34 ± 2.2\%$ of its normal expression was not compensated by VAP-B, nor did reduction in ORP3 alter VAP-A expression (Supplementary Fig. 5a, b). VAP-A or ORP3 silencing did not affect HeLa growth (not shown), as previously reported for other cell lines[37,38]. After 1-h infection, IN-2 was absent in NEIs of cells lacking VAP-A or ORP3. The expression of the latter proteins was evaluated using specific antibodies, while NEIs were highlighted using ER-GFP fusion protein as a marker of ONM and ER (Fig. 3a, upper panels). In contrast, IN-2 was detected in NEIs of control, i.e., scrambled shRNA- and shVAP-B-transfected, cells. When similar experiments were performed with HeLa cells expressing Rab7-RFP, both IN-2 and Rab7-RFP were detected in NEIs of scrambled shRNA- and shVAP-B-transfected cells as well as of untransfected cells (Fig. 3a, lower panels and Supplementary Fig. 6a–c and Supplementary Movies 5 and 6), but not of cells lacking VAP-A or ORP3 (Fig. 3a, lower panels). In the latter cases, a drastic reduction in IN-2-containing SUN2$^+$ NEIs was observed (Fig. 3b) with much less IN-2 in nucleoplasm than cytoplasm (Fig. 3c). Silencing VAP-A and ORP3 did not interfere with the retrograde transport of IN-2 from cell periphery to perinuclear regions (Fig. 3a), suggesting that the presence of HIV-1 IN/late endosomes in NEIs is important for the nuclear transfer of IN-2. The concentration of IN-2-containing late endosomes in one area around the nucleus (Fig. 3a, d and Supplementary Fig. 6d) suggests they are concentrating at the microtubule-organizing center (MTOC)[53–55].

Since VOR complex and NEIs are potential molecular and cellular paths leading to HIV-1 productive infections, we monitored GFP expression, which required nuclear import of HIV-1 and nuclear export of RNA coding for Gag-iGFP. Upon 6-h infection and 24 h-incubation in virus-free medium, VAP-A or ORP3-deficient cells produced no Gag-iGFP, as observed by immunocytochemistry (ICC) and CLSM, whereas Gag-iGFP was found in the cytoplasm of the scrambled shRNA and VAP-B-deficient cells (Fig. 3e). In Gag-iGFP$^+$ cells, punctate and scattered GFP signals were also observed at the plasma membrane (Fig. 3e), in agreement with a previous report[56]. Flow cytometry (FC) analysis and quantification of GFP$^+$ cells confirmed that productive infection in cells lacking VAP-A and ORP3 is low, i.e., $4.56 ± 2.03\%$ and $2.01 ± 0.09\%$, respectively, in contrast to >35% in scrambled shRNA or VAP-B-deficient cells (Fig. 3f, g), supporting the idea that the integrity of the VOR complex is necessary for this process. Raising the number

of MOI (e.g., 8 instead of 2) partially increased the number of GFP$^+$ cells in VAP-A-deficient cells from $4.56 ± 2.03\%$ to $8.07 ± 1.71\%$, which is lower than the corresponding number in scrambled shRNA, $48.81 ± 4.82\%$, but in line with the remaining amount of VAP-A (Supplementary Fig. 5a).

## Drug-mediated inhibition of the VOR complex impedes nuclear entry of VSV-G-pseudotyped HIV-1 and productive infection

We recently demonstrated that ICZ, but not its major metabolite hydroxy-(H)-ICZ, affects VOR complex integrity in colon carcinoma cells by disrupting the interaction of Rab7 with VAP-A/ORP3, resulting in absence of late endosomes in the NR[38]. Here, we evaluated the impact of ICZ on VOR complex integrity and the resulting effects on IN-2 subcellular localization. We also used a small chemical compound, PRR851, which mimics the effect of ICZ[38], but lacks the reactive triazole moiety necessary for the antifungal activity (Fig. 4a). PRR851 carries the same 2-butanyl side chain bound to the nitrogen at position 2 of the triazolone ring of ICZ. As controls, we used H-ICZ and PRR846. PRR846 lacks the side chain of PRR851 and is inactive against the VOR complex (Fig. 4a)[38]. Upon 90 min incubation with 10 μM ICZ, H-ICZ, PRR compounds, or DMSO (control), HeLa cells were collected and solubilized prior to ORP3 immunoisolation (IS) with a para-magnetic bead-based system and IB for ORP3, VAP-A and Rab7[38]. ICZ or PRR851 blocked Rab7 binding to VAP-A/ORP3, while H-ICZ and PRR846 did not (Fig. 4b). To impede the latter interaction, $IC_{50}$ values for ICZ and PRR851 were 4.9 and 4.8 μM, respectively, which were estimated using various drug concentrations and linear regression (Supplementary Fig. 7a). ORP3 often appeared as a double immuno-reactive band, the slower migrating band being the hyperphosphorylated form (see below and refs. 38,57). Next, cells were preincubated with drugs (10 μM) for 30 min before HIV-Gag-iGFP infection for 1 h. As observed by ICC and/or Rab7-RFP expression using CLSM, disruption of VAP-A/ORP3–Rab-7 interactions resulted in the absence of IN-2 in VAP-A$^+$ NEIs (Fig. 4c); likewise for IN-2 and Rab7-RFP in SUN2$^+$ NEIs (Fig. 4d). Quantification of SUN2$^+$ NEIs containing IN-2 revealed a significant reduction in cells treated with ICZ or PRR851 compared to DMSO- and PRR846-treated cells (Fig. 4e). The nuclear/cytoplasmic ratio of IN-2 was also altered, suggesting a significant reduction of IN-2 in the nucleoplasm (Fig. 4f). Of note, none of the drugs interfered with the retrograde transport of IN-2 from the cell periphery to one or two areas of the perinuclear region (Fig. 4g), as described for VAP-A and ORP3-silencing, suggesting that ICZ or PRR851 inhibits the selective entry of HIV-1 components into NEIs. These data support the view that NEIs act as an intermediate compartment in HIV-1 nuclear transfer. Finally, these microscopy data were validated by subcellular fractionation of 3-h infected cells into cytoplasmic and nuclear fractions followed by IB. The nuclear pool of HIV-1 IN was absent in both ICZ- and PRR851-treated cells (Fig. 4h).

Then, we determined whether ICZ or PRR851 blocked productive infection. HeLa cells were preincubated with drugs (10 μM) for 30 min

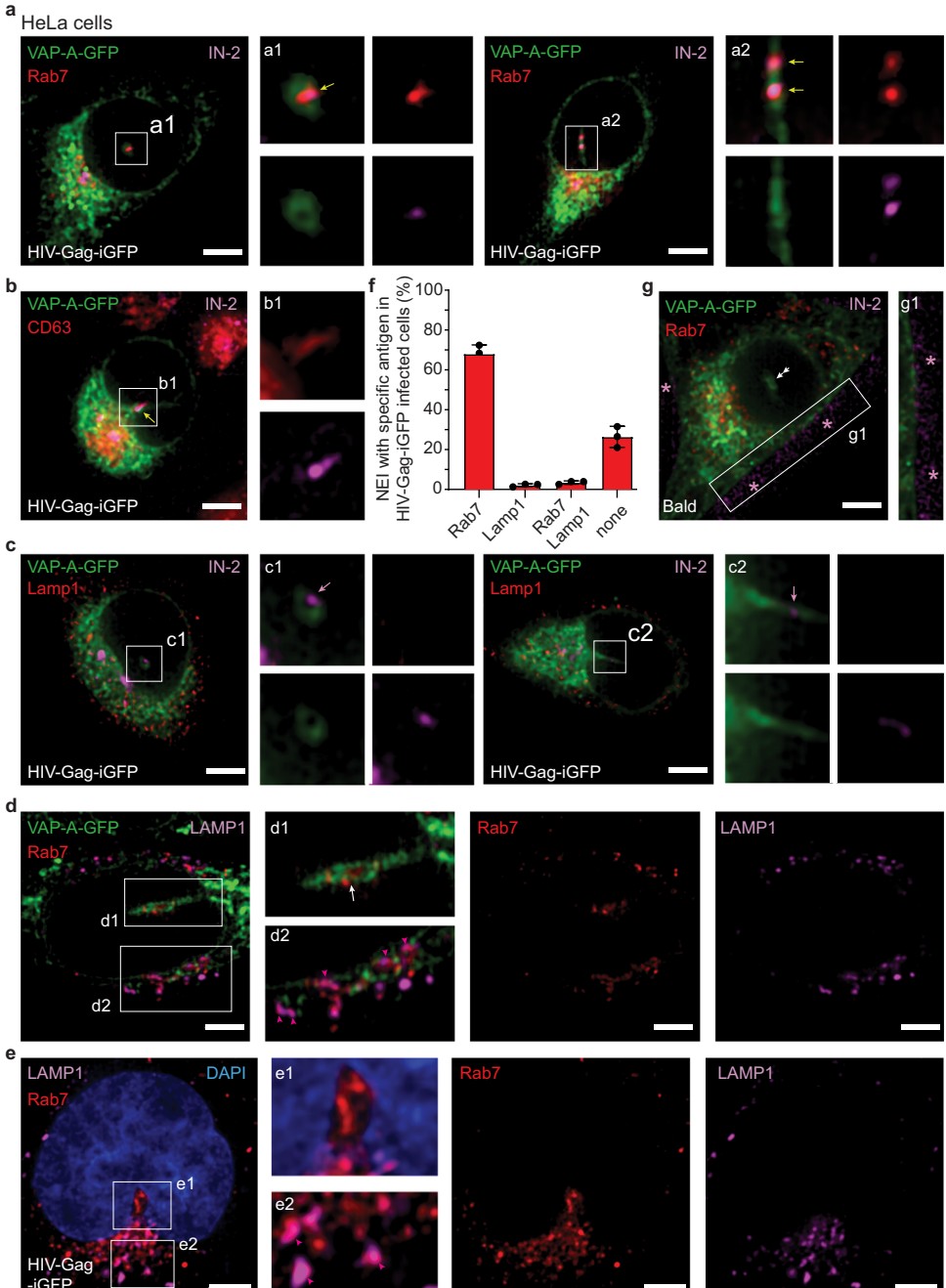

**Fig. 2 | Rab7+ late endosomes containing HIV-1 IN enter NEI.** HeLa cells stably transfected with VAP-A-GFP were 1-h HIV-Gag-iGFP infected prior to immunolabeling for HIV-1 IN and Rab7 (**a**), CD63 (**b**) or Lamp1 (**c**). Single sections show transverse (**a**, **c**) and/or longitudinal (**a–c**) sections of a NEI. Merge and single channels (insets a1, a2, b1, c1, c2) are presented. IN-2 colocalized with Rab7 or CD63, but not Lamp1 in NEIs (yellow and magenta arrow, respectively). **d** Noninfected VAP-A-GFP-expressing HeLa cells were Rab7 and Lamp1 immunolabeled. Single transverse section of a NEI is shown with the merge and single channels as indicated. Insets show the absence of Lamp1 in NEI (d1, white arrow) and the colocalization of Rab7 and Lamp1 in the cytoplasmic compartment (d2, pink arrowhead). **e**, **f** HeLa cells were 1-h HIV-Gag-iGFP infected prior to immunolabeling for Rab7 and Lamp1. Nuclei were counterstained with DAPI. Single longitudinal section of a NEI is shown with the merge and single channels as indicated (**e**). Insets show the absence of Lamp1 in NEI (e1) and the colocalization of Rab7 and Lamp1 in cytoplasmic compartment (e2, pink arrowhead). Quantification of NEIs containing specific antigen(s) as indicated in infected cells (**f**, 50 cells per experiment, $n = 3$, means ± S.D. are shown). **g** VAP-A-GFP-expressing HeLa cells were 1-h infected with "bald" virus, lacking VSV-G, prior to Rab7 and HIV-1 IN immunolabeling. A single section shows transverse section of a NEI. Note that most of IN-2 remains outside the cells (asterisk) and the absence of Rab7 in NEI (double arrow). For a lower power view as well as all sections see below Supplementary Fig. 15c. We used a MOI of 2 or equivalent in the case of the "bald" virus. Scale bars, 5 μm.

before HIV-Gag-iGFP infection (6 h). Afterward, they were incubated for 24 h in virus-free, drug-containing medium before being analyzed by ICC and FC. Both techniques revealed a significant reduction in Gag-iGFP expression in cells treated with ICZ and PRR851, but not with DMSO, PRR846 or H-ICZ (Fig. 4i–k), suggesting that ICZ and PRR851 blocked productive infection by inhibiting HIV-1 IN entry into the NR/nucleoplasm. The effects of PRR851 on the accumulation of HIV-1 IN in NEIs and productive infection were concentration-dependent (Supplementary Fig. 7b). IN-2+ NEIs and GFP expression also decreased in concert ($r = 0.91$) with increasing concentration of PRR851. These

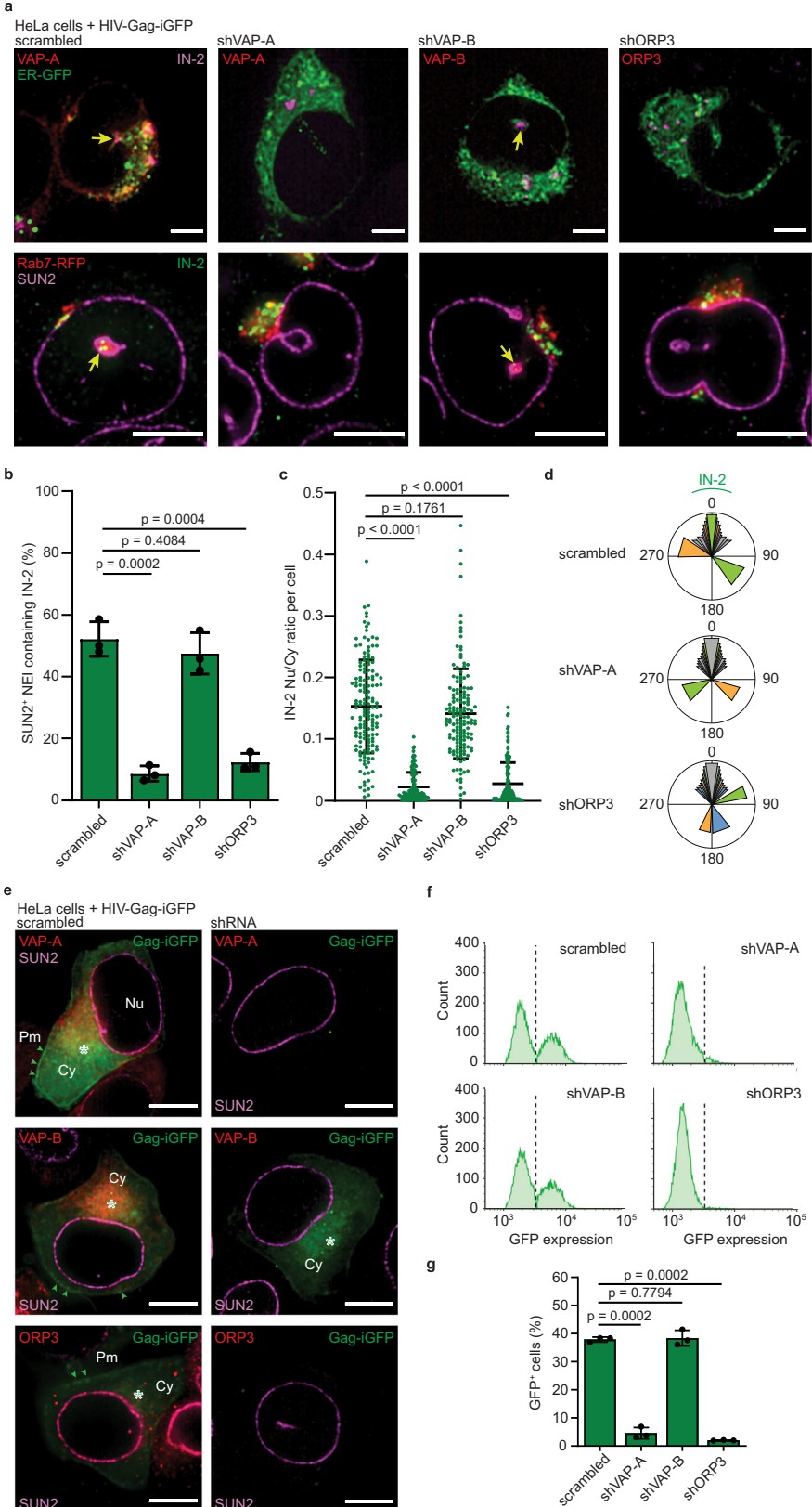

inhibitions were not due to side effects, as PRR851 was not toxic to HeLa cells based on a MTS tetrazolium assay. Up to 30 µM, neither ICZ, PRR851, nor PRR846 had major effects on cell growth after 48 h compared with DMSO. Only a ≈10% reduction was observed, whereas H-ICZ caused ≈60% inhibition (Supplementary Fig. 8a). At 100 µM, ICZ and H-ICZ inhibited cell growth by 50% and 100%, respectively, while

the effect of PRR851 and PRR846 was modest, i.e., ≈20%, suggesting that PRR851 inhibits the productive infection without impacting cell growth.

Overall, this first set of data demonstrated that VSV-G-pseudotyped HIV-1 infection recapitulated the nuclear entry of EVs in which inhibition of VOR complex formation prevented productive

**Fig. 3 | Silencing the VOR complex inhibits HIV-1 integrase transport into NR and productive infection. a–d** HeLa cells stably transfected with plasmids carrying scrambled shRNA, shVAP-A, shVAP-B or shORP3 and expressing either ER-GFP (**a**, upper panels) or Rab7-RFP (**a**, lower panels, **b–d**) were 1-h HIV-Gag-iGFP infected prior to immunolabeling for either VAP-A, VAP-B, ORP3 or SUN2 and HIV-1 IN (IN-2) as indicated. Samples were observed by CLSM. Single sections are displayed. IN-2 was detected in ER-GFP+ or SUN2+ NEI of scrambled and shVAP-B cells and quantified (**a** arrow; **b** 50 NEIs per experiment, *n* = 3). The ratio of IN-2 in the nucleoplasm versus cytoplasm per cell was determined (**c** *n* = 150 cells from three independent experiments). The distribution of IN-2 in perinuclear region of scrambled shRNA, VAP-A-, or ORP3-silenced HeLa cells is shown in a circular histogram (**d** 10 cells per cell line; color code indicates an individual cell). [For Methods, see Supplementary Fig. 6d]. Cells with dual IN-2 localization are in color. **e–g** HeLa cells were 6-h HIV-Gag-iGFP infected, washed and incubated for 24 h prior to ICC (**e**) and FC (**f**, **g**). Representative fluorescent images for the indicated proteins (**e**) and FC histograms (**f**) are displayed. Scrambled shRNA-transfected, noninfected cells were used for gating. Note the reduction in Gag-iGFP expression in cells lacking VAP-A and ORP3. The percentage of GFP+ cells is plotted (**g** *n* = 3). Asterisk and arrowhead indicate Gag-iGFP in the cytoplasm (Cy) and plasma membrane (Pm), respectively. Nu nucleus. We used a MOI of 2. Means ± S.D. and individual values for each experiment (**b**, **g**) or cell (**c**) are presented. *P* values are indicated. Scale bar, 5 (**a**, top) or 10 (**a**, bottom, **e**) μm.

infection. This information reinforces the relationship between viral particles and EVs.

## Drug-mediated inhibition of the VOR complex impedes the productive infection in activated CD4+ T cells infected with native Env-pseudotyped HIV-1 virus

All data described so far involving the VOR complex-dependent pathway for the nuclear transfer of viral components refer to HIV-1 pseudotyped with VSV-G. To obtain information about the native virus, we employed p89.6 ΔE ΔN SF-EGFP pseudotyped with autologous HIV 89.6 Env (abbr. HIV-89.6-EGFP) containing the bona fide native envelope glycoprotein[58]. Like virus particles pseudotyped with VSV-G (see above), those with native Env are poorly fluorescent (mean fluorescence per particle is 4.91 ± 1.32 RFU, *n* = 42) (Supplementary Fig. 1b). Two cell types were used as recipients, HeLa cells ectopically expressing CD4 and human primary CD4+ T cells, which are one of the main targets of HIV-1. The data obtained with CD4+ HeLa cells are described in Supplementary Note 1, as they essentially recapitulate those observed with HIV-1 pseudotyped with VSV-G. This includes the impact of drugs on productive infection (Supplementary Figs. 9 and 10), the role of CD4 and endocytosis (Supplementary Figs. 10 and 11) and the induction of NEIs. In addition, they revealed the subtle effect of the infection method, e.g., RetroNectin versus spinoculation, on virus uptake in CD4-deficient cells (Supplementary Fig. 12, see below).

To assess whether the novel VOR complex-dependent pathway and the impact of drugs on it are operational in cells endogenously expressing CD4, we used primary human T cells, obtained from a commercial vendor or prepared in one of our laboratories (see Methods). First, we activated them by 48-h incubation with phytohemagglutinin (PHA) and interleukin-2 (IL-2). Then, we preincubated them with drugs (10 μM) for 30 min before 6 h of RetroNectin-based infection with HIV-89.6-EGFP, followed by an additional 48 h in drug-containing medium. Infected cells were analyzed by FC for EGFP expression. They were gated for CD3 and CD4 (Supplementary Fig. 13). Similarly to data obtained with VSV-G pseudotyped virus in HeLa cells or native Env-pseudotyped HIV-1 virus in CD4+ HeLa cells, ICZ and PRR851 blocked the EGFP expression in PHA/IL-2-activated CD4+ T cells (Fig. 5a, b). Of note, PRR851 did not significantly inhibit T cell growth, as evaluated by the MTS tetrazolium assay (Supplementary Fig. 8b).

To further investigate the timing of drug action, two additional protocols were established (Fig. 5c). In addition to controls, i.e., no drug (protocol #i) and constant drug presence (protocol #ii), PRR851 was removed after 6 h of infection (protocol #iii) or added after it (protocol #iv). Interestingly, the inhibitory effect of PRR851 was still observed after its removal just after 6-h infection (Fig. 5d, #iii), but not when the drug was added after infection (Fig. 5d, #iv), suggesting that the presence of PRR851 before and during, but not after infection, is essential.

In addition to the expression of the *EGFP* reporter gene, which is not under the control of the HIV-1-LTR promoter in our plasmid construct (i.e., p89.6 ΔE ΔN-SF-EGFP), but is dependent on the internal spleen focus virus promoter, we monitored the expression of the viral structural protein Gag. To this end, after the 6-h infection, PHA/IL-2 activated T cells were incubated for 48, 72, and 96 h before being analyzed by FC for EGFP and HIV-1 Gag. The latter was detected using an anti-Gag antibody. Interestingly, after 48 h, the percentage of EGFP-expressing cells was similar to previous experiments (Fig. 5b), whereas the number of Gag-expressing cells as well as double-positive cells were significantly lower (Fig. 5e, f). Such differential expression between EGFP and Gag expression was previously observed and explained by an early expression of EGFP, i.e., before the expression of HIV-1 LTR-driven genes[59]. In contrast, after 72 and 96 h, the number of double-positive cells increased, whereas single EGFP+ cells decreased (Fig. 5e, f). Regardless of time course, addition of PRR851 blocked their expression, suggesting that the novel VOR complex-dependent pathway regulates nuclear transfer of viral components enabling expression of structural viral proteins highlighting productive infection.

Finally, to rule out any potential artifacts produced by the RetroNectin-based infection method, we applied an alternative method, namely spinoculation, which concentrates the cells of interest with the viral particles by a gentle centrifugation step at 4 °C[60]. This technique is widely used in the field of HIV-1. Again, spinoculation-based infection of PHA/IL-2-activated T cells with HIV-89.6-EGFP produced EGFP expression in a drug-dependent manner (Fig. 5g), as shown above with RetroNectin.

## HIV-1 infection promotes NEI formation through the VOR complex—Lessons from CD4+ T cells

Unlike adherent HeLa cells, CD4+ T cells grow in suspension, which has an impact on their morphology: the former are very spread out while the latter are rounded. These morphological alterations could influence the formation and/or induction of NEIs. This prompted us to re-examine the impact of viral infection on NEIs. Surprisingly, both types of NEIs were rarely found in noninfected, PHA/IL-2-activated T cells, whereas the number of cells harboring type II, not type I, NEIs increased after 3- or 6-h infection (Fig. 6a–d), indicating that the infection stimulates NEI formation. It is noteworthy that the proportion of cells with induced type II NEIs is similar to the proportion of EGFP+ cells (Fig. 5b vs Fig. 6b, d), suggesting that the former event is essential for the viral cycle. A similar observation was made with HeLa cells infected with HIV-Gag-iGFP (Fig. 1f vs Fig. 4k). In line with the previous data, pretreatment of PHA/IL-2-activated T cells with PRR851 reduced HIV-1-induced NEIs (Fig. 6c, d and Supplementary Fig. 14a), suggesting again that this new nuclear path is essential for nuclear entry. We then infected PHA/IL-2-activated T cells with different doses of virus (MOI ranging from 0.2 to 4) and determined the percentage of cells with induced NEIs relative to the amount of EGFP+ cells. First, we observed a consistent increase in both events with increasing virus concentration (Fig. 6e). Second, regardless of the amount of virus applied, there was a tight correlation (*r* = 0.88) between the number of cells showing NEIs and EGFP expression (Fig. 6e), confirming that NEI induction is essential.

The presence of NEIs in infected T cells was confirmed by Lamin B1 immunolabeling (Fig. 6f). In addition to the induction of NEIs, the overall shape of the nuclear membrane was strongly altered with the appearance of indentations (Fig. 6g). Nuclear solidity, defined as the

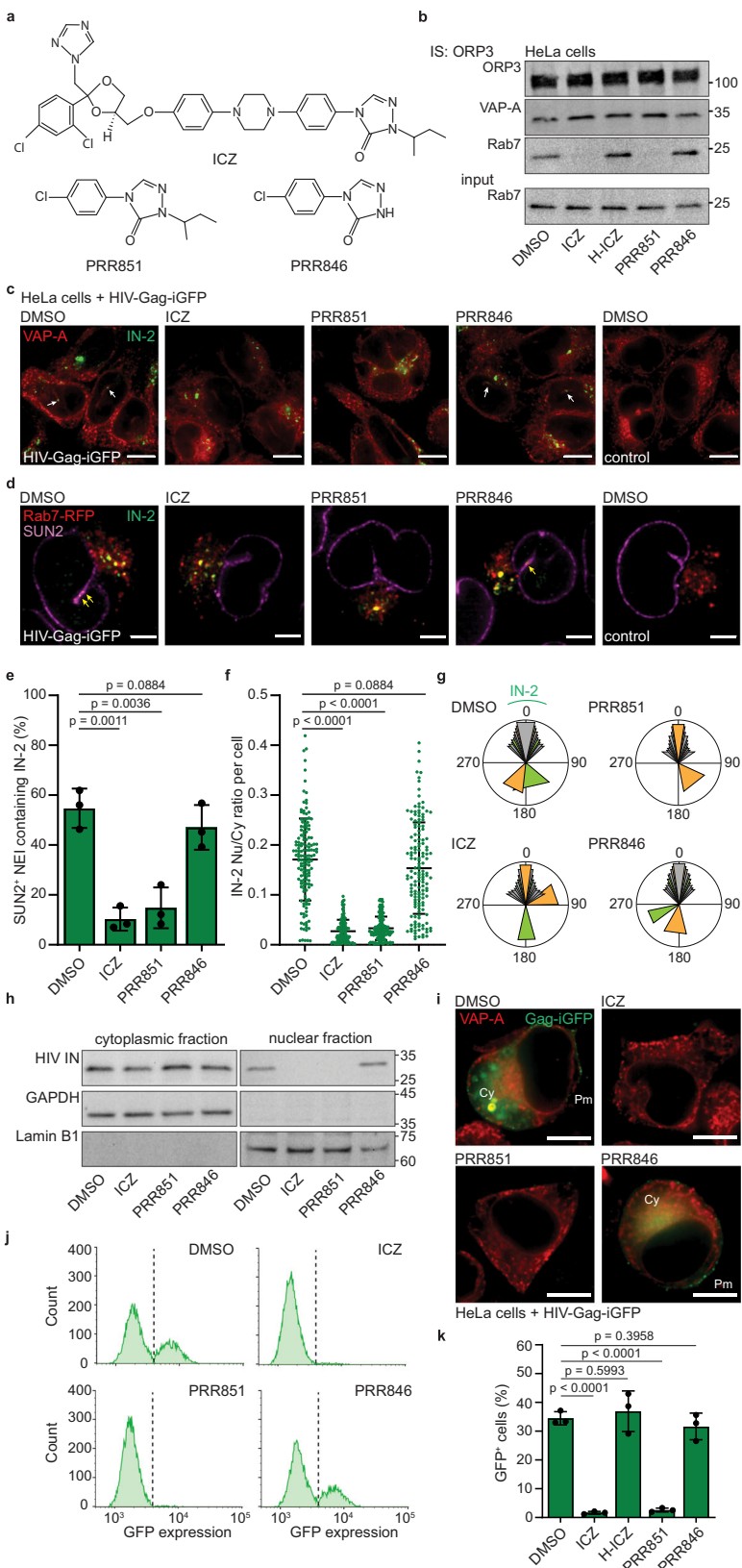

ratio of the measured area of the nucleus to the area of its convex hull shape (Fig. 6g), was reduced in infected compared to noninfected cells, whereas PRR851 attenuated this effect (Fig. 6h).

When PHA/IL-2-activated T cells were infected with the corresponding "bald" virus, none (or very few) cells containing type II NEIs were observed after 3 h of infection (Fig. 6i, j). Again, this reflects the

fact that a limited number of cells were infected as few of them showed cytoplasmic IN-2 and those that exhibited a lower level of IN-2 than HIV-89.6-EGFP-infected cells (Supplementary Fig. 14b, c). Most of the IN-2 signal carried by the "bald" virus remained at the periphery of the cells in a random distribution (Fig. 6i). In contrast, IN-2 was found in the cytoplasmic compartment and in the induced NEIs in HIV-89.6-EGFP-

**Fig. 4 | The VOR complex inhibition impedes the HIV-1 integrase transport into NR and productive infection. a** Drug structures. **b** HeLa cells were 90-min incubated with DMSO or various drugs (10 μM) as indicated, detergent solubilized and ORP3 IS. Bound fractions and Rab7 input (1/50) were probed for ORP3, VAP-A, and Rab7 by IB. **c** HeLa cells were 30-min pretreated with DMSO or drugs (10 μM) before 1-h HIV-Gag-iGFP infection in the presence of drugs. As a control, cells were noninfected. Samples were VAP-A and HIV-1 IN (IN-2) immunolabeled. Single sections are displayed. IN-2 was detected in NEIs of DMSO/PRR846-treated cells (arrow). **d**–**g** Rab7-RFP-expressing HeLa cells were treated as in **c**, and IN-2 and SUN2 immunolabeled. The presence of IN-2 in SUN2⁺ NEI (**d** arrow) was quantified (**e** 50 NEIs per experiment, *n* = 3). The IN-2 nucleoplasm/cytoplasm ratio was determined (**f** *n* = 150 cells from three independent experiments). Localization of IN-2 in the perinuclear region is shown using a circular histogram (**g** 10 cells/condition; color code indicates an individual cell). [For Methods, Supplementary Fig. 6d]. **h** Drug-

treated HeLa cells were 3-h HIV-Gag-iGFP infected, fractionated into cytoplasmic and nuclear fractions and probed for HIV-1 IN by IB. A quarter and the total material of the cytoplasmic and nuclear fractions were loaded. Cytoplasmic GAPDH and nuclear Lamin B1 were used as controls and run in parallel. They are derived from the same experiment (**b**, **h**). Molecular mass markers (kDa) are indicated. **i**–**k** HeLa cells were 30-min pretreated with DMSO or drugs before 6-h HIV-Gag-iGFP infection. They were 24-h incubated in the presence of drugs and then processed for ICC (**i**) or FC (**j**, **k**). For ICC, fixed-cells were VAP-A immunolabeled, while they were trypsinized for FC. Representative images (**i**) and FC histograms (**j**) are displayed. DMSO-treated, noninfected cells were used for gating. The GFP⁺ cells were quantified (**k** *n* = 3). We used a MOI of 2. Means ± S.D. and individual values for each experiment (**e**, **k**) or cell (**f**) are presented. *P* values are indicated. Scale bars, 10 (**c**, **i**) or 5 (**d**) μm.

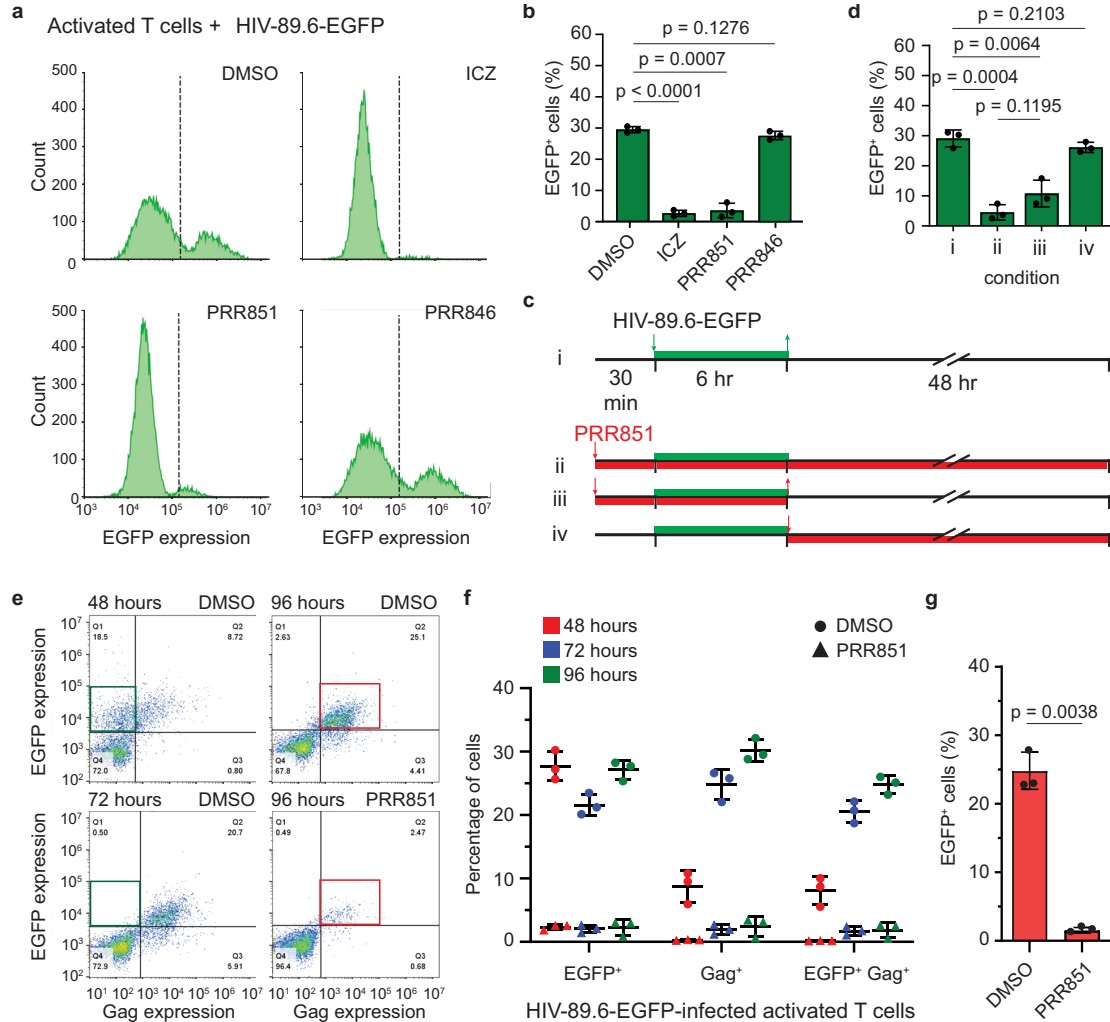

**Fig. 5 | PRR851 impedes the productive infection of native Env-pseudotyped HIV-1 in activated CD4+ T cells. a, b** PHA/IL-2-activated primary CD4⁺ T cells were 30-min pretreated with 10 μM ICZ, PRR851, PRR846, or DMSO before RetroNectin-based, 6-h HIV-89.6-EGFP infection. Cells were washed and incubated for 48 h in the presence of drugs. Samples were processed for FC and histograms are displayed (**a**). DMSO-treated, noninfected cells were used for the gating. The percentage of EGFP⁺ cells is plotted (**b** *n* = 3). **c, d** Representation of alternative protocols (**c**) and quantification of the EGFP expression in infected PHA/IL-2-activated CD4⁺ T cells by FC (**d** *n* = 3 per condition). In addition to negative and positive controls, i.e., without (#i) or in constant presence of PRR851 (10 μM) (#ii), respectively, the drug was removed after 6-h infection (#iii) or added after (#iv). **e, f** PHA/IL-2-activated CD4⁺ T cells were 30-min pretreated with DMSO or 10 μM PRR851 before RetroNectin-

based, 6-h HIV-89.6-EGFP infection, washed and further incubated for 48, 72 or 96 h in the presence of drugs. Samples were processed for FC for the expression of EGFP and HIV-1 Gag. Representative FC histograms are displayed (**e**). DMSO-treated, noninfected cells were used for the gating. The percentage of EGFP⁺, Gag⁺, or double-positive cells is plotted (**f** *n* = 3). Note that single EGFP⁺ cells (green boxes) decrease during incubation time, while double-positive cells increase in a drug-dependent manner (red boxes). **g** PHA/IL-2-activated CD4⁺ T cells were 30-min pretreated with DMSO or 10 μM PRR851 before 6-h HIV-89.6-EGFP infection by spinoculation instead of the RetroNectin method. After 48 h, samples were processed for FC and the percentage of EGFP⁺ cells is plotted (*n* = 3). We used a MOI of 2. In all cases, means ± S.D. and individual value are presented. *P* values are indicated.

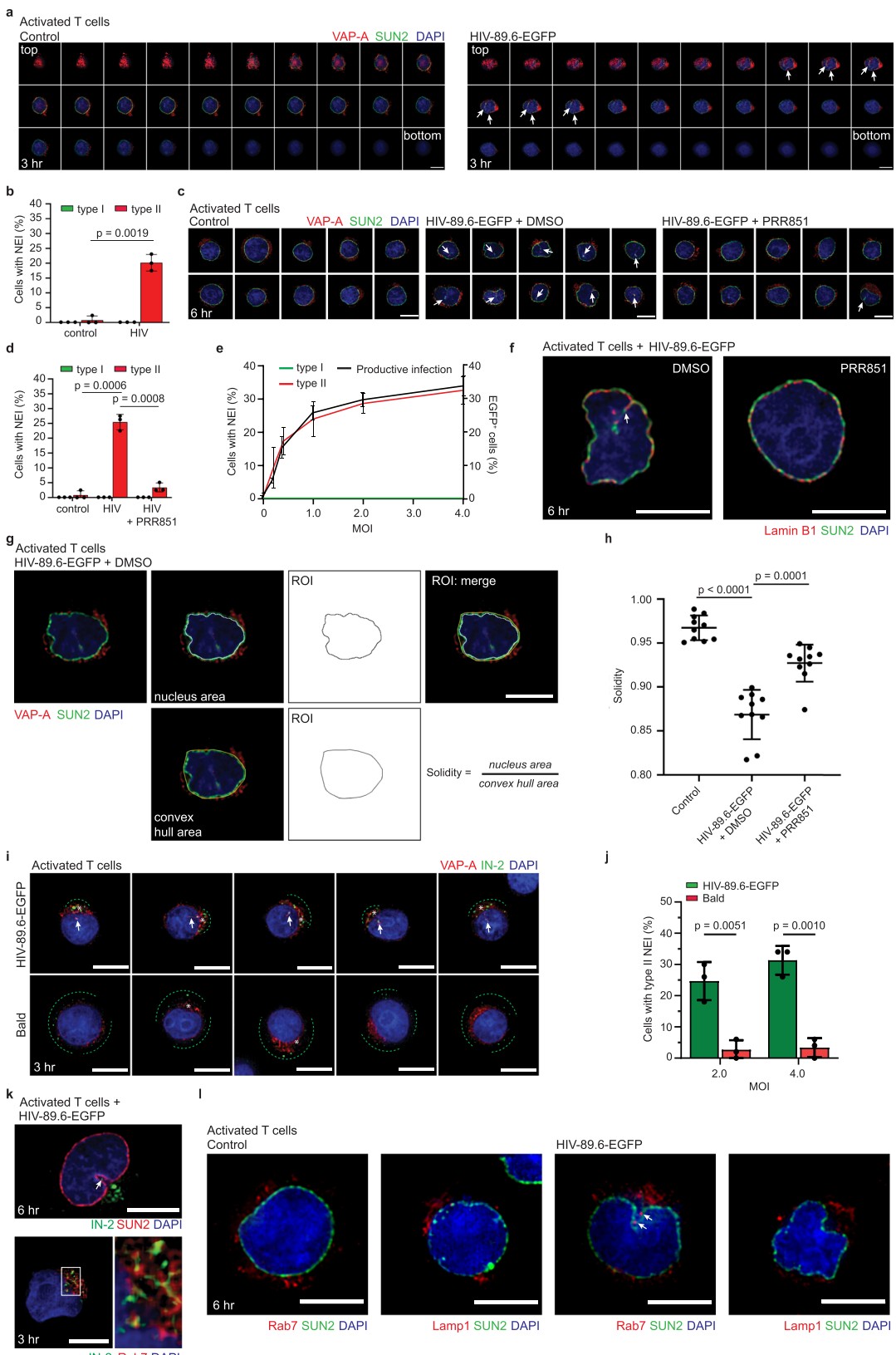

infected cells, suggesting that loading of late endosomes with viral components is important for these processes in CD4$^+$ T cells (Fig. 6i, k, upper panel), which are devoid of NEIs without infection. As observed with VSV-G-pseudotyped HIV-1 virus, IN-2 co-localized with Rab7$^+$ late endosomes (Pearson's R correlation coefficients: 0.74 ± 0.13, $n = 10$), where they accumulated at one pole of the nucleus (Fig. 6k, lower

panel and Supplementary Fig. 15a). Similarly, only Rab7$^+$ structures, but no Lamp1$^+$ ones, were detected in the NEIs of CD4$^+$ T cells infected with native Env virus, as shown above with VSV-G pseudotyped HIV-1 virus (Fig. 6l), consistent with an earlier report[61].

To confirm that the virus-induced formation of NEIs is mediated by the VOR complex, VAP-A-ORP3-Rab7 interactions were

**Fig. 6 | Native Env-pseudotyped HIV-1 infection of CD4+ T cells triggers type II NEIs biogenesis. a–f** PHA/IL-2-activated CD4$^+$ T cells were HIV-89.6-EGFP infected for 3 (**a**, **b**) or 6 (**c**–**e** left y axis, **f**) hours prior to immunolabeling for SUN2 and VAP-A (**a**–**e**) or Lamin B1 (**f**). Nuclei were counterstained with DAPI. In some experiments, cells were 30-min pretreated with PRR851 (10 μM) (**c**, **d**). Drugs were present during infection. As a control, cells were noninfected. They were observed by CLSM and serial sections of a representative cell (**a**) or single section of 10 cells are presented (**c**). Alternatively, after 6-h infection, cells were 48-h incubated prior to FC (**e** right y axis, n = 3). The percentage of cells with type I or II NEIs was quantified (**b**, **d**, **e**, 50 cells per experiment, n = 3). Note the presence of NEIs in infected cells without PRR851 (**a** right, **c** middle, **f** arrow). **g**, **h** The solidity was evaluated in noninfected and 6-h infected cells. After SUN2 and VAP-A immunolabeling, the surface areas were evaluated (**g**), and the resulting solidity presented (**h**, n = 10 cells).

**i, j** Activated T cells were 3-h infected with HIV-89.6-EGFP or corresponding "bald" virus prior to VAP-A and HIV-1 IN (IN-2) immunolabeling. Single section of representative cells is shown (**i**). IN-2 in induced NEIs (arrows) or in the cytoplasm (asterisks) are indicated. Green dashed lines indicate IN-2 distribution around the cell surface. The percentage of cells with type II NEIs were quantified (**j** ≥50 cells per condition, n = 3). **k, l** The presence of IN-2 in NEIs (arrow) and late endosomes of 6 or 3 h-infected cells was observed after immunolabeling for IN-2 and SUN2 (**k** upper panel) or Rab7 (**k** lower panel), respectively. Alternatively, cells were immunolabeled for Rab7 or Lamp1 and SUN2 (**l**). Note that Rab7 (arrows), but not Lamp1, is present in NEIs. We used a MOI of 2 or equivalent for the "bald" virus, unless otherwise stated. In all cases, means ± S.D. and individual values are shown. P values are indicated. Scale bars, 5 μm.

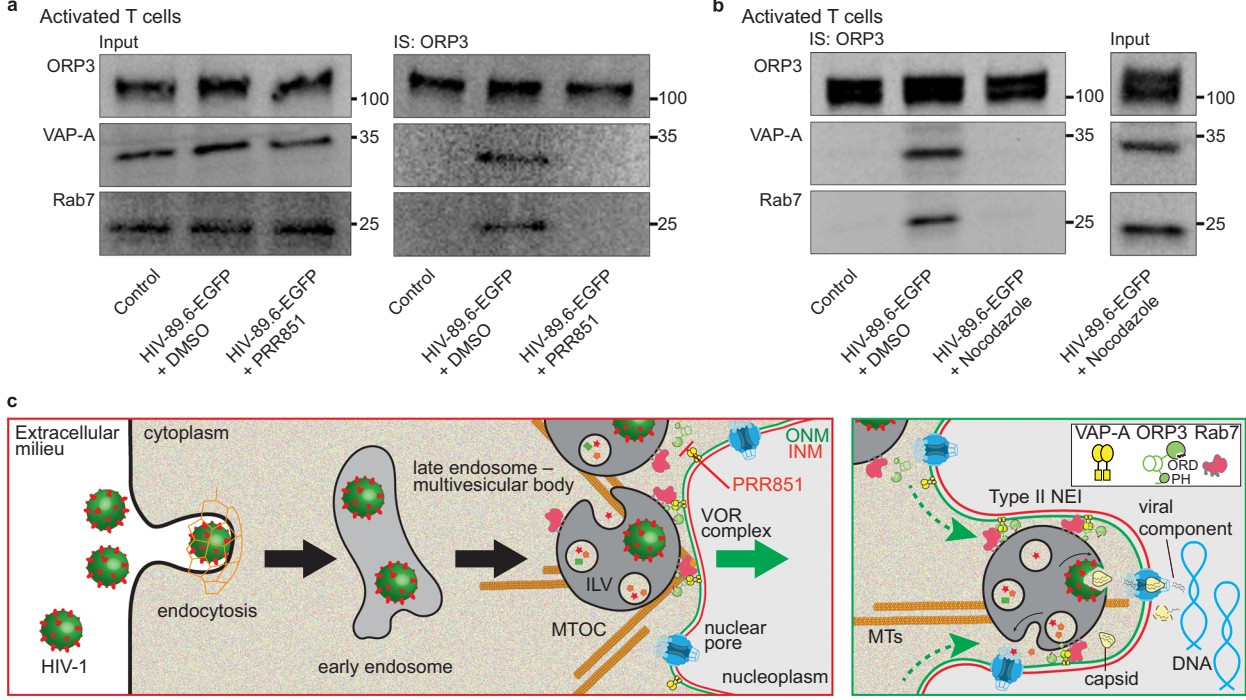

**Fig. 7 | HIV-1 infection promotes the VOR complex formation in a microtubule-dependent manner. a** PHA/IL-2-activated CD4$^+$ T cells were noninfected (control) or 6-h HIV-89.6-EGFP infected after 30-min pretreatment with DMSO or 10 μM PRR851. The drug was present during infection. Cells were then solubilized and subjected to ORP3 IS. The bound fractions and the input (1/50) were probed for ORP3, VAP-A, and Rab7 by IB. **b** Cells were 5-min pretreated with DMSO or nocodazole (1 μM) and then noninfected (control) or 1-h HIV-89.6-EGFP infected, solubilized and processed for ORP3 IS and ORP3, VAP-A and Rab7 IB. All samples came from the same experiment and were run in parallel (a, b). Molecular mass markers (kDa) are indicated. Representative experiments are shown. We used a MOI of 2. Note that VOR complex formation only occurs after HIV-1 infection and is hindered

by PRR851 and nocodazole. **c** Representation of the induction of type II NEIs by virus-laden late endosomes, a process mediated by the interaction of VOR complex proteins, namely ONM-associated VAP-A, cytoplasmic ORP3, and late endosome-associated Rab7 (left panel). Release of viral components from late endosomes into the cytoplasmic core of induced NEIs at the vicinity of the nuclear pore would facilitate their transfer to the nucleoplasm (right panel). PRR851 inhibits the interaction of the VOR complex proteins, and hence the NEI formation. ILV, intra-lumenal vesicles associated with late endosome/multivesicular body; INM/ONM, inner/outer nuclear membrane; MTOC, microtubule-organizing center; MTs, microtubules.

evaluated by the IS of ORP3 and IB of the three proteins of interest. In a PRR851-dependent process, viral infection promoted their interactions (Fig. 7a). Remarkably, no interaction between VAP-A, ORP3, and Rab7 was detected in control noninfected, PHA/IL-2-activated CD4$^+$ T cells (Fig. 7a). The latter data contrast with those obtained with HeLa cells (Fig. 4b), which contain NEIs in their native state (Fig. 1f and Fig. 4c, d), implying that these interactions occur selectively in NEIs, and/or induced new NEIs. These observations are in agreement with our previous data using the fluorescence

resonance energy transfer that showed a close contact of each protein pair (i.e., VAP-A/ORP3, ORP3/Rab7, and VAP-A/Rab7) at the nuclear membrane of NEIs[37]. An intact microtubule network, as demonstrated by the treatment with nocodazole – a microtubule de-polymerizing agent – prior and during the infection, is important for the interaction of VOR complex proteins (Fig. 7b). These data implicate microtubules in the retrograde transport of virus-laden late endosomes from cell periphery toward perinuclear area where formation of the VOR complex occurs (Fig. 7c, see also Discussion).

It remains to be demonstrated whether microtubules could be the driving force for NEI formation.

### ORP3 hyperphosphorylation is necessary for the VOR complex integrity and nuclear entry – Lessons from quiescent CD4+ T-lymphocytes

Next, we performed the same set of experiments with quiescent CD4+ T cells, which were not exposed to PHA and IL-2. In agreement with other studies[62], almost no EGFP production was observed upon exposure to HIV-89.6-EGFP, either by ICC or FC (Fig. 8a, b). Less than 2% of cells were positive for EGFP (mean of 0.9%, $n = 3$, Fig. 8c). Viral particles were nonetheless internalized as shown by HIV-1 IN and Rab7 immunolabeling (Fig. 8d, Pearson's R correlation coefficients: $0.60 \pm 0.08$, $n = 10$). Again, virus-laden Rab7+ late endosomes accumulated in one nuclear pole of infected quiescent CD4+ T cells, whereas Rab7+ late endosomes were more evenly distributed around the nucleus in noninfected cells (Fig. 8e, f). The latter situation was also observed in noninfected HeLa cells or in "bald" virus-infected HeLa cells (Supplementary Fig. 15b, c, see corresponding legend), confirming that loading the late endosomes with viral particles promoted their translocation to MTOC.

Although the virus was endocytosed in quiescent CD4+ T cells and the late endosomes accumulated near one nuclear pole, no significant increase in the number of type II NEIs was observed, in sharp contrast to PHA/IL-2-activated CD4+ T cells (Fig. 8g), suggesting that an intrinsic factor triggered by PHA and IL-2-mediated activation is required to induce NEI biogenesis. To assess the potential lack of VOR complex protein interactions in noninfected and HIV-89.6-EGFP-infected quiescent cells, ORP3 was IS and the bound fractions were analyzed by IB. PHA/IL-2-activated CD4+ T cells were used for comparison. In noninfected cells, no VOR protein interactions were detected, independent of CD4+ T cell activation (Fig. 8h). In infected quiescent CD4+ T cells, VAP-A and Rab7 did not bind to ORP3, contrasting with infected PHA/IL-2-activated CD4+ T cells (Fig. 8h). Indeed, the VAP-A/ORP3 immunoreactivity ratio was significantly reduced in the former (Fig. 8i). Moreover, in quiescent CD4+ T cells, the expression level of ORP3, but not VAP-A and Rab7, was remarkably reduced and no hyperphosphorylation was observed (Fig. 8h). In gel electrophoresis, the hyperphosphorylated form of ORP3 corresponds to a slower migrating band, as shown by λ-phosphatase treatment of detergent lysate of PHA/IL-2-activated CD4+ T cells (Fig. 8j). Knowing that this post-translational modification regulates ORP3 binding to VAP-A and Rab7[38,57], the absence of ORP3 hyperphosphorylation may explain, at least in part, the resistance of quiescent CD4+ T cells to HIV-1 infection. Altogether, hyperphosphorylation of ORP3 which occurs during CD4+ T cell activation could promote its interaction with VAP-A and, in conjunction with HIV-1 infection, stimulate late endosome-associated Rab7 binding in the perinuclear region, leading to the formation of type II NEIs, and eventually the nuclear transfer of viral components (Fig. 8k).

### Protein kinase C is involved in ORP3 phosphorylation and the VOR complex-dependent upstream events

Lastly, we sought the potential protein kinase involved in ORP3 hyperphosphorylation. The major candidate protein is protein kinase C (PKC), which was previously suggested to be involved in ORP3 phosphorylation[39–41]. To that end, we incubated CD4+ T cells with Sotrastaurin (AEB071; 40 μM), a cell-permeant inhibitor of PKC[63], during their 48-h activation with PHA and IL-2 and the subsequent 6-h infection with HIV-89.6-EGFP. As previously reported[64], cell proliferation was reduced during activation under these conditions (data not shown). Nevertheless, we found by ICC that the infection-triggered induction of NEIs was hindered by Sotrastaurin (Fig. 9a, b). The overall architecture of the nucleus, as assessed by nuclear solidity, remained uniform despite the infection, with a solidity ratio similar to that in

PRR851-treated cells (Fig. 9c; see above). Consistent with these data, no EGFP expression was observed (Fig. 9d). Unexpectedly, after cell solubilization and IB, we observed that ORP3 expression was significantly reduced in the presence of Sotrastaurin, irrespective of the infection (Fig. 9e, top panels). Rab7 was also partially reduced, while VAP-A expression was unchanged in the presence of the PKC inhibitor. ORP3 hyperphosphorylation was also inhibited. Indeed, both the expression of ORP3 and its phosphorylation levels were similar to those observed in quiescent CD4+ T cells (Fig. 8h). Consequently, no interaction between ORP3 and VAP-A or Rab7 were observed after the IS of ORP3 (Fig. 9e, bottom panels).

## Discussion

The results of this study suggest that the view of HIV-1 endocytosis in CD4+ T cells as a dead-end pathway leading to degradation deserves reconsideration[14,65,66]. This study adds three major insights into how viruses reach and enter cell nuclei: (i) viruses are packaged into late endosomes that move to the MTOC, and a minute fraction of them translocate to, and/or induce, type II NEIs; (ii) the interaction between VAP-A, ORP3, and Rab7 is required for the latter step, which is a prerequisite for HIV-1 nuclear entry; and (iii) PKC-mediated ORP3 hyperphosphorylation occurring during the CD4+ T cell activation is essential for these processes (Fig. 8k).

These conclusions are based on HIV-1 infection, irrespective of the env proteins (VSV-G or *Env* gene products), of two distinct cell types, heterologous HeLa cells and native primary CD4+ T cells. The former cells grow in adherence, while the latter are in suspension. These considerations are important since it has been shown that phosphorylation of ORP3 stimulates, at the plasma membrane, its interaction with ER-associated VAP-A and controls cell-substrate adhesion by modulating β1-integrin activity[39–41]. The latter phenomena could not occur in non-adherent CD4+ T cells, and hence support the action of hyperphosphorylated ORP3 at the ONM.

Promotion of hyperphosphorylation of ORP3 by PKC, as suggested by Sotrastaurin experiments, is in agreement with previous reports showing that ORP3 is a PKC target and phosphorylation stimulates its interaction with VAP-A[39–41], which in turn regulates Rab7 binding (Fig. 8k). Thus, PKC-mediated hyperphosphorylation of ORP3 in PHA/IL-2-activated CD4+ T cells could participate in virus-induced NEI formation and contribute to their permissiveness among other factors, while quiescent CD4+ cells were refractory. As PKC activity is required for CD4+ T cell activation and proliferation[67–69], the fact that ORP3 status is similar between quiescent and Sotrastaurin-incubated cells during PHA/IL-2-induced activation could be more than coincidental. Little is known about ORP3 function(s), but its knockout was reported to impact the expansion of lymphoid progenitors and favor aneuploidy[70]. Further studies should investigate the impact of the absence of ORP3 on HIV-1 infection in vivo.

Only a minute fraction of the late endosomes loaded with viral particles was found in the NR, which may explain why previous work on viral entry into the nucleus has overlooked this pathway. Indeed, it is necessary to scan the entire nuclear compartment to visualize the viral components within type II NEIs, which themselves occupy a limited volume. The endosome-based, NEI-dependent pathway for HIV-1 infections suggests some hypotheses about the timing and location of interactions with cytoplasmic co-factors and components of the nuclear membrane. Nuclear import could be facilitated by the exit of viral components from the late endosomes, i.e., after HIV-1 fusion with the limiting membrane, and their concentration in the core of the NEIs, which could create a protective microenvironment and favor their interactions with cytosolic factors and NPC components (e.g., Nup358, Nup153) en route to nucleoplasm[26,71–73]. Likewise, the exclusion of Lamp1+ lysosomes from NEIs would prevent late endosome-lysosome fusion. The segregation of HIV-1 particles and Lamp1+ lysosomes has been previously noted in CD4+ T cells[61]. Determination of the binding

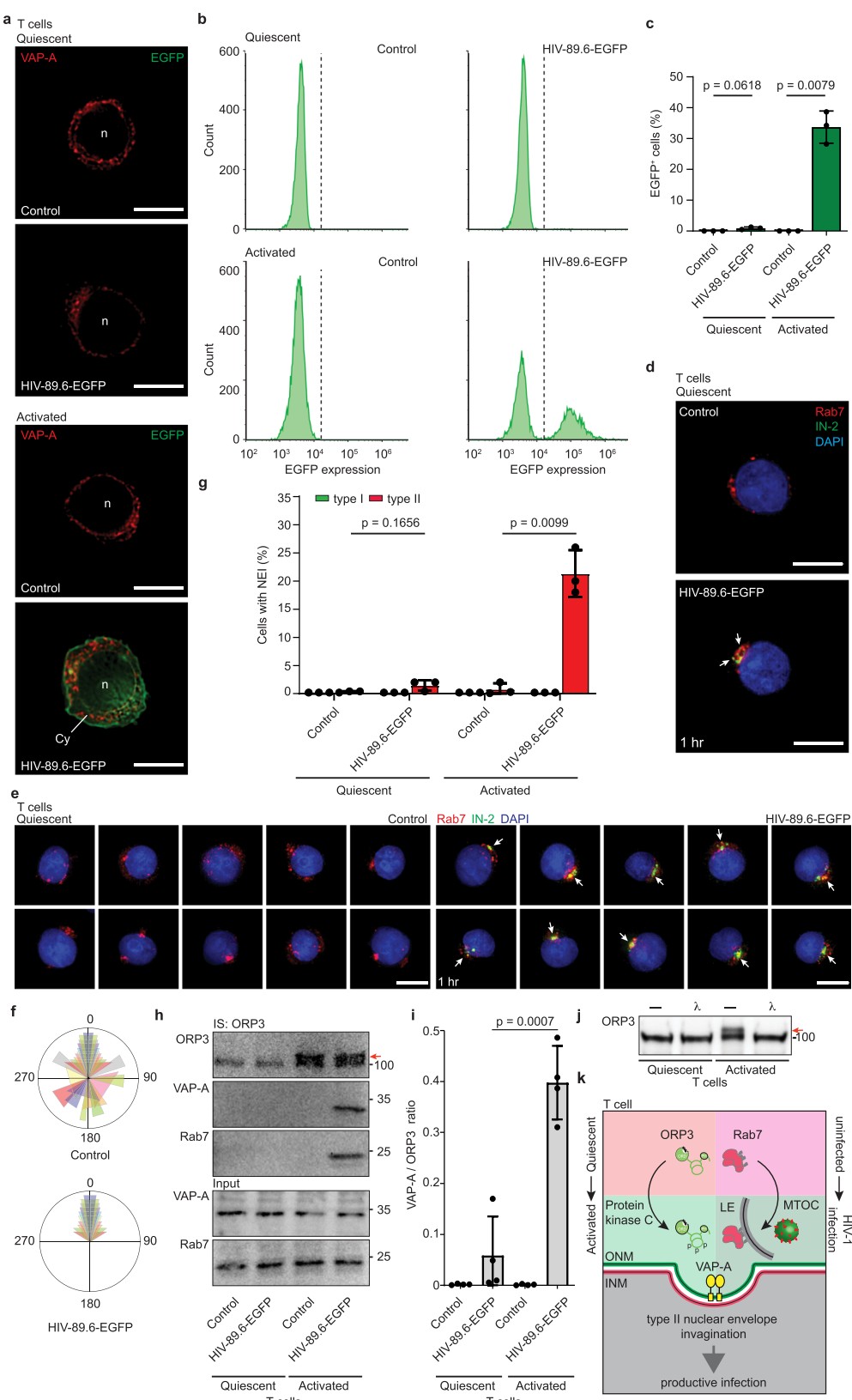

of HIV-1 capsid to host proteins such as cyclophilin A and cleavage and polyadenylation specificity factor 6 in relation to the perinuclar region, particularly NEIs, will be of interest[74,75]. By penetrating deep into the nuclear compartment, often reaching the nucleolus[34,36], NEIs may facilitate the transfer of HIV-1 to specific nuclear subcompartments[76–79]. This is an important consideration because viral complexes can bypass intranuclear movement or diffusion to reach integration sites.

Our data do not contradict the fact that most of the endocytosed viruses end up in the lysosomal compartment, as observed by the colocalization of HIV-1 IN and Lamp1 in the cytoplamic compartment, or that a fraction of them are released from endosomal compartment

**Fig. 8 | ORP3 hyperphosphorylation is important for VOR complex interactions and NEI formation. a–c** Quiescent and PHA/IL-2-activated CD4⁺ T cells were non-infected (control) or 6-h HIV-89.6-EGFP infected, and then 48-h incubated prior to either ICC (**a**) or FC (**b, c**). For ICC, fixed-cells were immunolabeled for VAP-A. Representative images (**a**) and FC histograms (**b**) are displayed. Noninfected activated cells were used for the gating. **b** The percentage of EGFP⁺ cells is plotted (**c** n = 3). **d–g** Cells as indicated were noninfected (control) or HIV-89.6-EGFP infected for 1 (**d–f**) or 3 (**g**) hours prior to immunolabeling for HIV-1 IN (IN-2) and Rab7 (**d–f**) or SUN2 and VAP-A (**g**). Nuclei were counterstained with DAPI. A single section of a representative cell (**d**) or composite images of 25 sections of 10 cells (**e**) are presented. The localization of Rab7 in the perinuclear region is shown in a circular histogram (**f**, 10 cells per condition; color code indicates an individual cell). Note the colocalization of IN-2 and Rab7 at MTOC of infected cells (arrow). The percentage of cells with NEIs of type I or II was quantified (**g** 50 cells per experiment,

n = 3). **h–j** Cells as indicated were noninfected (control) or 6-h HIV-89.6-EGFP infected, solubilized and subjected to ORP3 IS. The bound fractions and input (1/50) were probed for ORP3, VAP-A, and Rab7 (**h**). Note the absence of slower-migrating ORP3 species in quiescent cells (red arrow). The protein ratio of the indicated pair was quantified (**i** n = 4). Detergent cell lysates were treated with λ-phosphatase (λ) or without (−) prior to ORP3 IB (**j**). Molecular mass markers (kDa) are indicated. All samples were from the same experiment and were run in parallel (**h, j**). We used a MOI of 2. Means ± S.D. and individual values of each experiment are shown. P values are indicated. **k** Representation of type II NEI induction that requires both T cell activation leading to ORP3 hyperphosphorylation (left) and viral infection leading to the accumulation of Rab7⁺ late endosomes (LE) at the microtubule-organizing center (MTOC, right). INM/ONM, inner/outer nuclear membrane. Scale bars, 5 μm.

outside the NEIs, where interactions with host cell factors might determine the fate of viral particles and infectivity. Within the NEIs (Fig. 7c, right panel), it is also possible that the extreme curvature of nuclear membrane may facilitate the passage of the intact -60 nm HIV-1 capsid[28], which is much larger than the previously proposed ~40 nm limit for the central channel diameter of the human NPCs[80,81]. These physical constraints could impact the structure and/or composition of NPCs in terms of nucleoporins, which is consistent with the heterogeneity of NPCs[33,82]. A relationship between nuclear membrane organization mediated by its structural components (e.g., SUN1 and SUN2) and HIV-1 infectivity has recently been demonstrated[83]. Whether pre-formed NEIs, present in HeLa cells, but not in CD4⁺ T cells, or only those induced by HIV-1, are involved in the nuclear transfer of HIV-1 components has yet to be explored. In addition to, or in synergy with, the VOR complex, a possible contributory role for HIV-1 Vpr protein in NEI induction cannot be excluded, based on the finding that Vpr induced transient, localized herniations (probably NEI) in the nuclear envelope, associated with defects in the nuclear lamina[84]. Interestingly, a recent study attributed to Vpr in HIV-1 virions a reprogramming role of resting T cells into tissue-resident memory T cells[85]. Despite the occurrence of numerous NPCs around the nuclear periphery, the accumulation of virus-laden late endosomes at a perinuclear pole might explain, at least in part, why only few virions are able to enter the target cell nucleus[86].

Perinuclear localization of HIV-1-containing late endosomes as a crucial microtubule-dependent step that mediates VOR complex formation and induces NEIs is consistent with the role of the microtubule network in the retrograde transport of HIV-1 and late endosomes from the cell periphery toward perinuclear area and MTOC (Fig. 7c, left panel)[53–55,87]. It remains to be determined whether endocytosed viral particles impact the composition and/or maturation of late endosomes as well as their cytoplasmic transport in the NEI core. The role of microtubule motor proteins notably in their transport within NEIs will need additional consideration[54,88]. Interestingly, directed microtubule-dependent intranuclear trajectories have also been observed for herpes simplex virus and adeno-associated virus (AAV)[89,90]. AAV particles were found not only in the perinuclear area but also in the NR 2 h after infection of HeLa cells[91], suggesting that other viruses may utilize the newly described VOR-complex dependent nuclear pathway[92]. Further studies will be required to identify the site(s) of reverse transcription of the viral RNA genome into double-stranded DNA in relation to the VOR complex-mediated nuclear pathway. Because of the rarity of reverse transcription events in an infected cell, the sub-viral reverse transcription complex is currently poorly characterized[28,93]. Our findings are consistent with reverse transcription occurring in intact capsids[94,95], perhaps inside endosomes located outside and/or in NEIs. The possible link between reverse transcriptase activity and HIV-1 uncoating[96], would suggest, however, that this event occurs after the release of HIV capsid in the core of NEIs and near NPCs[30,31].

Our model could also support the evidence that reverse transcription is completed after HIV-1 nuclear import inside the nucleus[32,33,86].

Besides VSV-G and Env proteins, it remains to be determined whether viral internalization also involves nonspecific adhesion or aggregation of HIV-1 particles to the cell surface notably in heterologous cellular system such as HeLa cells, which are lacking CD4[66]. Our data presented in Supplementary Note 1, where a minute fraction (about 2%) of CD4-negative HeLa cells showed a productive infection as detected by double EGFP⁺Gag⁺ cells using RetroNectin-based, but not spinoculation-based, infection methods, suggest this. RetroNectin method relies on a recombinant fibronectin fragment that binds to cells via its interaction with cell surface integrin receptors. Therefore, in the absence of CD4, a tiny fraction of the RetroNectin-virus complex present in the vicinity of the cell membrane can penetrate by other non-specific mechanisms of endocytosis, e.g., macropinocytosis. This technique has already been used to overcome limitations associated with the resistance of CD34⁺ hematopoietic stem/progenitor cells to HIV infection in vitro[47]. These CD34⁺ cells have a low level of CD4 expression. Likewise, it will be interesting to further dissect whether a particular dynasore-sensitive mechanism of internalization/endocytosis such as the clathrin-mediated pathway, or others can determine the subcellular localization of late endosomes in NR[97,98]. It should be noted that our data do not exclude that fusion of HIV-1 with the plasma membrane or early endosomes occurs. Whether HIV-1 capsids released at an early stage into the cytoplasmic compartment followed the same microtubule-dependent pathway as those endocytosed in late endosomes remains to be investigated. Distinct, but complementary, nuclear import pathways can coexist and collectively contribute to productive infection[32]. However, interference with translocation of late endosomes into the NR, or formation of type II NEIs, prevents productive infection as demonstrated in CD4⁺ T cells. Our finding that functional HIV-1 infection requires the VOR complex proteins and presence and/or the induction of type II NEIs elucidates a missing piece of the viral life cycle. Future work will aim to integrate the current findings with other host and viral molecules and processes known to influence viral entry into the nucleus.

The discovery of a fundamental gateway to nucleoplasm induced by viruses could be therapeutically important, as revealed by experiments with drugs. In particular, PRR851 inhibited the productive infection of HIV-1 as effectively as FDA-approved ICZ without cell growth inhibition side-effects. Other advantages of PRR851 over ICZ are that it lacks ICZ moieties associated with anti-fungal and other off-target activities. Moreover, PRR851 is expected to be more potent in patients because seventy percent of ICZ is converted to H-ICZ by the liver, and this metabolite, while retaining antifungal activity[99], does not disrupt the VOR complex and does not inhibit productive infection. From a mechanistic point of view, the drugs prevented the VOR complex-mediated docking and transport of Rab7⁺ late endosomes in the NR by interacting with the conserved C-terminal OSBP-related domain (ORD) of ORP3[38] (Fig. 7c, left panel), whose hydrophobic

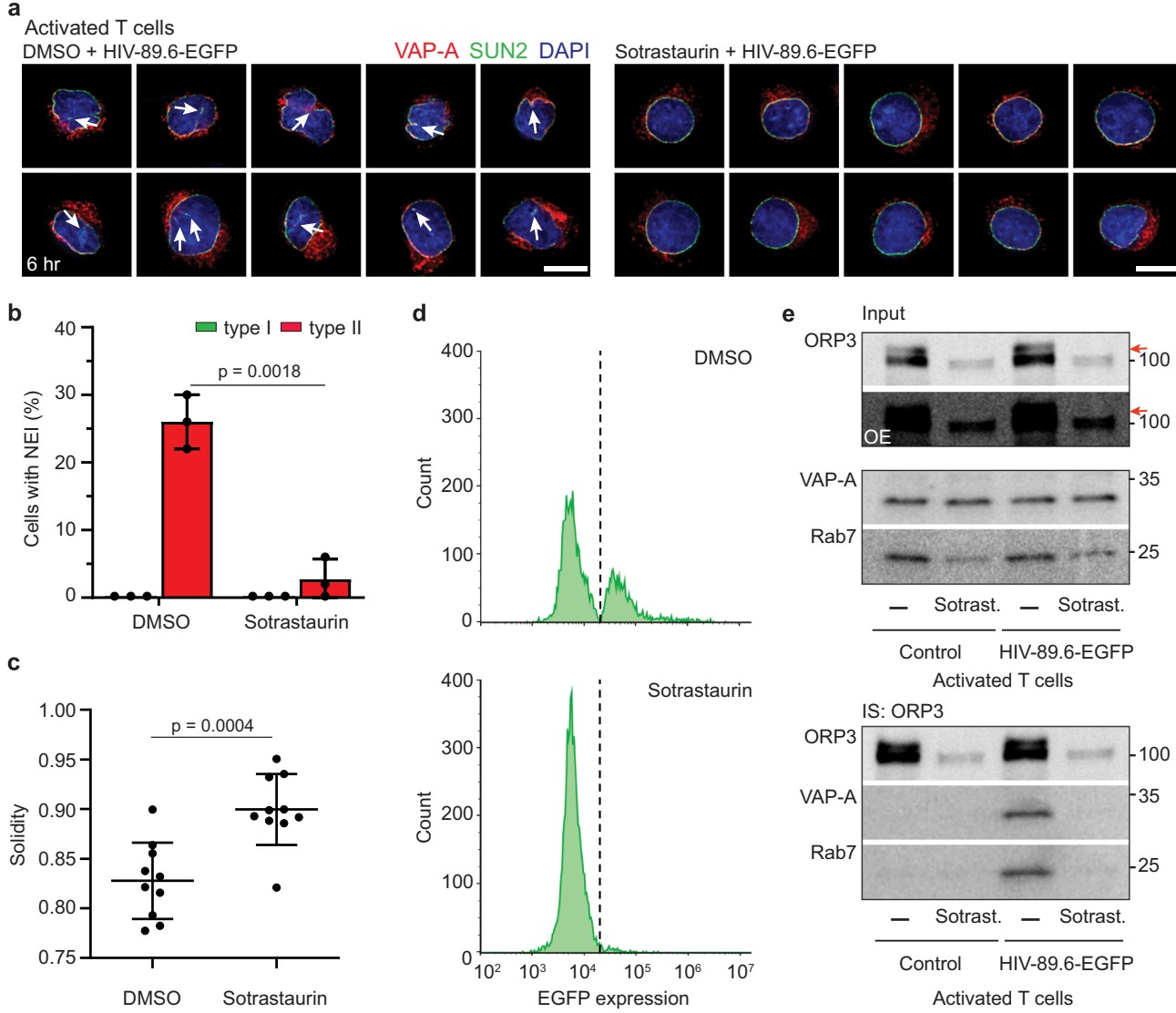

**Fig. 9 | Sotrastaurin-mediated PKC inhibition prevents the formation of both HIV-1-induced NEIs and the VOR complex. a–c** Quiescent CD4⁺ T cells were pretreated with DMSO (DMSO, −) or 40 μM Sotrastaurin (Sotrast.) for 30 min before their activation with PHA and IL-2 for 48 h. Afterward, they were 6-h HIV-89.6-EGFP infected in the presence of Sotrastaurin prior to SUN2 and VAP-A immunolabeling. Nuclei were counterstained with DAPI. Samples were observed by CLSM. Single section of 10 representative cells are presented. Note the presence of NEIs in the infected cell in the absence of Sotrastaurin (left, arrows). The percentage of cells with type I or II NEIs was quantified (**b** 50 cells per experiment, *n* = 3) and the nuclear solidity was determined (**c** *n* = 10 cells). **d** After 6-h infection, cells were incubated for 48 h prior to FC analysis for EGFP expression. Representative FC histograms are displayed. DMSO-treated, noninfected cells were used for the gating. **e** Cells were treated as in **a**, solubilized and subjected to ORP3 IS. As a control, cells were noninfected. The input (1/50) and bound fractions were probed for ORP3, VAP-A, and Rab7 by IB. Note the absence of slower-migrating ORP3 immunoreactive species in Sotrastaurin-treated CD4⁺ T cells regardless of infection (red arrow). OE overexposed blot. All samples were from the same experiment. Molecular mass markers (kDa) are indicated. We used a MOI of 2. In all cases, means ± S.D. and individual values are shown. *P* values are indicated. Scale bars, 5 μm.

pocket binds a single sterol, notably cholesterol and 25-hydroxycholesterol[100,101]. The lack of off-target toxicity of PRR851 (this study; see also ref. 38) warrants further studies for potential clinical therapeutic applications in HIV-1 infection as well as in infection by other viruses that need access to the nucleus for their replication. PRR851 represents the prototype of a novel class of HIV-1 drugs for prevention and treatment of AIDS.

Finally, the parallel nature of retroviruses and exosomes, based on shared physical and chemical characteristics and biogenesis pathways, is the rational for the Trojan exosome hypothesis: that retroviruses exploit cell-encoded pathways of vesicular traffic[42,44,102]. Our data have demonstrated that nuclear transfer of extracellular materials associated with viruses (this study) and EVs[34,37,38] after endocytosis has a shared mechanism, and that both increase the formation of type II NEIs

(this study and ref. 34), adding to the general knowledge about the action of extracellular entities on their target cells. Besides natural viruses and EVs, it has been proposed that exogenous lipid-protamine-DNA nanoparticles exploit tubular invaginations of nuclear envelope as entry-points towards the nucleoplasm[103]. Thus, the inclusion of foreign materials in the type II NEI could be an alternative pathway to nuclear envelope breakdown for their nuclear transfer into non-dividing cells.

## Methods

### Ethics statement

The research carried out in this study complies with ethical regulations approved by the Biosafety Committee of Touro University Nevada. No animals or human patients were involved in this study.

## Chemicals

ICZ (cis-4[4-4-4[[2-(2-4-dichlorophenyl)-2-(1H-1,2,4,triazol-1-methyl)-1,3-dioxolan-4-yl]-1-piperazinyl]phenyl]-2,4-dihydro-2-(1-methyl-propyl)-3H-1,2,4-triazol-3-one; MW: 705.64) and nocodazole were purchased from Sigma-Aldrich (catalog numbers (#) I6657 and #M1404, respectively). H-ICZ and Sotrastaurin were purchased from Cayman Chemical (#22576 and #16726, respectively) and DNS from Santa Cruz Biotechnology (#sc-202592). The chemical drugs PRR851 and PRR846 were synthetized by one of our labs as described[38]. All drugs were diluted in dimethyl sulfoxide (DMSO; VWR International) and used at a final concentration of $10\,\mu M$ for ICZ, H-ICZ, PRR851, and PRR846, $80\,\mu M$ for DNS, or $1\,\mu M$ for nocodazole. λ-phosphatase was purchased from Sigma-Aldrich (#P69614) at a stock solution of 400,000 units (U) per ml.

## Antibodies

The following primary antibodies against human proteins were used for ICC, IS, IB, and/or FC as indicated. (See also Supplementary Table 1 for validation of antibodies). Anti-HIV-1 IN (clone IN-2; ICC (dilution 1:50) and IB (1:200)), anti-ORP3 (clone D-12; ICC (1:50), IS (1:200), and IB (1:500)), anti-actin (clone C-2; IB (1:1000)), and anti-SUN2 (clone A-10; ICC (1:50)) mouse monoclonal antibodies were purchased from Santa Cruz Biotechnology (#sc-69721, #sc-398326, #sc-8432, and #sc-515330, respectively), while the anti-SUN2 rabbit monoclonal antibody (clone ARC2311; ICC (1:100)) was from Thermo Fisher Scientific (#PA5-51539). Anti-ORP3 (IB (1:1000)), anti-VAP-A and VAP-B rabbit polyclonal antibodies (both for ICC (1:100) and IB (1:1000)) were obtained from Bethyl Laboratories (#A304-557A, #A304-366A, and #A302-894A, respectively). Anti-Lamin B1 (clone 119D5-F1; ICC (1:100) and IB (1:1000)), anti-HIV-1 p24 (clone 5; ICC (1:100)), anti-Lamp1 (clone H4A3; ICC (1:100)) mouse monoclonal as well as anti-Rab7 (clone EPR7589; ICC (1:50) and IB (1:500)), anti-CD63 (clone EPR22458-280; ICC (1:100)) and anti-Lamp1 (clone EPR24395-31; ICC (1:100)) rabbit monoclonal antibodies were purchased from Abcam (#ab8982, #ab63958, #ab25630, #ab137029, #ab252919, and #ab278043, respectively). Anti-GAPDH rabbit polyclonal antibody (IB (1:1000)) was obtained from Novus Biologicals (#NB300-326). Phycoerythrin (PE)-conjugated anti-CD3 (clone HIT3a; FC (1:20)), Brilliant Violet 421 (BV421)-conjugated anti-CD4 (clone OKT4, FC (1:20)), and RD1-conjugated anti-HIV-1 Gag (clone KC57, FC (1:20)) mouse monoclonal antibodies were purchased Thermo Fisher Scientific (#12-0039-42), BioLegend (#317434), and Beckman Coulter (#6604667), respectively. The following secondary antibodies were used. Fluorescein (FITC)-conjugated donkey anti-mouse and anti-rabbit IgG (both for IB (1:100)) and tetramethylrhodamine donkey anti-rabbit IgG (ICC (1:50)) were obtained from Jackson ImmunoResearch Laboratories (#715-095-150, #711-095-152 and #711-025-152, respectively). Alexa Fluor™ 647-conjugated goat anti-mouse and anti-rabbit IgG (A-21237 and A-21246, respectively) and Alexa Fluor™ 488-conjugated goat anti-mouse and anti-rabbit IgG (#A-11017 and #A-11070, respectively) were purchased from Thermo Fisher Scientific (all for ICC (1:1000)). DyLight® 550-conjugated goat anti-mouse IgG was obtained from Abcam (#ab98758; ICC (1:500)).

## Cell culture

The human cervix epithelioid cell line HeLa (ATCC®CCL-2™) and embryonic kidney cell line 293 T (CRL-3216) were obtained from the American Type Culture Collection (ATCC), while CD4+ HeLa cells (ARP-1109, clone 1022) were acquired from NIH HIV Reagent Program, Division of AIDS, NIAID, NIH[104]. They were cultured in DMEM (#11995065, Thermo Fisher Scientific). Human helper T (CD4+) cells were obtained from Discovery Life Sciences (#C4T0015-Z1110032763081618BA). We used five distinct lots of CD4+ T cells. As stated by the vendor, these primary cells were obtained from a Caucasian male (patient ID:110032763). Alternatively, CD4+ T cells were isolated from human peripheral blood mononuclear cells of a healthy volunteer using the MojoSort™ human CD4+ T cell isolation kit from BioLegend (#480010). FEMX-I cells were originally derived from a lymph node metastasis of a patient with malignant melanoma and found to be highly metastatic in immunodeficient mice[105,106]. They are wild type for BRAF, PTEN, and NRAS[34]. They were authenticated by morphology and proteomics[37,107] (See also Supplementary Table 2 for validation of cells). Both CD4+ T cells and FEMX-I cells were kept in RPMI-1640 medium (#10-041-CV, Corning Inc.). All cell culture media were supplemented with 10% fetal bovine serum (FBS, #26140079), 2 mM L-glutamine (#25030081), 100 U/mL penicillin and 100 μg/mL streptomycin (#15140122), all from Thermo Fisher Scientific. For CD4+ T cells, culture media was additionally supplemented with 10 μg/ml PHA (#11249738001, Sigma-Aldrich) and 0.01 μg/ml recombinant human IL-2 (#200-02, PeproTech) to activate them for 2 days prior to starting experiments[108]. For experiments with quiescent, non-activated T cells, PHA and IL-2 treatment was omitted. All cells were incubated at 37 °C in a 5% $CO_2$ humidified incubator. They were regularly verified for absence of mycoplasma contamination by either staining with 4',6-diamidino-2-phenylindole (DAPI; #D9542, Sigma-Aldrich) and visualization under a fluorescent microscope or polymerase chain reaction using the MycoSEQ™ Mycoplasma Detection Kit (#4460626, Thermo Fisher Scientific), according to the manufacturer's protocol.

## Plasmids

The HIV NL4-3 Gag-iGFP ΔEnv (ARP-12455), p89.6 ΔE ΔN-SF-EGFP (ARP-12487) and pcDNA 89.6 env (ARP-12485) plasmids were obtained from the NIH HIV Reagent Program. NL4-3 Gag-iGFP ΔEnv carries the full-length HIV-1 clone with a frame shift mutation in the *env* open reading frame[56]. In the latter construct, GFP is introduced between the matrix (MA) and capsid (CA) domains of the Gag protein, as denoted by i (interdomain) in iGFP. The p89.6 ΔE ΔN-SF-EGFP expresses the molecular clone 89.6 bearing both a 707 base pairs deletion in Env and a Nef deletion. Enhanced (E) GFP is expressed off the internal spleen focus forming virus (SFFV) promoter[59]. When transfected into producer cells together with the pcDNA 89.6 env expression plasmid, which constitutively expresses 89.6 env from a CMV promotor, it generates an infectious virus for single cycle infection. The expression plasmid PSF-CMV-VSVG (#OGS592, Sigma-Aldrich) was utilized for VSV-G-pseudotyping of HIV-1 particles.

The C-terminal GFP-tagged fusion protein versions of human VAP-A (GenBank Accession Number NM_194434.2; #HG11412-ACG; Sino Biological, Beijing, China) or CD9 (GenBank Accession Number NM_001769; #RG202000, OriGene Technologies, Rockville, MD) carried in the vectors pCMV3-C-GFPSpark (VAP-A-GFP plasmid) and pCMV6-AC-GFP (CD9-GFP plasmid), which contain the hygromycin and neomycin resistance genes, respectively, were used. For the specific gene knockdown, short hairpin RNA (shRNA) plasmids targeting VAP-A (#HSH022333-nH1; Accession No. NM_003574.5), VAP-B (#HSH022331-nH1; Accession No. NM_004738.3), and ORP3 (#HSH006982-nH1; Accession No. NM_015550.2) were purchased from GeneCopoeia (Rockville, MD). A scrambled shRNA plasmid (#CSHCTR001-nH1; GeneCopoeia) was used as a control. All of the shRNA plasmids contained the puromycin resistance gene. For each protein of interest, a pool of target sequences was utilized after their individual evaluation[37]. Two distinct shRNA plasmids were selected for VAP-A (5′- CCACAC AGTGTTTCACTTAAT-3′ and 5′- GCACATTGAGTCCTTTATGAA-3′) and VAP-B (5′- GGATGACACCGAAGTTAAGAA-3′ and 5′- GGTAAATTGGA TTGGTGGATC-3′), whereas four shRNA plasmids were selected for ORP3 (5′- CCATGTTTCCACATGAAGTTA-3′, 5′-CCTCCAATCCTAATTT GTCAA-3′, 5′- GCCCATAAAGTTTACTTCACT-3′, and 5′- GGAGAAACAT ATGAATGTATT-3′).

## Transfection

The transient co-transfections of 293 T cells ($4 \times 10^6$) growing on 100-mm culture dishes were performed with HIV NL4-3 Gag-iGFP ΔEnv and

PSF-CMV-VSVG plasmids or p89.6 ΔE ΔN-SF-GFP and pcDNA 89.6 env plasmids using lipofectamine 3000 (#L3000008, Thermo Fisher Scientific) in a 1 part DNA: 2 parts lipid ratio. Cells were incubated for 24 h, medium replaced, then incubated further for 48 h at 37 °C under 5% $CO_2$ prior to the virus collection (see below). In some experiments, only HIV NL4-3 Gag-iGFP ΔEnv or p89.6 ΔE ΔN-SF-GFP plasmid was transfected, thus producing a "bald" virus to be used as control.

HeLa cells were stably transfected with VAP-A-GFP plasmid (500 ng) using lipofectamine 3000. Transfected cells were then selected by introducing 200 µg/ml of hygromycin (#H3274, Sigma-Aldrich) into the culture medium for 7 days. FEMX-I cells stably transfected with CD9-GFP plasmid were previously described[20]. To inhibit VAP-A, VAP-B, or ORP3 expression, HeLa cells were transfected with pooled shRNA plasmids (500 ng). After transfection, stable cell lines were selected by 1 µg/ml puromycin (#P9620, Sigma-Aldrich) for 7 days. All selection antibiotics were removed from the medium at least 1 week before experiments.

## Baculovirus-based expression
The baculovirus-based BacMam 2.0 CellLight® Early Endosomes-red fluorescent protein (RFP) and Late Endosomes-RFP (#C10587 and #C10589, respectively, Thermo Fisher Scientific) were used to induce the expression of Rab5a-RFP and Rab7-RFP, which highlight early and late endosomes, respectively. CellLight™ ER-GFP (#C10590), which consists of an ER-GFP fusion protein containing the ER signal sequence of calreticulin with the KDEL (ER retention signal) motif and GFP, was used to label the ER. Viral particles were added at a concentration of 30 per cell for 24 h, as recommended by the manufacturer. Afterward, Rab-RFP⁺ cells were infected with HIV-Gag-iGFP viruses.

## HIV-1 production and titration
After 72 h post-transfection, viral particles were harvested by collecting the conditioned culture medium of 293 T cells and spinning at $500 \times g$ for 5 min at 4 °C. The supernatant was passed through sterile 0.45-µm filter, and then concentrated by centrifugation at $3000 \times g$ for 30 min at 4 °C using a Macrosep® Advance Centrifugal Device 100 K (#89131-992, VWR International). The resulting viral supernatant was then aliquoted and stored at −80 °C. To determine the titer, HeLa or CD4⁺ HeLa cells ($1 \times 10^5$) were seeded on 24-well plate and upon adherence, were infected with various dilutions of viral stocks of HIV-Gag-iGFP or HIV-89.6-EGFP, and further incubated for 48 h. Cells were trypsinized and analyzed by FC (see below). Wells with less than 40% fluorescent-positive cells were considered for calculations. Titer was measured as transduction units (TU)/ml = number of infected cells × % GFP-positive cells / volume of viral stock used in ml. The calculated titer was $2.6 \times 10^5$ and $3.8 \times 10^5$ TU/ml for HIV-Gag-iGFP and HIV-89.6-EGFP, respectively. For the "bald" HIV-Gag-iGFP or HIV-89.6-EGFP viruses, titers were zero, and the number of viral particles used in the control experiment was equalized to the VSV-G-pseudotyped or native counterpart based on the number of fluorescent GFP-positive particles counted by high-resolution microscopy (Supplementary Fig. 1a) or by employing equal volumes of supernatant as in the corresponding enveloped viral particles.

## HIV-1 infection and drug treatment
To infect HeLa cells, including those transfected with a scrambled shRNA plasmid or silenced for VAP-A, VAP-B, or ORP3, CD4⁺ HeLa cells or native CD4⁺ T cells, either HIV-Gag-iGFP or HIV-89.6-EGFP and corresponding "bald" viruses were preloaded onto RetroNectin®-coated plates. Herein, non-tissue culture treated 24-well plates (#15705-060, VWR International) or uncoated 35-mm confocal dishes (#P35G-1.5-14-C, MatTek Corp., USA) were coated with 50 µg/ml RetroNectin® recombinant human fibronectin fragment (#T100B, Takara Bio USA) for 2 h at room temperature (RT). A blocking solution (PBS containing 2% bovine serum albumin (BSA, #97061-420, VWR International)) was added for 30 min at RT to block nonspecific binding, then washed with PBS. HIV-Gag-iGFP or HIV-89.6-EGFP (both with MOI of 2) or bald viruses were then added onto the plates or dishes and centrifuged at $960 \times g$ for 30 min at 4 °C. In some experiments as indicated in the figure legend, lower or higher MOI were used. The supernatant was removed, and plates or dishes washed with PBS before addition of cells, pretreated or not with drugs. In the latter case, cells were seeded on a separate plate or dishes were treated with 80 µM DNS, 10 µM ICZ, H-ICZ, PRR851 or PRR846 for 30 min, trypsinized (in the case of HeLa and CD4⁺ HeLa cells) and added to the viral preparation. Cells were infected for 1, 3 or 5 h in the presence of the drugs prior ICC analysis. For the productive infection, cells were infected for 6 h, medium replaced, and then incubated further for 24, 48, 72, or 96 h in the presence or absence of drugs. DMSO alone was used as solvent vehicle control.

In some experiments involving quiescent or PHA/IL-2-activated CD4⁺ T cells, after 1, 3, or 6 h of infection, cells were fixed and processed for ICC (see below). Alternatively, they were pretreated with 1 µM nocodazole for 5 min, then loaded onto HIV-89.6-EGFP virus-containing, RetroNectin®-coated plates and infected for 1 h, followed by ORP3 immunoisolation (see below). For Sotrastaurin treatment, a 40 µM concentration[64] was used for 30 min before their activation with PHA and IL-2 for 48 h. Sotrastaurin was maintained during infection with HIV-89.6-EGFP.

We also used a spinoculation infection technique[59] to confirm the data obtained with RetroNectin-based method. Briefly, cells were seeded on 24-well plates, HIV-89.6-EGFP was added (MOI of 2), and the plates were centrifuged at $2095 \times g$ for 30 min at 4 °C. The cells and virus were then incubated at 37 °C for 6 h, after which the cell medium was replaced before a further 48-h incubation. In some experiments, cells were treated with DMSO or PRR851 (10 µM) before addition of the virus. The drug was maintained during all incubations.

## Isolation of extracellular vesicles
EVs were prepared from parental or CD9-GFP-transfected FEMX-I cells cultured in serum-free medium supplemented with 2% B-27 supplement (#17504044, Thermo Fisher Scientific) on 6-well plates pre-coated with 20 µg/ml poly(2-hydroxyethyl methacrylate) (#P3932, Sigma-Aldrich) to prevent their attachment as described[34]. After 72 h, EVs were enriched by differential centrifugation according to the latest guidelines of the International Society for Extracellular Vesicles[109]. Briefly, after low-speed centrifugations (300 and $1200 \times g$) of conditioned medium, the supernatant was centrifuged at $10,000 \times g$ for 30 min at 4 °C. The resulting supernatant was centrifuged at $200,000 \times g$ for 60 min at 4 °C. The pellet was resuspended in 200 µl PBS. The average size of $148 \pm 1.83$ and $152 \pm 4.81$ nm (means ± standard deviations (S.D.)) and concentration of $1.32 \times 10^{10}$ and $4.33 \times 10^9$ particles per ml for parental and CD9-GFP-positive FEMX-I EVs, respectively, were determined by nanoparticle tracking analysis using ZetaView (Particle Metrix GmbH, Meerbusch, Germany) according to the manufacturer's protocol. The EVs were previously characterized by IB, three-dimensional (3D) direct stochastic optical reconstruction microscopy, and electron microscopy[38,110], and acquired data were deposited in the EV-TRACK knowledgebase (EV-TRACK, https://evtrack.org/; ID: EV210180, author: Santos)[111].

## EV−cell incubation
HeLa cells were seeded at a density of $1 \times 10^5$ cells/ml on 35-mm poly-D-lysine-coated glass-bottom dishes (#P35GC-1.5-14-C, MatTek Corp.). Upon adherence, they were incubated with $1 \times 10^9$ EVs per ml derived from FEMX-I cells for 3 h at 37 °C. Afterward, cells were processed for ICC and visualized by CLSM (see below). Note that when viruses (HIV-Gag-iGFP or corresponding "bald") were used to infect HeLa cells, approximately $2.2 \times 10^8$ particles/ml corresponding to EVs were present in the virus preparation, as deduced from the conditioned culture

medium of non-transfected 293 T cells assessed from the nanoparticle tracking analysis.

## Immunocytochemistry

Adherent cells grown on RetroNectin®-coated or uncoated glass-bottom dishes (MatTek Corp.) or RetroNectin®-coated plates were processed for ICC after drug treatment and/or HIV-Gag-iGFP or HIV-89.6-EGFP infection. Cells were washed with PBS, fixed in pre-chilled 100% methanol at −20 °C for 10 min, washed twice with PBS, permeabilized with 0.2% Tween 20 in PBS (permeabilization buffer) for 15 min, and blocked with 1% BSA in PBS, for 1 h at RT. They were then incubated for 60 min at RT with primary antibody against specific proteins as indicated in each figure (see Supplementary Table 1), washed twice with PBS, incubated with appropriate fluorescent secondary antibody for 30 min, and washed twice prior to observation. All antibodies were diluted in permeabilization buffer containing 1% BSA. Nuclei were counterstained with DAPI. ONM and INM were detected using VAP-A and SUN-2 antibodies, respectively. Cells were observed in PBS by CLSM using the Nanoimager high-resolution microscope (ONI, Oxford, UK) with 100X oil-immersion objective. For CD4$^+$ T cells, they were fixed, permeabilized, and immunolabeled as above but in suspension, washing twice with PBS by centrifugation at $300 \times g$ for 10 min in-between each incubation. After final wash, they were resuspended in 2X PBS and allowed to sediment on a glass coverslip for 30 min prior to imaging. All images were acquired under the same microscope settings for subsequent calculations of mean fluorescence and recorded using Nanoimaging (ONI). Fluorescence signal was quantified using Fiji software (http://fiji.sc/wiki/index.php/Fiji). To evaluate the ratio of nucleoplasmic and cytoplasmic-associated IN-2 signal, regions of interest (ROI) were drawn around the nucleus and cytoplasm, with the latter excluding the nuclear compartment. The "measure" function on Fiji was then applied to determine the fluorescent signal of 21–32 individual 0.3-µm step x-y optical sections. The total cell fluorescence was calculated as, integrated density – (area × background mean fluorescence). In some settings, viral particles and CD9-GFP-positive EVs were adhered to #1.5 glass coverslip and observed by CLSM. The signal intensity of Gag-iGFP within the viral particles was artificially made visible by adjusting the image gamma using the process>math>gamma function in Fiji. For 3D image reconstruction, z stacks of entire cells at 0.3-µm per slice were acquired, and the images were rendered in 3D using Imaris software (version 9.6, Bitplane, Concord, MA).

## Flow cytometry

HeLa, CD4$^+$ HeLa, or quiescent and PHA/IL-2-activated CD4$^+$ T cells seeded on 24-well plates were processed for FC after drug treatments and/or HIV-Gag-iGFP or HIV-89.6-EGFP infection. Cells were trypsinized (for HeLa cells only) using Trypsin/EDTA solution (#25-052-CI, Corning Inc.) for 3 min, blocked by the addition of respective cell medium (see above), centrifuged at $200 \times g$ for 5 min, and resuspended in PBS. They were then immediately analyzed for (E)GFP expression using the CytoFlex flow cytometer system (Beckman Coulter). Alternatively, CD4$^-$ and CD4$^+$ cells were first immunolabeled with fluorochrome-conjugated anti-CD4 and/or CD3 antibodies diluted in PBS containing 5% FBS (FACS buffer) for 45 min on ice, washed twice by centrifugation at $300 \times g$ for 10 min, and resuspended in FACS buffer prior to analysis. In other experiments, cells were fixed in pre-chilled 100% methanol at −20 °C for 10 min, washed twice with FACS buffer by centrifugation at $300 \times g$ for 10 min, permeabilized with 0.2% Tween 20 in FACS buffer for 15 min, and blocked with 1% BSA in FACS buffer for 1 h on ice. They were then incubated for 60 min on ice with anti-HIV-1 Gag antibody diluted in FACS buffer and washed twice with FACS buffer prior to analysis. Detector gain for all channels was kept constant for all experiments. Scrambled shRNA transfected- or DMSO-treated cells without viral infection or antibody staining were used to

gate the negative cell population (Supplementary Fig. 16). Cells were further gated based on CD4 and/or CD3 positivity. At least 100,000 events were acquired. Data were analyzed using FlowJo software (version 10.7.2, BD Life Sciences).

## Subcellular fractionation

Drug-treated HeLa cells were infected with HIV-Gag-iGFP for 3 h, and then subjected to subcellular fractionation using the NE-PER kit (#78833, Thermo Fisher Scientific). Briefly, cells were trypsinized, washed with PBS, then resuspended in cytoplasmic extraction reagent (CER) I solution and incubated on ice for 10 min. Cold CER II solution was added and further incubated for 1 min. Resulting lysate was then centrifuged at $16,000 \times g$ for 5 min at 4 °C. The supernatant containing the cytoplasmic fraction was then placed in cold, fresh tube, and the remaining pellet (nuclei) was washed twice with PBS by centrifugation. Nuclear extraction reagent (NER) I was then added to further lyse the nuclei and incubated for 40 min on ice. The centrifugation step was repeated, and the supernatant containing the nuclear fraction was placed in cold, fresh tube. Cytoplasmic and nuclear extracts were separated and immunoblotted as described below.

## Immunoisolation and immunoblotting

HeLa cells or those deficient in VAP-A, VAP-B, or ORP3, and CD4$^+$ T cells were solubilized in pre-chilled lysis buffer (1% Triton X-100, 150 mM NaCl, 50 mM Tris-HCl, pH 8.0, supplemented with Set III protease inhibitor cocktail (#539134, Calbiochem, Merck)) on ice for 30 min, followed by centrifugation ($12,000 \times g$) for 10 min at 4 °C. For the IS, we used 0.5% Triton X-100 instead of 1% in lysis buffer. For the dephosphorylation experiments, λ-phosphatase (2000 U/ml) was added to detergent extracts of CD4$^+$ T cells and the reaction was supplemented with 2 mM MnCl$_2$. The lysates were incubated for 3 h at 30 °C. As control, no enzyme was added. The ORP3-based IS protocol was described previously[38]. Briefly, the anti-ORP3 antibody (clone D-12) was added to the detergent lysates followed by Protein G-conjugated magnetic beads according to the manufacturer's protocols (#130-071-101, Miltenyi Biotec). Samples were applied to µ Columns (#130-042-701, Miltenyi Biotec). Materials retained in columns were washed (4x) with 1 ml lysis buffer and rinsed once with 20 mM Tris-HCL pH 7.5. Pre-heated (95 °C) sodium dodecyl sulfate (SDS) buffer (1% SDS, 50 mM DTT, 1 mM EDTA, 10% glycerol, 0.005% bromophenol blue, 50 mM Tris-HCL, pH 6.8) was applied to the column to elute the bound fraction.

Protein samples (10–15 µg) were separated by SDS-polyacrylamide gel electrophoresis using a 4–20% Tris-glycine pre-cast gel (#4561095, Bio-Rad) along with the Trident pre-stained protein molecular weight ladder (#GTX50875, GeneTex), and transferred to a nitrocellulose membrane (#88018, Thermo Fisher Scientific) overnight at 4 °C. Membranes were incubated in a blocking buffer (PBS containing 1% BSA) for 60 min at RT and then probed with a given primary antibody for 90 min at RT. After three washing steps of 10 min each with PBS containing 0.1% Tween 20 (washing buffer; VWR International), the membranes were incubated with appropriate FITC-coupled secondary antibody for 30 min at RT. All antibodies were diluted in the blocking buffer. The membranes were rinsed three times with washing buffer, once with distilled and deionized H$_2$O and the antigen-antibody complexes were visualized using the iBright FL1000 imaging system (Thermo Fisher Scientific).

## Half maximal inhibitory concentration (IC$_{50}$)

IC$_{50}$ values of ICZ and PRR851 for the inhibitory binding of Rab7 to ORP3 were determined by exposing detergent HeLa cell lysates to an increasing concentration (1–10 µM) of compounds for 30 min on ice followed by IS of ORP3. The bound fractions were probed by IB for ORP3 and Rab7. The ratio of Rab7/ORP3 for each condition and compared with control, i.e., DMSO solvent, was used to generate a dose-

response data of the percentage of inhibition. A simple linear regression with an equation $y = 10.08 \times x + 1.573$ was then used to determine $IC_{50}$ using GraphPad Prism 8 software.

## Growth inhibition assay

HeLa cells were seeded at a density of $4 \times 10^3$ cells/well on 96-well plates and incubated overnight to allow their adhesion, while PHA/IL-2-activated CD4$^+$ T cells were grown in suspension at $1 \times 10^4$ cells/well. Both cell types were then treated with 2, 5, 10, and 30 μM ICZ, H-ICZ, PRR851, or PRR846 for 48 h. DMSO was used as control. At the appropriate time, a CellTiter96 AQueous One Solution Cell Proliferation Assay (1:5 dilution; #G3580, Promega) was added for 2 h at 37 °C. The reagent utilizes the biochemical reaction of a tetrazolium compound [3-(4,5-dimethylthiazol-2-yl)-5-(3-carboxymethoxyphenyl)-2-(4-sulfophenyl)-2H-tetrazolium, MTS] to produce a colored, soluble formazan product that is proportional to the number of live cells. The absorbance value was measured at 490 nm using the Varioskan Flash plate reader (Thermo Fisher Scientific).

## Data, statistics and reproducibility

All the experiments were carried out with at least three independent replicates. The micrographs shown are representative of the data obtained. Data are presented as the means ± S.D. Relationships between MOI or PRR851 and induced NEI versus EGFP expression were estimated in repeated measures linear models, and between NEI and EGFP using Pearson's R correlations. Colocalization of IN-2 or p24 with Rab proteins was measured by drawing a ROI around the cell cytoplasm and calculating Pearson's R coefficient using the coloc function in Fiji. At least 10 cells were evaluated per experiment and 21–32 individual 0.3-μm step optical sections per cell were used. All bar graphs and scatter dot plots were created using GraphPad Prism 8. The solidity was evaluated as described[112] using the ratio of the nucleus' measured area to the area of its convex hull shape (see Fig. 6g). Statistical significance was determined using a two-tailed Student's $t$-test (Figs. 1c, f, h, 3b, c, g, 4e, f, k, 5b, d, g, 6b, d, h, j, 8c, g, I, 9b, c, Supplementary Figs. 3b, 5a, b, 9c, 10b, c, e, g, 14b) or two-way ANOVA with Tukey's multiple comparison test (Supplementary Figs. 11c, h, 12a–c, 14c). All $p$ values are indicated.

## Reporting summary

Further information on research design is available in the Nature Portfolio Reporting Summary linked to this article.

## Data availability

All additional data are provided in the Supplementary Information. Data used to generate the graphs as well as all uncropped blots are provided in the Source Data file. Source data are provided with this paper.

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

## Acknowledgements

The authors thank F. Kashanchi, C. Okeoma, and Y. Lazebnik for their suggestions. The Human Immunodeficiency Virus Type 1 (HIV-1) NL4-3 Gag-iGFP ΔEnv Non-Infectious Molecular Clone (ARP-12455) contributed by Dr. Benjamin Chen, HIV-1 89.6 Env Expression Vector (pcDNA 89.6 env; ARP-12485); p89.6 ΔE ΔN-SF-EGFP (ARP-12487), contributed by Dr. K. Collins and Dr. R. Collman and CD4$^+$ HeLa cells (Clone 1022; ARP-1109), contributed by Dr. B. Chesebro, were obtained through the NIH HIV Reagent Program, Division of AIDS, NIAID, NIH. NIH is supporting A.L. and G.R. (grant number 1R15CA252990).

## Author contributions

M.F.S., D.Carbone, D.W., and F.A. conducted the experiments. M.F.S., G.R., J.K., D.M., C.V., D.C., and A.L. analyzed the data. P.D., G.C., and A.L. provided biological and chemical resources. M.F.S. and D.C. made the figures. D.C. and A.L. oversaw the design of the entire study and supervised the experiments. G.R. and A.L. acquired the funding. C.V. and A.L. administrated the project. D.C. and A.L. wrote the manuscript with input from all authors.

## Funding

## Competing interests

The United Kingdom patent application GB2598624A (applicants: M.F.S., G.R., P.D., G.C., A.L., and Technische Universität Dresden; inventors: M.F.S., G.R., P.D., G.C., D.C., and A.L.), European patent application EP3864409A1 (applicants: M.F.S., G.R., A.L. and Technische Universität Dresden; inventors; M.F.S., G.R., D.C. and A.L.) and United States provisional patent number US20210353616A1 (applicants: M.F.S., G.R., A.L. and Technische Universität Dresden; inventors: M.F.S., G.R., D.C., and A.L.) are pending. The patent EP3864409A1/US20210353616A1 is entitled: Inhibition of a tripartite VOR protein complex in multicellular organisms. The patent GB2598624A is entitled: Use of triazole analogues for inhibition of a tripartite VOR protein complex in multicellular organism. These patents are related to the use of itraconazole and triazole analogues to inhibit the VOR protein complex, which prevents the nuclear transfer of materials transported by extracellular particles (e.g., extracellular vesicles and viruses). The authors declare no other competing interests.
