## [Peer Review file · Nature Communications]

REVIEWER COMMENTS

Reviewer #1 (Remarks to the Author):

The manuscript by Santos and colleagues reports their results obtained while investigating the entry of HIV into the nucleus of an infected cell. They demonstrate that nuclear entry of endocytosed HIV is associated with invaginations of the nuclear envelope (NEI), which involve a complex of VAP23A, hyperphosphorylated ORP3, and Rab7. They also show that drugs targeting the VAP23A-ORP3-Rab7 complex inhibit HIV nuclear entry and infection. This work follows on a series of studies by Melikyan's group, who suggested that HIV fuses with endosomes, rather than plasma membrane. This notion was confirmed in later studies, although not for all HIV-susceptible cell types. It appears to hold for CD4+ T cells, the major target of HIV, so results of the current study are highly relevant and important. Nevertheless, a number of key questions remain obscure, so the reader is left without a clear understanding of the process. The issues listed below need to be addressed experimentally, or at least discussed based on available evidence.

1. The authors show that NEI are induced by late endosomes carrying HIV. Late endosomes are commonly associated with the lysosomal degradation pathway. It is important to determine why late endosomes with HIV inside do not acidify, or if only a small proportion does not acidify what determines this protection.
2. The time course is important to determine, as this has a big impact on the viral infection. Does virus-endosome fusion occur when endosomes dock to the MTOC? How is this timing regulated?
3. How do NEI help virus to pass through the nuclear pore? Are they just a by-product or a functional structure? ORP3 findings do not address this question, as the complex may be needed for nuclear pore function as well as for NEI generation.
4. Findings with the drugs targeting the VAP23A-ORP3-Rab7 complex are very important and require standard time-course analysis of the drugs' effect on viral replication, preferably with primary HIV isolates and CD4+ T cells.
5. Animal experiments are out of place here, as no HIV infection has been done in mice. Demonstration that mice do not lose weight is meaningless without knowing the effective dose, measuring PK and PD. Instead, the authors should do a more careful analysis of drug toxicity in primary T cells.

6. Fig. 1: just one cell is shown at high resolution. Please show several cells at lower resolution.

7. There is no staining for endosomes, how do the authors know that endosomes are involved?

8. Fig. 5: there is no evidence that ORP3 hyperphosphorylation is essential. The only conclusion that can be made is that it associates with complex formation nuclear entry. To show necessity, mutagenesis experiments should be involved.

Reviewer #2 (Remarks to the Author):

Santos et al propose a very intriguing mechanism by which HIV-1 viruses in endosomal vesicles invaginate the nuclear envelope and enter the nuclear compartment. Although this is a very interesting hypothesis, the data presented here is not convincing and lack important experiments and controls. Imaging helps, but the authors need biochemical evidence, where all viruses are accounted, i.e., cellular fractionation into nuclear and cytosolic content for the same type of experiments. Isolation of these EVs by gradient-centrifugation etc. The authors showed that depletion of VAP-A and the use of ICS blocks HIV-1; however, the stage of the block is not studied (this should correlate with their model). MOIs are not provided. Overall, the data presented here does not support the conclusions.

Major issues:

Fig1A: bald virus should show IN in vesicles in the inside the cytosol since bald particles do not have VSV-G; however, they have all the other components (i.e. IN), and they are endocytosed. The bald infected control used by the author is not correct (after an hour, the authors should see vesicles and virus in the surface). Lack MOI information.

Fig1A and B: the authors need to show how many of these events they see (nuclear envelope invaginations), is this rare? What is the MOI. Statistics are important here. Showing images of single cells is not convincing.

Fig1H: DNS should also accumulate particles in the surface, or near the surface, of the cell. Again showing one picture does not provide convincing evidence. One of the very well-known problems of

imaging HIV-1 is that bald particles can be seen in the cytoplasm(in endocytic vesicles); Therefore the use of dynosore(DNS) should allow the investigator to see particles in the surface/near-the-surface of the cell.

Fig2A: the authors need a proof of knockdown expression for VAP-A, VAP-B and ORP3. This Figure needs an MOI.

Fig2F: needs a curve of infection using different amounts of virus when depleting VAP-A, not only one point. In addition, the investigators need to figure out where the block is when depleting VAP-A(reverse transcription, nuclear import, integration etc..)

Fig3: the assay that quantifies SUN2+ NEI containing IN-2 should be validated biochemically by fractionating cells in to nuclear and cytosolic content, or isolation of these EVs by gradient-centrifugation etc. This assay is used all over the manuscript and is not clear to this reviewer how reliable it is since the bald viruses show no virus in the cell, which is unlikely.

Fig 3: the authors show that ICZ blocks viral infection, they should provide the stage of the block(reverse transcription, nuclear import, integration etc..)

Fig4B: what is the stage at which HIV-1 is inhibited by these drugs(reverse transcription, nuclear import, uncoating, integration etc..)

The paper is written with very little information in the text and figure legends. For a reviewer to assess the data, the authors need to provide important details, MOI, time of infection, how the quantification of IN+ is performed, etc. The paper is hard to read, please describe the experiments in the figure legends better.

Line 53-59: references suggesting that nuclear import of the HIV-1 core precedes uncoating are missing.

Reviewer #3 (Remarks to the Author):

Santos and colleagues have studied the role of the VOR complex comprising VAP-A, ORP3 and Rab7 in HIV infection. They provide data suggesting that it is an essential cofactor for efficient HIV infection and possibly a determinant of T cell infection that may contribute to explaining why activation of T cells is required to make them permissive, at least in vitro. It's an intriguing study and I enjoyed reading it.

I can see that these authors have not worked previously in HIV and as such may not be familiar with the enormous body of literature relating to HIV traffic in cells via endosomes and the process of capsid nuclear entry. They also haven't discussed modern models of what dictates resting T-cell infection and I'm afraid that without incorporating, at least discussing, some of the key dogmas of HIV biology, this study is going to get a tough time in review and not be widely appreciated by the field.

I like it because many of the experiments are compelling and are clearly telling us something important that we didn't know about cofactor requirements for HIV infection. But at the moment the study is written without taking into account some of the key HIV literature and in places the authors do not support the conclusions with data.

I'd like to suggest re-consideration of some key points.

1. The HIV particles that actually infect cells are not thought to be in endosomes when they arrive at the nucleus. They are thought to be naked in the cytoplasm and able to interact with cytoplasmic and nuclear pore associated cofactors as a prerequisite to nuclear transport. As well as all the studies describing capsid binding cofactor dependence, there are the microscope studies eg (Zila et al Cell 2021), and also from Vinay Pathak, that confirm that HIV is not in endosomes at this point. Its important to understand that you can see HIV particles doing all kinds of things but the question has been, which ones are infectious, and what do they do that is different. The answer appears to be, arrive at the nuclear pore as a free capsid and interact directly with nuclear pores. In other words, if HIV-1 is located within endosomes that promote nuclear envelope invaginations for nuclear import, how do the authors explain interactions between cytoplasmic co-factors and capsid regulating infection and nuclear transport? How would these co-factors access the capsid within endosomes? The authors need to present a model in which their data stacks up with previous work.

2. I'm afraid the conclusion that infection of resting T cells is dictated by Orp3 phosphorylation is not supported by the data. There is no measurement of Orp3 phosphorylation. It is true they see 2 bands in the activated cells and one in the resting cells but we don't know this is phosphorylation, it could be another PTM. We also don't know that this is why the VAP complex interacts in activated but not resting cells on HIV infection or that this additional band, whatever it is, dictates HIV permissivity. To make these claims the authors need to provide some direct evidence, not indirect observation.

3. Also please bear in mind the previous literature. Many studies show viral DNA synthesis is poor in resting T cells, explaining poor infection. One study assigns this to SAMHD1 expression (PMID: 22972397). I think its OK to make these measurements and form hypothesis, but the current version feels like conclusion with very little evidence for the model and more or less complete disregard for an extensive literature.

4. I don't like the title because I feel it trivialises the importance of the study's findings. I appreciate that its true as stated but it doesn't mention the new HIV cofactors discovered here, which I would expect to see in the title. Nuclear invaginations feel like a rather imprecise description and I imagine many things, including artefacts?, might induce nuclear invaginations. I would like to make the case that what the study has discovered is a really important role for the VOR complex in HIV nuclear entry and infection, which is important.

I don't want to insist on a tile change but I expect that something like this would attract a greater HIV readership.

"The VOC complex regulates HIV nuclear entry and explains infection of activated T-cells"

Of course, the authors should choose their title.

5. I would really like to see a viral titration correlating the increase in NEI's as titre increases with the level of infection. This would make this study much more compelling and really tightly link NEI induction to infection. I think that the percent infection where measured is slightly higher than the NEI count. Does this hold over a titration. Keen to see that, with some explanation/discussion, even if hypothetical.

6. Figure 2E, I'm not seeing puncta. Are there puncta? Can we see better images?

7. Line 176 and throughout, I've never heard the word immunoisolation. I think, in the HIV field, we call this immunoprecipitation. If this is right, please change throughout. Its semantics but helpful for clarity.

8. Line 179 ORP3 is not obviously appearing as a double band here. Perhaps the authors can include a better example. This may be true but its not clear here. Resolve with a different gel density?

9. Line 177, its not clear how the IC50s were calculated here. There is no titration. Please expand on how this was done and show data.

10. In figure 3, the drug concentrations used are very high and apparently way above what is required to inhibit HIV infection. Could the authors titrate the drug down and again, relate the level of infection to the number of NEI formed? Do they correlate over different levels of infection in a linear way. Such an experiment could really support the proposed mechanism. Without titration to change NEI count and infection we can't easily quantitate how these 2 things relate quantitatively.

11. Line 218, I don't understand the connection between cells being stuck to the plate or not and the NEI infection connection. Does this relate to the fact that Hela make NEI but they're only seen on infection in T cells? Please explain.

12. Line 223, this point would be much more compellingly made with titrations of virus or drug that allow NEI/infection to be related quantitatively over a range of infectiousness. I refer to point 5.

13. It would be nice to see quantitative data for 4J. 1 cell is not compelling.

14. Line 243, please explain what the FRET data are that the data herein agree with.

15. Line 246 Figure 4L and S9A, please explain what nocodazole does, ie inhibits microtubules and thereby prevents virus reaching the nucleus. Readers may not get this without further detail.

16. End of page 13 the text drifts away from explaining the experiments. Please explain better. For example, the gel shown (5H) doesn't show any VAP-A interaction. Ie explain 5I in the text.

17. There is a literature suggesting that the HIV-1 Vpr protein causes nuclear invagination, or at least nuclear envelope disruption. Of course, Vpr is in the particle. There's also a recent hint that Vpr helps HIV infection of resting T cells. See PMID: 35417711. The authors might read those studies and tell us what they think (if they make any sense) in the discussion. This feels relevant but it may not be and they should decide.

REVIEWER COMMENTS

Reviewer #1 (Remarks to the Author):

Reviewer's comment:

The manuscript by Santos and colleagues reports their results obtained while investigating the entry of HIV into the nucleus of an infected cell. They demonstrate that nuclear entry of endocytosed HIV is associated with invaginations of the nuclear envelope (NEI), which involve a complex of VAP23A, hyperphosphorylated ORP3, and Rab7. They also show that drugs targeting the VAP23A-ORP3-Rab7 complex inhibit HIV nuclear entry and infection. This work follows on a series of studies by Melikyan's group, who suggested that HIV fuses with endosomes, rather than plasma membrane. This notion was confirmed in later studies, although not for all HIV-susceptible cell types. It appears to hold for CD4+ T cells, the major target of HIV, so results of the current study are highly relevant and important. Nevertheless, a number of key questions remain obscure, so the reader is left without a clear understanding of the process. The issues listed below need to be addressed experimentally, or at least discussed based on available evidence.

Author's response:

We are pleased to read that the reviewer appreciated our work and noted that it was important. We thank him/her for the constructive comments. We have experimentally addressed all of her/his concerns.

Reviewer's comment:

1. The authors show that NEI are induced by late endosomes carrying HIV. Late endosomes are commonly associated with the lysosomal degradation pathway. It is important to determine why late endosomes with HIV inside do not acidify, or if only a small proportion does not acidify what determines this protection.

Author's response:

We addressed this intriguing question using different markers associated with either late endosomes/multivesicular bodies or lysosomes. These data are presented in the new Figure 3. We now show that alongside Rab7, the exosomal marker CD63, typically associated with luminal vesicles in late endosomes/multivesicular bodies, is co-labeled with HIV-1 integrase (IN-2) in nuclear envelope invaginations (NEIs), whereas Lamp1, a lysosome marker, is excluded. Quantification of double-immunolabeled cells indicates that only Rab7⁺, but not Lamp1⁺ organelles, penetrate and/or induce type II NEIs. As indicated in the revised Discussion, it remains to be determined whether endocytosed viral particles impact the composition and/or maturation of late endosomes as well as their cytoplasmic transport into the NEI core. The mechanism by which a subset of late endosomes avoid acidification doesn't seem to be clear in the literature, assuming it is not simply based on probabilistic encounters. We hope that our findings will spur studies aimed at a deeper understanding of the decision-making process of late endosomes to acidify and its timing.

Reviewer's comment:

2. The time course is important to determine, as this has a big impact on the viral infection. Does virus-endosome fusion occur when endosomes dock to the MTOC? How is this timing regulated?

Author's response:

Our time course experiments (now shown in Figure 2) indicated that viral particles, as monitored by HIV-1 integrase (IN-2), move from early to late endosomes, where they accumulate at the

MTOC. A fraction of them then end up in NEIs, as shown by colocalization of Rab7 and IN-2, suggesting that virus-endosome fusion occurs in the cytoplasmic core of NEIs, not at the MTOC. Of course, such fusion could also occur at the MTOC, as suggested by HeLa cells lacking VAP-A or ORP3, or those treated with itraconazole or PRR851, where accumulation of Rab7⁺ late endosomes containing IN-2 is restricted considering their absence in NEIs. In the latter cases, though, neither nuclear transfer of viral components nor productive infection was observed, suggesting that late endosome entry into the NEI and virus-endosome fusion therein are essential.

Reviewer's comment:

3. How do NEI help virus to pass through the nuclear pore? Are they just a by-product or a functional structure? ORP3 findings do not address this question, as the complex may be needed for nuclear pore function as well as for NEI generation.

Author's response:

First, NEIs are not just a byproduct, as their appearance is related to productive infection. In a new set of experiments, we correlated NEI formation and productive infection with the amount of virus applied (see new panel I in Figure 6). This issue is now introduced in the revised Results section:

“To evaluate the relationship between the formation of NEIs and productive infection, we infected PHA/IL-2-activated T cells with different doses of virus (MOI ranging from 0.2 to 4) and determined the percentage of cells with induced NEIs relative to the amount of EGFP⁺ cells. First, we observed a consistent increase in both events with increasing virus concentration (Fig. 6I). Second, regardless of the amount of virus applied, there was a tight correlation ($r = 0.88$) between the amount of cells showing NEIs and EGFP expression (Fig. 6I), confirming that NEI induction is a prerequisite for productive infection.”

Moreover, the relationship between NEIs and productive infection is also illustrated in infected quiescent T cells where an accumulation of virus-laden late endosomes at a nuclear pole is detected without producing NEIs and without generating productive infection (Figure 7a-g). In addition, in the revised version we employed Sotrastaurin, a protein kinase C (PKC) inhibitor, and found that it inhibits NEI formation, productive infection and, perhaps coincidentally, ORP3 hyperphosphorylation. Altogether, these observations suggest that NEIs are actively involved in the nuclear transfer of HIV-1 components. The exact mechanism remains to be dissected as well as the advantage of using NEIs for such process.

Second, we addressed these issues in the revised Discussion section. In particular, we pointed out that the cytoplasmic core of NEIs could create a protective environment for HIV-1 components where they could interact with host factors. Exclusion of Lamp1⁺ lysosomes from this core could also prevent fusion of late endosomes with them, and hence protect HIV-1 degradation. In addition, by penetrating deep into the nuclear compartment, often reaching the nucleolus, NEIs could facilitate HIV-1 transfer to specific nuclear sub-compartments. Finally, it is also possible that the extreme curvature of nuclear membrane in NEIs facilitates the passage of the intact ~60 nm HIV-1 capsid (Zila et al. 2021, PMID: 33571428), which is much larger than the previously proposed ~40 nm central channel diameter of the human nuclear pore complexes. Of note, a recent publication by Bhargava et al. (2021, PMID: 34592156) has shown a relationship between nuclear membrane organization mediated by its structural components and HIV-1 infectivity has recently been demonstrated. Such information is now added in the revised Discussion.

Reviewer's comment:

4. Findings with the drugs targeting the VAP23A-ORP3-Rab7 complex are very important and require standard time-course analysis of the drugs' effect on viral replication, preferably with primary HIV isolates and CD4+ T cells.

Author's response:

We agree with the reviewer and this important issue was addressed using CD4+ T cells. The new data are presented in Figure 6f, g. In summary, we have shown that it is essential to add the drug (PRR851) before infection, not after, to prevent productive infection. In addition, we also investigated, using HeLa cells, the impact of PRR851 concentration on HIV-1 integrase accumulation in NEIs and productive infection (see new Supplementary Figure 7b).

Reviewer's comment:

5. Animal experiments are out of place here, as no HIV infection has been done in mice. Demonstration that mice do not lose weight is meaningless without knowing the effective dose, measuring PK and PD. Instead, the authors should do a more careful analysis of drug toxicity in primary T cells.

Author's response:

We have performed the MTS assay on CD4+ T cells as we did on HeLa cells. These data are presented in a new Supplementary Figure 8b. As suggested by the reviewer, we have removed the animal experiments.

Reviewer's comment:

6. Fig. 1: just one cell is shown at high resolution. Please show several cells at lower resolution.

Author's response:

First, it is important to mention that each NEI is located in a given section of the x-y plane of the nucleus, which varies from cell to cell. Therefore, the probability of having a cross-section of several cells with NEIs is very low. For these reasons, we have shown representative images throughout the manuscript. This issue is now introduced in the revised Results section:

"Of note, the entire nuclear compartment needs to be scanned because NEIs are restricted to certain optical x-y planes. Thus, IN-2 within the NEIs can appear at any z-level of the nucleus, a phenomenon that varies from cell to cell."

Second, we have added as requested lower resolution images as well as all x-y optical sections through the nucleus and composite images of all optical sections. See the new Supplementary Figure 11.

Reviewer's comment:

7. There is no staining for endosomes, how do the authors know that endosomes are involved?

Author's response:

In addition to the use of RFP fusion proteins (Rab5 and Rab7), we have now provided immunostaining of Rab7 and its colocalization with HIV integrase (IN-2) in NEIs (see new Figure 3a), where cross-sections and longitudinal sections of NEIs are shown. Besides Rab7, we now show that CD63, a marker of intraluminal vesicles found in late endosomes/multivesicular bodies, also colocalizes with IN-2 in NEIs (Figure 3b), whereas LAMP1, a marker of lysosomes, does not (Figure 3c-e). In these experiments, NEIs were highlighted with VAP-A-GFP.

Reviewer's comment:

8. Fig. 5: there is no evidence that ORP3 hyperphosphorylation is essential. The only conclusion that can be made is that it associates with complex formation nuclear entry. To show necessity, mutagenesis experiments should be involved.

Author's response:

First, we and others have previously demonstrated that the phosphorylation of ORP3 is important for its interactions with VAP-A (Weber-Boyvat et al. 2015, PMID: 25447204; Santos et al. 2021, PMID: 34429859). For example, treatment of ORP3 with λ -phosphatase abolished its interactions with VAP-A and Rab7. The action of λ -phosphatase was demonstrated by a shift of ORP3 mobility on electrophoresis gel (Santos et al. 2021, PMID: 34429859), where the hyperphosphorylated form corresponds to the slower migrating band (Figure 5H, now Figure 7h). In the revised manuscript, we treated the activated T cells with λ -phosphatase and showed the same shift of the molecular weight of ORP3 (see new Figure 7j).

Second, given that ORP3 phosphorylation is very complex (i.e. multiple sites; see Weber-Boyvat et al. 2015, PMID: 25447204) and PKC has been suggested to be involved, among others (Lehto et al. 2008, PMID: 18270267), we incubated CD4⁺ T cells with Sotrastaurin, a PKC pan-inhibitor, and showed that such a treatment reduced the ORP3 expression and its phosphorylation (see new Figure 8). The latter phenomena mimic the situation observed in quiescent T cells. In both cases, quiescent T cells and those incubated with Sotrastaurin, neither NEI formation nor productive infection was observed. Even with these new data, we cannot exclude that other factors impacted by PKC inhibition are involved and therefore we have modified our statement that ORP3 hyperphosphorylation is essential (see our answers to other reviewers below).

Reviewer #2 (Remarks to the Author):**Reviewer's comment:**

Santos et al propose a very intriguing mechanism by which HIV-1 viruses in endosomal vesicles invaginate the nuclear envelope and enter the nuclear compartment. Although this is a very interesting hypothesis, the data presented here is not convincing and lack important experiments and controls. Imaging helps, but the authors need biochemical evidence, where all viruses are accounted, i.e., cellular fractionation into nuclear and cytosolic content for the same type of experiments. Isolation of these EVs by gradient-centrifugation etc. The authors showed that depletion of VAP-A and the use of ICS blocks HIV-1; however, the stage of the block is not studied (this should correlate with their model). MOIs are not provided. Overall, the data presented here does not support the conclusions.

Author's response:

We agree with the Reviewer's criticism and we have addressed each issue as answers to the specific points below raised by her or him. Throughout the manuscript, we used multiplicities of infection (MOI) of 2 unless otherwise noted. This is now indicated at the beginning of the revised results section as well as in the figure legends.

Reviewer's comment:**Major issues:**

Fig1A: bald virus should show IN in vesicles in the inside the cytosol since bald particles do not have VSV-G; however, they have all the other components (i.e. IN), and they are endocytosed. The bald infected control used by the author is not correct (after an hour, the authors should see vesicles and virus in the surface). Lack MOI information.

Author's response:

We thank the reviewer for raising this issue. We used "bald" virus as negative control because cell entry is impaired without VSV-G. Even though a small fraction of HIV-1 is endocytosed, we think it is still a valid control. We have now repeated our experiments and, as correctly anticipated by the reviewer, we show that most of HIV-1 integrase (IN-2) is located outside the cells, near their membrane (see new Figure 3g and Supplementary Figure 11c), as in cells treated with dynasore (Figure 2A, DNS). Moreover, as pointed out by the reviewer, we could observe a tiny fraction of IN-2 in the cytoplasmic compartment in few cells. We modified the text accordingly:

"Almost no IN-2 was observed in the cytoplasmic compartment of cells infected with "bald" virus for 1 hour (Fig. 3g, inset g1). In the latter case, an accumulation of IN-2 was observed outside the cells, similar to the DNS treatment. Only a minute fraction of IN-2 from "bald" virus was able to enter some cells in a non-specific manner (see below), which is consistent with other studies^{47, 48}." MOI is now indicated.

Reviewer's comment:

Fig1A and B: the authors need to show how many of these events they see (nuclear envelope invaginations), is this rare? What is the MOI. Statistics are important here. Showing images of single cells is not convincing.

Author's response:

We have now clarified the text (lines 117-124) to better show that we indeed quantified the number of NEIs (type I and II) per 100 HeLa cells (Figure 1g) and per single cell (Supplementary Figure 2A, now Supplementary Figure 3a), and the impact of viruses on them (Supplementary Figure

2B, now Supplementary Figure 3b). Thus, in HeLa cells, the findings suggest that NEIs are not rare events, but their number increases after infection. For CD4⁺ T cells, almost no NEI is observed in the absence of infection (Figure 4F, now Figure 6i). These data were also quantified.

In addition, we quantified the number of NEIs containing HIV-1 integrase (IN-2) after 1 hour of infection. These data were already presented in Figures 2B and 3E (now Figure 4b and Figure 5e).

As mentioned above, we used a MOI of 2 for these experiments unless indicated.

Reviewer's comment:

Fig1H: DNS should also accumulate particles in the surface, or near the surface, of the cell. Again showing one picture does not provide convincing evidence. One of the very well-known problems of imaging HIV-1 is that bald particles can be seen in the cytoplasm(in endocytic vesicles); Therefore the use of dynasore (DNS) should allow the investigator to see particles in the surface/near-the-surface of the cell.

Author's response:

We clarified the description of the data that showed the presence of VSV-G-pseudotyped viral particles (see IN-2 immunolabeling) outside the cells and near the surface after dynasore treatment (Fig. 1H, right panel, asterisks, now Figure 2a, right panel, green asterisk). This information was indicated in the legend of the corresponding figure, and we have now indicated such information in the main text (Results) of the revised manuscript. Fig. 1H was not related to the Bald virus, but we repeated the experiments with Bald virus (see new Figure 3g) as mentioned above, and we now show all x-y sections of a low power view in Supplementary Figure 11c. Therein, we observed that the vast majority of IN-2 is found outside the cells. Only few sporadic IN-2 signals appear inside few cells indicating that the non-specific uptake is minor under our experimental setting. We revised the Results section accordingly (see previous point).

Reviewer's comment:

Fig2A: the authors need a proof of knockdown expression for VAP-A, VAP-B and ORP3. This Figure needs an MOI.

Author's response:

In the revised manuscript, we have now added a new set of images showing the absence of IN-2 in NEIs of cells lacking VAP-A and ORP3, but not in scrambled control and shVAP-B. See new Figure 4a (top panels). In the original version of the manuscript, we demonstrated the silencing of VAP-A, VAP-B and ORP3 by immunoblotting and quantified their reduction (see original Supplementary Figure 4, now Supplementary Figure 5). Likewise, we showed that the lack of VAP-A and ORP3 impeded the productive infection (Figure 2E, now Figure 4e).

As indicated above, we have now indicated that we used a MOI of 2 for all experiments unless noted otherwise.

Reviewer's comment:

Fig. 2F: needs a curve of infection using different amounts of virus when depleting VAP-A, not only one point. In addition, the investigators need to figure out where the block is when depleting VAP-A (reverse transcription, nuclear import, integration etc..)

Author's response:

We performed the requested experiments using an MOI of 8 instead of 2, and the new data are described in the revised Results section as follows:

“Raising the MOI (e.g., 8 instead of 2) partially increased the number of GFP⁺ cells in VAP-A-deficient cells from $4.56 \pm 2.03\%$ to $8.07 \pm 1.71\%$, which is lower than the corresponding number in scrambled shRNA, $48.81 \pm 4.82\%$, but in line with the remaining amount of VAP-A (Supplementary Fig. 5a).”

The depletion of VAP-A results in a block of nuclear import of both HIV-1 (this study) and extracellular vesicles (Santos et al., 2021, PMID: 34429859) and corresponds to the effects of PRR851, that disrupts the VOR complex, preventing nuclear import. We trust that our study will spur further investigations on the larger issues of timing of capsid disassembly and reverse transcription in relation to the newly discovered nuclear pathway and its genetic or pharmacological inhibition. We added the following paragraph to the Discussion section to make clear the need for additional studies in this area:

“Further studies will be required to identify the site(s) of reverse transcription of the viral RNA genome into double-stranded DNA in relation to the VOR complex-mediated nuclear pathway. Because of the rarity of reverse transcription events in an infected cell, the sub-viral reverse transcription complex is currently poorly characterized^{28, 86}. Our findings are consistent with reverse transcription occurring in intact capsids^{87, 88}, perhaps inside endosomes located outside and/or in NEIs. The possible link between reverse transcriptase activity and HIV-1 uncoating⁸⁹, would suggest, however that this event occurs after the release of HIV capsid in the core of NEIs and near NPCs^{30, 31}. Our model could also support the evidence that reverse transcription is completed after HIV-1 nuclear import inside the nucleus^{32, 33, 79}.”

Reviewer's comment:

Fig3: the assay that quantifies SUN2+ NEI containing IN-2 should be validated biochemically by fractionating cells into nuclear and cytosolic content, or isolation of these EVs by gradient-centrifugation etc. This assay is used all over the manuscript and is not clear to this reviewer how reliable it is since the bald viruses show no virus in the cell, which is unlikely.

Author's response:

We agree with the reviewer. These biochemical experiments were performed and new data, which support the information in the original manuscript, are presented in new panel g of Figure 5 (see below). Again, both itraconazole and PRR851 blocked the nuclear transfer of HIV-1 integrase.

Figure 5g. Itraconazole and PRR851 block the nuclear transfer of HIV-1 integrase. Drug-treated HeLa cells as indicated were infected with HIV-Gag-iGFP for 3 hours, then fractionated into cytoplasmic and

nuclear fractions and probed by immunoblotting for HIV IN. The 1/4 and total fractions were loaded for cytoplasmic and nuclear fractions, respectively. As controls, fractions were immuno-probed for GAPDH (cytoplasmic marker) and Lamin B1 (nuclear marker). Note the absence of IN-2 in nuclear fraction of ICZ and PRR851-treated cells.

The EVs used were characterized by nanoparticle tracking analysis using ZetaView (Particle Metrix GmbH, Meerbusch, Germany). Our EV preparations were also characterized by immunoblotting for the presence and absence of particular markers, three-dimensional direct stochastic optical reconstruction microscopy, and electron microscopy in our previous studies (Santos et al. 2021, PMID: 34429859; Rappa et al, 2013, PMID: 23318676) and this information was submitted to EV track (EV-TRACK, <https://evtrack.org/>; ID: EV210180, author: Santos). The details above are indicated in the revised Methods section.

As mentioned above, we repeated our experiments with Bald virus, and only sporadic and limited labeling of HIV-1 integrase (IN-2) was found in some cells, while the vast majority of IN-2 remained outside the cells and near their membrane (see new Figure 3g, g1 and Supplementary Figure 11c).

Reviewer's comment:

Fig 3: the authors show that ICZ blocks viral infection, they should provide the stage of the block (reverse transcription, nuclear import, integration etc..)

Author's response:

As monitored by the trafficking of HIV-1 integrase (IN-2), our data have shown that nuclear import is blocked, and consequently, all following steps, including the viral integration, are also blocked. The larger issue associated with when reverse transcription occurs relative to nuclear transport is more complex, as discussed above, and it requires significant study beyond what can be done in this already very long revised manuscript. We have addressed these issues with the appropriate references in the revised Discussion section (see answer to a previous comment). As mentioned above, we trust that our study will spur further investigations on the timing of capsid disassembly and reverse transcription in relation to the newly discovered nuclear pathway and its genetic or pharmacological inhibition.

Reviewer's comment:

Fig4B: what is the stage at which HIV-1 is inhibited by these drugs(reverse transcription, nuclear import, uncoating, integration etc..)

Author's response:

We share the reviewer's eagerness to seek answers about the timing of capsid disassembly and reverse transcription (RT). As indicated above, a paragraph was added to the Discussion to point out that the mechanism for nuclear entry that is identified and supported in this manuscript does not exclude current theories regarding the timing of RT. Given the substantial work needed to shed light on this important and unanswered question about the HIV life cycle and the size of the current manuscript, we feel that this question is a future direction for this work. See answer to the point above.

Reviewer's comment:

The paper is written with very little information in the text and figure legends. For a reviewer to assess the data, the authors need to provide important details, MOI, time of infection, how the quantification of IN+ is performed, etc. The paper is hard to read, please describe the experiments in the figure legends better.

Author's response:

We have significantly improved the writing of the manuscript by adding details about the experiments and the purposes of the experiments. We hope that our manuscript will not be too long, but we are willing to shorten it if necessary. We apologize for the lack of information. In particular, we have specified the MOI and time of infection for every experiment and described the quantification of HIV-1 integrase (IN-2) positivity in the Immunocytochemistry section of the Methods.

Reviewer's comment:

Line 53-59: references suggesting that nuclear import of the HIV-1 core precedes uncoating are missing.

Author's response:

We added the relevant references and discussed this issue in detail in the revised Discussion section, as mentioned above.

Reviewer #3 (Remarks to the Author):**Reviewer's comment:**

Santos and colleagues have studied the role of the VOR complex comprising VAP-A, ORP3 and Rab7 in HIV infection. They provide data suggesting that it is an essential cofactor for efficient HIV infection and possibly a determinant of T cell infection that may contribute to explaining why activation of T cells is required to make them permissive, at least in vitro. It's an intriguing study and I enjoyed reading it. I can see that these authors have not worked previously in HIV and as such may not be familiar with the enormous body of literature relating to HIV traffic in cells via endosomes and the process of capsid nuclear entry. They also haven't discussed modern models of what dictates resting T-cell infection and I'm afraid that without incorporating, at least discussing, some of the key dogmas of HIV biology, this study is going to get a tough time in review and not be widely appreciated by the field.

Author's response:

We are delighted to hear that the reviewer has appreciated our effort. Her/his comments and suggestions were instrumental in improving our manuscript and discuss the current status of the vast literature on HIV-1 intracellular transport. For more details, see below.

Reviewer's comment:

I like it because many of the experiments are compelling and are clearly telling us something important that we didn't know about cofactor requirements for HIV infection. But at the moment the study is written without taking into account some of the key HIV literature and in places the authors do not support the conclusions with data.

Author's response:

Again, we are pleased to read that the reviewer liked our study. Following her/his suggestions, we have significantly improved our manuscript by including new experiments, especially those related to dose-response relationships. We have revised the Introduction and Discussion sections in light of the current literature.

Reviewer's comment:

I'd like to suggest re-consideration of some key points.

Author's response:

We appreciate the reviewer's effort and her/his suggestions to improve our manuscript.

Reviewer's comment:

1. The HIV particles that actually infect cells are not thought to be in endosomes when they arrive at the nucleus. They are thought to be naked in the cytoplasm and able to interact with cytoplasmic and nuclear pore associated cofactors as a prerequisite to nuclear transport. As well as all the studies describing capsid binding cofactor dependence, there are the microscope studies eg (Zila et al Cell 2021), and also from Vinay Pathak, that confirm that HIV is not in endosomes at this point. Its important to understand that you can see HIV particles doing all kinds of things but the question has been, which ones are infectious, and what do they do that is different. The answer appears to be, arrive at the nuclear pore as a free capsid and interact directly with nuclear pores. In other words, if HIV-1 is located within endosomes that promote nuclear envelope invaginations for nuclear import, how do the authors explain interactions between cytoplasmic co-factors and capsid regulating infection and nuclear transport? How would these co-factors access the capsid

within endosomes? The authors need to present a model in which their data stacks up with previous work.

Author's response:

Before interacting with the nuclear pores, the HIV-1 capsid will escape from the endosomal compartment upon fusion of the viral particles with the limiting membrane of the late endosomes. In the revised Discussion section, we have now pointed out these issues as well as the potential interaction of HIV-1 capsid with host proteins such as cyclophilin A and cleavage and polyadenylation specificity factor 6 and nuclear pore components, as follows:

“The endosome-based, NEI dependent pathway for HIV-1 infections suggests some hypotheses about the timing and location of interactions with cytoplasmic co-factors and components of the nuclear membrane. Nuclear import could be facilitated by the exit of viral components from the late endosomes, i.e. after HIV-1 fusion with the limiting membrane, and their concentration in the core of the NEIs, which could create a protective microenvironment and favor their interactions with cytosolic factors and NPC components (e.g., Nup358, Nup153) en route to nucleoplasm^{26, 63-65}. Likewise, the exclusion of Lamp1⁺ lysosomes from NEIs would prevent late endosome-lysosome fusion. The segregation of HIV-1 particles and Lamp1⁺ lysosomes has been previously noted in CD4⁺ T cells⁶⁶. Determination of the binding of HIV-1 capsid to host proteins such as cyclophilin A and cleavage and polyadenylation specificity factor 6 in relation to the perinuclear region, particularly NEIs, will be of interest^{67, 68}. By penetrating deep into the nuclear compartment, often reaching the nucleolus^{34, 36}, NEIs may facilitate the transfer of HIV-1 to specific nuclear subcompartments⁶⁹⁻⁷². This is an important consideration because viral complexes can bypass intranuclear movement or diffusion to reach integration sites.

Our data do not contradict the fact that most of the endocytosed viruses end up in the lysosomal compartment, as observed by the colocalization of HIV-1 IN and Lamp1 in the cytoplasmic compartment, or that a fraction of them are released from endosomal compartment outside the NEIs, where interactions with host cell factors might determine the fate of viral particles and infectivity. Within the NEIs (Fig. 6q, right panel), it is also possible that the extreme curvature of nuclear membrane may facilitate the passage of the intact ~60 nm HIV-1 capsid²⁸, which is much larger than the previously proposed ~40 nm limit for the central channel diameter of the human NPCs^{73, 74}. These physical constraints could impact the structure and/or composition of NPCs in terms of nucleoporins, which is consistent with the heterogeneity of NPCs^{33, 75}.”

To summarize a part of our findings, we described a model in Figure 6q (see below).

Figure 6q. Schematic representation of the induction of type II NEIs by virus-laden late endosomes, a process mediated by the interaction of VOR complex proteins, namely ONM-associated VAP-A, cytoplasmic ORP3 and late endosome-associated Rab7 (left panel). Release of viral components from late endosomes into the cytoplasmic core of induced NEIs at the vicinity of the nuclear pore would facilitate their transfer to the nucleoplasm (right panel). PRR851 inhibits the interaction of the VOR complex proteins, and hence the NEI formation.

Reviewer's comment:

2. I'm afraid the conclusion that infection of resting T cells is dictated by Orp3 phosphorylation is not supported by the data. There is no measurement of Orp3 phosphorylation. It is true they see 2 bands in the activated cells and one in the resting cells but we don't know this is phosphorylation, it could be another PTM. We also don't know that this is why the VAP complex interacts in activated but not resting cells on HIV infection or that this additional band, whatever it is, dictates HIV permissivity. To make these claims the authors need to provide some direct evidence, not indirect observation.

Author's response:

First, we performed a new experiment showing the shift of the upper band of ORP3 after λ -phosphatase treatment of detergent cell lysates prepared from PHA/IL-2 activated T cells (new Figure 7j). Thus, we were able to exclude other post-translational modifications responsible for this slower migratory ORP3 band. These data are in line with those we recently reported where the enzymatic dephosphorylation of ORP3 impeded its binding to VAP-A and Rab7 (Santos et al. 2021, PMID: 34429859).

Second, we are not claiming that the hyperphosphorylation of ORP3 is the direct cause of HIV permissivity; we simply observed a correlation between the status of ORP3 phosphorylation and the viral infection of CD4⁺ T cells. The latter is further supported by treatment of T cells with Sotrastaurin, a PKC pan-inhibitor, during their activation and infection, which prevents the formation of the VOR complex and HIV-1-induced NEIs. Productive infection is also inhibited under these conditions (see new Figure 8), as it is in HeLa cells lacking ORP3. We have now modified the Discussion accordingly, as follows:

“Thus, PKC-mediated hyperphosphorylation of ORP3 in PHA/IL-2-activated T cells could, among other factors, contribute to their permissiveness, while quiescent cells were refractory to productive infection. As PKC activity is required for T cell activation and proliferation⁵⁹⁻⁶¹, the fact that ORP3 status is similar between quiescent and Sotrastaurin-incubated cells during PHA/IL-2-induced activation could be more than coincidental. Little is known about ORP3 function(s), but an ORP3 knockout has been reported to impact the expansion of lymphoid progenitors and favor aneuploidy⁶². Further studies should investigate the impact of the absence of ORP3 on HIV-1 infection in vivo.”

Reviewer's comment:

3. Also please bear in mind the previous literature. Many studies show viral DNA synthesis is poor in resting T cells, explaining poor infection. One study assigns this to SAMHD1 expression (PMID: 22972397). I think its OK to make these measurements and form hypothesis, but the current version feels like conclusion with very little evidence for the model and more or less complete disregard for an extensive literature.

Author's response:

We have now been more careful to at least mention in the Discussion the vast HIV literature and substantial work required to fully integrate this new pathway into the existing models of HIV infection. We have now 104 publications in the Reference list. As mentioned above, we now provided additional data showing that the inhibition of PKC impedes the phosphorylation of ORP3, induction of nuclear membrane invagination and productive infection.

Reviewer's comment:

4. I don't like the title because I feel it trivialises the importance of the study's findings. I appreciate that its true as stated but it doesn't mention the new HIV cofactors discovered here, which I would expect to see in the title. Nuclear invaginations feel like a rather imprecise description and I imagine many things, including artefacts?, might induce nuclear invaginations. I would like to make the case that what the study has discovered is a really important role for the VOR complex in HIV nuclear entry and infection, which is important. I don't want to insist on a title change but I expect that something like this would attract a greater HIV readership. "The VOR complex regulates HIV nuclear entry and explains infection of activated T-cells" Of course, the authors should choose their title.

Author's response:

We fully agree with the Reviewer's suggestion. We have changed the title in: "HIV-1-induced nuclear invaginations mediated by VAP-A, ORP3, and Rab7 complex explain infection of activated T-cells: A novel targetable pathway".

Reviewer's comment:

5. I would really like to see a viral titration correlating the increase in NEI's as titre increases with the level of infection. This would make this study much more compelling and really tightly link NEI induction to infection. I think that the percent infection where measured is slightly higher than the NEI count. Does this hold over a titration. Keen to see that, with some explanation/discussion, even if hypothetical.

Author's response:

We agree with the reviewer and we have now experimentally addressed her/his comment using CD4⁺ T cells. The use of increasing amounts of HIV-89.6-EGFP MOI (0.2, 0.4, 1, 2, and 4) correlated with increasing numbers of type II NEIs and EGFP⁺ cells as a measure of productive infection. These new data are presented in Figure 6l. The text has been rewritten accordingly:

To evaluate the relationship between the formation of NEIs and productive infection, we infected PHA/IL-2-activated T cells with different doses of virus (MOI ranging from 0.2 to 4) and determined the percentage of cells with induced NEIs relative to the amount of EGFP⁺ cells. First, we observed a consistent increase in both events with increasing virus concentration (Fig. 6l). Second, regardless of the amount of virus applied, there was a tight correlation ($r = 0.88$) between the amount of cells showing NEIs and EGFP expression (Fig. 6l), adding further evidence that NEI induction is a prerequisite for productive infection."

Reviewer's comment:

6. Figure 2E, I'm not seeing puncta. Are there puncta? Can we see better images?

Author's response:

We now indicated them, i.e. GFP signal at the plasma membrane, with green arrowheads (see new Figure 4e and Figure 5i). Such punctate and scattered Gag-GFP signals were previously observed by other investigators (Hübner et al. 2007, PMID: 17728233).

Reviewer's comment:

7. Line 176 and throughout, I've never heard the word immunoisolation. I think, in the HIV field, we call this immunoprecipitation. If this is right, please change throughout. Its semantics but helpful for clarity.

Author's response:

Our approach is based on immuno-affinity purification using super paramagnetic beads conjugated to protein G (Miltenyi Biotec). The immune complex is recovered using a column placed in the magnetic field of a separator (i.e. a magnet). Unlike conventional immunoprecipitation using Sepharose/agarose beads for example, neither sedimentation nor precipitation (i.e. centrifugation) is required to isolate the protein of interest. For these reasons, we have used the term "immunoisolation". To clarify this issue, we now indicated the paramagnetic bead-based system used in the revised Results section.

Reviewer's comment:

8. Line 179 ORP3 is not obviously appearing as a double band here. Perhaps the authors can include a better example. This may be true but its not clear here. Resolve with a different gel density?

Author's response:

Because we use the same blots to probe ORP3 with VAP-A or Rab7, we limit sample migration by gel electrophoresis. The double bands of ORP3 in HeLa cells are highlighted in the original Supplementary Figure 4 (now Supplementary Figure 5a) and the new Supplementary Figure 7a (see point #9, below). The hyperphosphorylation of the upper band was further demonstrated in PHA/IL-2-activated T cells by treating them with λ -phosphatase (see new panel j in Figure 7).

Reviewer's comment:

9. Line 177, its not clear how the IC50s were calculated here. There is no titration. Please expand on how this was done and show data.

Author's response:

This information is now added as a new paragraph in the revised Methods section. Moreover, we provided all data concerning this issue, i.e. immunoblot and linear regression approach used to estimate the IC50. These data are presented in a new Supplementary Figure 7a.

Supplementary Fig. 7. Impact of PRR851 concentration on Rab7 binding to ORP3, entry of HIV-1 IN in NEIs and productive infection. See manuscript for the corresponding legend. Note the double bands of ORP3 (a).

**Reviewer's comment:**

10. In figure 3, the drug concentrations used are very high and apparently way above what is required to inhibit HIV infection. Could the authors titrate the drug down and again, relate the level of infection to the number of NEI formed? Do they correlate over different levels of infection in a linear way. Such an experiment could really support the proposed mechanism. Without titration to change NEI count and infection we can't easily quantitate how these 2 things relate quantitatively.

Author's response:

To complement and strengthen the HeLa cell data presented in the original Figure 3 (now Figure 5), we evaluated the effects of lower concentrations of PRR851 on the presence of HIV-1 integrase in NEIs and productive infection. These data are now presented in a new Supplementary Figure 7b. The text has been rewritten accordingly:

"The effects of PRR851 on the accumulation of HIV-1 IN in NEIs and productive infection were concentration-dependent (Supplementary Fig. 7b). IN-2⁺ NEIs and GFP expression also decreased in concert ($r = 0.91$) with increasing levels of PRR851."

Reviewer's comment:

11. Line 218, I don't understand the connection between cells being stuck to the plate or not and the NEI infection connection. Does this relate to the fact that HeLa make NEI but they're only seen on infection in T cells? Please explain.

Author's response:

First, we have completely rewritten this section, better explaining the experiments performed, their rationale and the resulting data.

Second, we have expanded our ideas on the link between suspension growth versus adhesion and NEI formation. The text has been rewritten accordingly:

"Unlike adherent HeLa cells, CD4⁺ T cells grow in suspension, which has an impact on their morphology: the former are very spread out while the latter are rounded. These morphological alterations could influence the formation and/or induction of NEIs. This prompted us to re-examine the impact of viral infection on NEIs."

Reviewer's comment:

12. Line 223, this point would be much more compellingly made with titrations of virus or drug that allow NEI/infection to be related quantitatively over a range of infectiousness. I refer to point 5.

Author's response:

Based on the new data presented (Figure 6l, see point 5 above), we repeated PRR851 toxicity using lower concentrations (2, 5, 10, and 30 μ M) and this was done on HeLa cells and CD4⁺T cells (see new Supplementary Figure 8a). The text has been rewritten accordingly:

"Up to 30 μ M, neither ICZ, PRR851, nor PRR846 had major effects on cell growth after 48 hours compared with DMSO. Only a \approx 10% reduction was observed, whereas H-ICZ caused \approx 60% inhibition (Supplementary Fig. 8a)."

Reviewer's comment:

13. It would be nice to see quantitative data for 4J. 1 cell is not compelling.

Author's response:

The sole purpose of this image is to show that another marker (Lamin B1), similar to SUN2, can highlight virus-induced nuclear deformation. We used SUN2 to quantify this phenomenon, which is blocked by the VOR complex inhibitor, PRR851 (see new Supplementary Figure 10b, c).

Reviewer's comment:

14. Line 243, please explain what the FRET data are that the data herein agree with.

Author's response:

We apologize for the lack of information, which is now introduced in the revised Results section: *"These observations are in agreement with our previous data using the fluorescence resonance energy transfer that showed a close contact of each protein pair (i.e. VAP-A/ORP3, ORP3/Rab7 and VAP-A/Rab7) at the nuclear membrane of NEIs¹⁹."*

Reviewer's comment:

15. Line 246 Figure 4L and S9A, please explain what nocodazole does, ie inhibits microtubules and thereby prevents virus reaching the nucleus. Readers may not get this without further detail.

Author's response:

Thank you for pointing out this issue. We have indicated this in the revised results section: *"An intact microtubule network, as demonstrated by the treatment with nocodazole – a microtubule de-polymerizing agent – prior and during the infection, is important for the interaction of VOR complex proteins (Fig. 6p). These data implicate microtubules in the retrograde transport of virus-laden late endosomes from cell periphery toward perinuclear area where formation of the VOR complex occurs (Fig. 6q, see also Discussion). It remains to be demonstrated whether microtubules could be the driving force for NEI formation."*

Reviewer's comment:

16. End of page 13 the text drifts away from explaining the experiments. Please explain better. For example, the gel shown (5H) doesn't show any VAP-A interaction. Ie explain 5I in the text.

Author's response:

We apologize for the lack of information. We have added the techniques used, i.e., immunoisolation of ORP3 and probing of the bound fraction by immunoblotting. Also, we explain in more detail the data presented in the original Figure 5H, I (now Figure 7h, i).

Reviewer's comment:

17. There is a literature suggesting that the HIV-1 Vpr protein causes nuclear invagination, or at least nuclear envelope disruption. Of course, Vpr is in the particle. There's also a recent hint that Vpr helps HIV infection of resting T cells. See PMID: 35417711. The authors might read those studies and tell us what they think (if they make any sense) in the discussion. This feels relevant but it may not be and they should decide.

Author's response:

We agree with the Reviewer about the potential relevance of the Vpr findings. We have now mentioned the relevant literature on Vpr in the Discussion section (see below):

"In addition to, or in synergy with, the VOR complex, possible contributory role for HIV-1 Vpr protein in NEI induction cannot be excluded, based on the finding that Vpr induced transient, localized herniations (probably NEI) in the nuclear envelope, associated with defects in the nuclear lamina⁷⁷. Interestingly, a recent study attributed to Vpr in HIV-1 virions a reprogramming role of resting T cells into tissue-resident memory T cells⁷⁸."

REVIEWER COMMENTS

Reviewer #1 (Remarks to the Author):

The authors have addressed my previous concerns. They added new experiments and, most importantly, provided a model that puts their findings in the context of current understanding of HIV nuclear import. Now, the proposed model does not look like an attempt to discard previously established mechanisms, but adds to these mechanisms identifying new players and pathways.

Reviewer #2 (Remarks to the Author):

I appreciate the author efforts to improve the manuscript, but I am still very conflicted with the images that are not convincing and that the paper is missing so much HIV-1 basic knowledge. Particularly the bold particles(HIV-1 like particles without VSV-G) should be seen everywhere(surface and endocytic pathway) and they do not represent real infection; therefore, it is really hard to believe that this is an infection assay since the real virus will behave the same. The authors do not show the bald virus in real cells anymore since I think this will invalidate their assay.

The image of viruses treated with dinosaur(DNA) are not convincing, one cannot appreciate the virus in the surface of the cell(very faint green stain on the surface(Figure 2A)).

The authors provided proof of protein depletion for VAP-A, VAP-B and ORF3. Although it seems that knocking out/down VAP-A and ORP-3 affects HIV-1 infection, the infections are not convincing, they need to use an HIV-1 reporter virus(the GFP). Also I think, if these factors are really required for infection(which is not shown here), this is a strength in the paper and should be documented properly using an HIV-1 reporter. In addition, viral staging should be perform, meaning the investigators should determine where the block is(reverse transcription by measuring viral cDNA over time, nuclear import by measuring formation of 2-LTR circles, integration by alu-PCR to measure integration sites etc) upon protein knock down or the use of different drugs such as ICZ.

Reviewer #3 (Remarks to the Author):

The reviewers have addressed my points questions with new data and explanation.

However, like reviewer 2, I remain concerned about the use of VSV-G pseudotyped virus and the reduced amount of data using bald viruses.

I therefore have two further points to make

1. The authors should show that a bald virus made without any envelope protein does not associate with NEI and particularly does not induce them. This is because bald viruses make an excellent control for things that happen independently of the envelope dependent infection event. This would be a really persuasive approach to show that the NEI induction is truly an infection dependent event. In the case that NEI are induced by bald virus, the conclusions of the paper can be changed to the observation that while NEI are induced, they are not induced by envelope dependent events and therefore may not be related to the infectious event, and might simply be induced by virus particles in endosomes.

2. Furthermore, the reviewers should use HIV that enters cells via a CD4 dependent event. That is, they should use some kind of HIV envelope-receptor dependent infection to show that NEI induction and VAP-A sensitivity is truly related to HIV infection and not something that is only true when a VSV-G envelope is used. I think that this is really important and will add hugely to making this work compelling. As it stands its very hard to be sure that the work is relevant to HIV because the VSV-G envelope is a very different, pH dependent endosomal route of entry that is somewhat different to the route normally used by HIV via CD4.

It is true that VSV-G is used for convenience in many studies but it seems particularly important to use an HIV envelope in this particular study because the results are so unexpected and aiming to change the model by which HIV enters and infects cells.

Ideally the authors should use replication competent wild type HIV-1 However, this isn't essential and delta Env HIV can be pseudotyped using an HIV-1 gp160 expression construct in the same way that the VSV-G construct is used. Of course, the target cells must make CD4 but there are many CD4 positive cell lines available for this experiment.

I think these 2 experiments are very important for making this study completely compelling.

POINT-BY-POINT RESPONSE TO THE REVIEWERS' COMMENTS

Reviewer #1

Reviewer's comment:

The authors have addressed my previous concerns. They added new experiments and, most importantly, provided a model that puts their findings in the context of current understanding of HIV nuclear import. Now, the proposed model does not look like an attempt to discard previously established mechanisms, but adds to these mechanisms identifying new players and pathways.

Author's response:

Again, we are delighted to read that the reviewer appreciated our work and noted that the proposed pathway and associated molecular players added new information to existing mechanisms.

Reviewer #2

Reviewer's comment:

I appreciate the author efforts to improve the manuscript, but I am still very conflicted with the images that are not convincing and that the paper is missing so much HIV-1 basic knowledge. Particularly the bold particles (HIV-1 like particles without VSV-G) should be seen everywhere (surface and endocytic pathway) and they do not represent real infection; therefore, it is really hard to believe that this is an infection assay since the real virus will behave the same. The authors do not show the bald virus in real cells anymore since I think this will invalidate their assay.

Author's response:

Following the advice of the reviewer (as well as the third reviewer), we have repeated our experiments with viral particles (HIV-89.6-EGFP) pseudotyped with native env produced by co-transfection with the plasmids p89.6 $\Delta E \Delta N$ SF-EGFP and HIV 89.6 Env. As cell targets, we used the CD4⁺ HeLa cell line or PHA/IL-2-activated primary CD4⁺ T cells as the "real" cells. The new data are presented in a new panel c of Supplementary Figure 4, in new Supplementary Figures 9, 10, and 12, and in the main text.

We observed that both HIV-1 IN and p24 capsid protein enter the endocytic pathway, as they co-localized with Rab7 in CD4⁺ HeLa cells (new Supplementary Figure 4c) and CD4⁺ T cells (Supplementary Figure 14a) and they are found in NEIs (new Supplementary Figures 9d, e and 12a). The latter process is blocked with PRR851 drug (Supplementary Figures 9d, e), as we previously demonstrated with VSV-G-pseudotyped virus (original Figure 5d, e).

Moreover, although viral particles pseudotyped with native env induced NEIs in primary T cells (original Figure 6i, j), the corresponding "bald" virus did not (new Supplementary Figure 12a, b). Indeed, "bald" virus showed very little HIV-1 IN inside the T cells after 3 hours infection and none inside the CD4⁺ HeLa cells after 1 hour infection compared to viral particles pseudotyped with native env. In both cases, the IN-2 signal remained at the cell periphery near the plasma membrane. As mentioned in the Methods section (Immunocytochemistry), it is important to note that all cells were washed with PBS before fixation, which removed the majority of free and unattached viral particles from the cell surface.

Reviewer's comment:

The image of viruses treated with dinosaur(DNA) are not convincing, one cannot appreciate the virus in the surface of the cell(very faint green stain on the surface (Figure 2A).

Author's response:

We have repeated the experiments involving the cell-permeable dynamin inhibitor dynasore, with HIV-89.6-EGFP pseudotyped with native env on CD4-negative and positive HeLa cells and quantified the amount of HIV-1 IN (IN-2) within the cells after treatment (see new Supplementary Figure 10a-f). Again, we observed the absence (or strong reduction) of HIV-1 IN (IN-2) inside the cells and its presence outside the cells. As noted above, all cells were washed with PBS before fixation, which removed the majority of unattached viral particles from the cell surface.

Reviewer's comment:

The authors provided proof of protein depletion for VAP-A, VAP-B and ORF3. Although it seems that knocking out/down VAP-A and ORP-3 affects HIV-1 infection, the infections are not convincing, they need to use an HIV-1 reporter virus (the GFP). Also I think, if these factors are really required for infection (which is not shown here), this is a strength in the paper and should be documented properly using an HIV-1 reporter. In addition, viral staging should be perform, meaning the investigators should determine where the block is(reverse transcription by measuring viral cDNA over time, nuclear import by measuring formation of 2-LTR circles, integration by alu-PCR to measure integration sites etc) upon protein knock down or the use of different drugs such as ICZ.

Author's response:

Here, we are puzzled by the comments. Both viruses used in this study have GFP (or EGFP) reporter protein. Indeed, the very weak GFP signal inside the virus particles can be followed after processing the micrographs, as shown in Supplementary Figures 1 and 2, where GFP⁺ virus particles are found in the NEIs.

We share the reviewer's eagerness to seek answers about the timing of capsid disassembly and reverse transcription (RT). In our previous revised version we added a paragraph to the Discussion to point out that the mechanism for nuclear entry that is identified and supported in this manuscript does not exclude current theories regarding the timing of RT. Clearly, the larger issue associated with when reverse transcription occurs relative to nuclear transport is more complex and it requires significant study beyond what can be done in this already very long revised manuscript. We trust that our study will spur further investigations on the timing of capsid disassembly and reverse transcription in relation to the newly discovered nuclear pathway and its genetic or pharmacological inhibition.

Reviewer #3**Reviewer's comment:**

The reviewers have addressed my points questions with new data and explanation.

Author's response:

We are pleased to read that the reviewer appreciated our efforts to address her/his initial key concerns.

Reviewer's comment:

However, like reviewer 2, I remain concerned about the use of VSV-G pseudotyped virus and the reduced amount of data using bald viruses.

Author's response:

As mentioned above, reviewer 2 raised valid concerns about the more extensive use of VSV-G pseudotyped viruses than those with native envelope components, particularly in CD4⁺ HeLa cells. These issues were addressed in the revised manuscript with additional experiments involving the bald viruses in other types of cells, including the use of the corresponding "bald" virus with CD4⁺ HeLa cells and PHA/IL-2-activated primary CD4⁺ T cells (see new Supplementary Figures 9, 10, and 12 and new panel c in Supplementary Figure 4).

Reviewer's comment:

I therefore have two further points to make:

1. The authors should show that a bald virus made without any envelope protein does not associate with NEI and particularly does not induce them. This is because bald viruses make an excellent control for things that happen independently of the envelope dependent infection event. This would be a really persuasive approach to show that the NEI induction is truly an infection dependent event. In the case that NEI are induced by bald virus, the conclusions of the paper can be changed to the observation that while NEI are induced, they are not induced by envelope dependent events and therefore may not be related to the infectious event, and might simply be induced by virus particles in endosomes.

Author's response:

In the first version of the manuscript, we demonstrated that "bald" virus did not induce type II NEI in HeLa cells (Supplementary Figure 3b). We have now shown in the revised manuscript that the incubation of primary PHA/IL-2 activated CD4⁺ T cells with "bald" virus also did not induce type II NEI (see new Supplementary Figure 12a, b). These data are consistent with the observation that only a small number of cells "infected" with the "bald" virus contain HIV-1 IN in their cytoplasm, and those that do have a very small amount (see new Supplementary Figure 12b, c).

Thus, induction of type II NEIs is directly related to cellular entry of virus particles, regardless of the nature of the env components (i.e. VSV-G or env gene products) present in the virus particles. As pointed out by the reviewer, loading of the endosomal compartment with viral particles is an essential step, as is the hyperphosphorylation of ORP3, which occurs in PHA/IL-2-activated T cells, leading to the formation of NEIs.

Reviewer's comment:

2. Furthermore, the reviewers should use HIV that enters cells via a CD4 dependent event. That is, they should use some kind of HIV envelope-receptor dependent infection to show that NEI induction and VAP-A sensitivity is truly related to HIV infection and not something that is only true when a VSV-G envelope is used. I think that this is really important and will add hugely to making this work compelling. As it stands its very hard to be sure that the work is relevant to HIV because the VSV-G envelope is a very different, pH dependent endosomal route of entry that is somewhat different to the route normally used by HIV via CD4.

Author's response:

In the revised manuscript, we compared the impact of CD4 on uptake of viral particles pseudotyped with native env (see below for plasmids), induction of NEIs and nuclear entry using native versus CD4⁺ HeLa cells as targets. The effect of dynasore was also evaluated. These new data are presented in new Supplementary Figures 9 and 10 and within the main text. They show that the absence of CD4 in the host cells severely impaired uptake, induction of NEIs and nuclear entry, indicating the importance of CD4. In all cases, PRR851 blocked HIV-1 IN and p24 nuclear entry via NEIs in both cell lines (Supplementary Figure 9a, d). Moreover, while dynasore completely blocks viral particle uptake in native (i.e. CD4-negative) HeLa cells, as was monitored with HIV-1 IN, a small fraction of HIV-1 IN nonetheless escapes dynasore interference in CD4⁺ HeLa cells, particularly when a higher concentration of virus is applied (Supplementary Figure 10a-d). These new data are also commented in the revised Discussion (lines 579-582).

The impact of virus particles pseudotyped with native env on the induction of type II NEIs was originally shown in primary PHA/IL-2 activated T cells (original Figure 6i, j), and now in CD4⁺ HeLa cells (see main text, lines 293-295). Again, PRR851 blocks the induction of NEIs (original Figure 6j, k) and the latter is corroborated by the production of infection as monitored by EGFP expression (original Figure 6l). As mentioned above, the corresponding "bald" virus did not induce NEIs in native cells such as PHA/IL-2 activated primary T cells (new Supplementary Figure 12a, b).

Reviewer's comment:

It is true that VSV-G is used for convenience in many studies but it seems particularly important to use an HIV envelope in this particular study because the results are so unexpected and aiming to change the model by which HIV enters and infects cells.

Author's response:

We agree with the reviewer, for these reasons we have repeated our experiments using native HIV envelope (and corresponding "bald" virus) with CD4-negative versus -positive HeLa cells (see new Supplementary Figures 9 and 10) or PHA/IL-2-activated primary CD4⁺ T cells (see new Supplementary Figure 12).

Note that the first 5 figures in the main manuscript are for VSV-G pseudotyped viruses, while the next 3 are for viral particles pseudotyped with native env. Now, we have also 5 Supplementary figures showing data with viral particles pseudotyped with native env.

Reviewer's comment:

Ideally the authors should use replication competent wild type HIV-1. However, this isn't essential and delta Env HIV can be pseudotyped using an HIV-1 gp160 expression construct in the same way that the VSV-G construct is used. Of course, the target cells must make CD4 but there are many CD4 positive cell lines available for this experiment. I think these 2 experiments are very important for making this study completely compelling.

Author's response:

We have used an HIV delta Env (i.e., p89.6 $\Delta E \Delta N$ SF-EGFP) and an autologous HIV 89.6 Env to generate viral particles. These plasmids were obtained from the NIH HIV Reagent Program. As target cells, we used CD4⁺ HeLa cells (original Figure 6a-c, new Supplementary Figures 9 and 10) and primary CD4⁺ T cells (original Figures 6d-p, 7 and 8; new Supplementary Figure 12). As mentioned above, we also used native (CD4 negative) HeLa cells and showed that viral particle uptake (as monitored by HIV-1 IN and p24), induction of NEIs and nuclear entry are severely impaired in the absence of CD4 in the host cells (new Supplementary Figure 9 and main text).

Reviewers' comments:

Reviewer #3 (Remarks to the Author):

I'm still bothered that the majority of the work characterising the NEI and effect of VAP-A and Orp3 on HIV infection is done in HeLa cells infected with VSV-G pseudotyped HIV-1 vector. I applaud the eventual use of HIV-1 with a bona fide gp160 Env and CD4 dependent entry but I think these experiments deserve to be highlighted in the study and not consigned to the supplementary data. The CD4 independence is problematic for me because infection is expected to be very much more dependent on CD4 than the data here suggest and so I don't know how the entry process is working, or whether its bona fide infection and therefore whether the experiments are really relevant to actual HIV infection or simply to situations where infection is dependent on pathways not normally used by HIV.

The T cell experiments rescue the situation somewhat.

I wonder if the HeLa are simply a terrible model for receptor dependent entry but I must say they are pretty well characterised cells and CD4 independent infection is peculiar.

1. For absolute clarity on the model being used in each figure please label all Figs with the cell and virus type eg VSV-G pseudotyped HIV-1 in HeLa. Gp160 pseudotyped HIV-1 in CD4 cells.

This is important because, as discussed in previous reviews, VSV-G has a well defined endosome dependent pH sensitive entry route whereas the HIV-1 env can fuse at the surface making the whole infection process less dependent on endosomes.

If the goal is to say that HIV is much more endosome dependent than we thought, and dependent on this new endosome dependent pathway of infection, involving VAP-A and Orp3, then the reader must be focused on the experiments using the CD4/gp160 dependent pathway. Its meaningless to characterise VSV-G mediated entry as relevant to HIV. So the key experiments should be labelled appropriately and should be in the main text, not in the supplementary data where nobody looks at them.

2. It is unexpected, to say the least, that HeLa cells can be "infected" with HIV pseudotyped with gp160 even when they don't express CD4. This makes me very nervous about the infection assay being used. Do the authors have any idea why this is. The notion that this is due to "other lectin-like cell surface factors" is unconvincing and these references are out of date and not relevant here. Attachment to cells does not equal infection and infection is expected to be fully receptor (CD4) dependent.

Sensitivity of this "background" to the inhibitor is also worrisome. Its not necessarily viral components being transferred into the cell here, it might just be GFP.

I'm reminded of reports of "pseudotransduction" in which free GFP in high concentrations in a viral prep can make all of the target cells go slightly green and read out as positive in a flow experiment. Typically only a few cells become green enough to cross the line in the flow scatter plot but this can lead to a background of envelope independent infection. This is evidenced by the cells just crossing the line and being a lot less bright than cells actually expressing GFP. Is this what is happening here? I must say flow plots in supp Fig 9a do not look like pseudotransduction so I'm not sure what's going on.

When it comes to the association of NEIs with viral infection the presence of CD4 only doubles the number of NEI from 18 to 33 (line 292). Tripling the titre by expressing CD4 and doubling the virus induced NEI with CD4 undermines the whole message of the paper because infection and any processes on which infection depends are expected to be way more dependent on receptor than a 2-3 fold increase on receptor expression.

These experiments are really important for the paper because they use an actual HIV envelope and therefore the relevant entry pathway and they must be compelling. Can the authors explain what's going on here.

The experiments with T cells simply use bald virus and this is more compelling. As I say above I wonder if the Hela CD4 are just a terrible model for this.

Line 563, I object to the statement "it remains to be determined whether viral internalization also involves nonspecific adhesion or aggregation of HIV-1 particles to the cell surface⁹⁸. The model that HIV infection is CD4 dependent is well established and reference 98, and others cited to explain the strange data are way out of date, this one 1998. Using cells in which infection is apparently not receptor dependent is not helping the case here.

Reviewer #3 (Remarks to the Author):

I'm still bothered that the majority of the work characterising the NEI and effect of VAP-A and Orp3 on HIV infection is done in HeLa cells infected with VSV-G pseudotyped HIV-1 vector. I applaud the eventual use of HIV-1 with a bona fide gp160 Env and CD4 dependent entry but I think these experiments deserve to be highlighted in the study and not consigned to the supplementary data. The CD4 independence is problematic for me because infection is expected to be very much more dependent on CD4 than the data here suggest and so I don't know how the entry process is working, or whether its bona fide infection and therefore whether the experiments are really relevant to actual HIV infection or simply to situations where infection is dependent on pathways not normally used by HIV. The T cell experiments rescue the situation somewhat.

Author's response

In agreement with the reviewer's suggestion, we have refocused the main manuscript on native CD4⁺ T cells using HIV-89.6-EGFP, which expresses a genuine env gp160. We have added all the necessary experiments using these cells to reinforce our findings (see new Figures 5e-f and 6l). We have also moved data originally presented in Supplementary Materials into the main text.

Data with VSV-G-pseudotyped HIV-1 are initially presented to demonstrate that membrane viruses known to enter cells by endocytosis follow the same intracellular route used by endocytosed extracellular membrane vesicles (EVs) to deliver their materials into the nuclear compartment of recipient cells. This is very important because the similarity between enveloped viruses and EVs and the pathways of their formation and action are of high interest nowadays.

We moved all data with respect to the heterologous cell system, i.e. CD4⁺ HeLa cells, including the new data described below (see new Supplementary Figures 1b, 10b, c and 12a-c)) and the corresponding text, into the Supplementary Materials because they are not essential to our conclusions, but nevertheless address the concern of the reviewer and further validate our data. They also showed that methods of infection (i.e. RetroNectin versus spinoculation) have an impact on productive infection on cells lacking CD4. Thus, the CD4 dependence in viral entry into HeLa cells is addressed in the revised manuscript in the light of new data (see below).

Reviewer's comment

I wonder if the HeLa are simply a terrible model for receptor dependent entry but I must say they are pretty well characterised cells and CD4 independent infection is peculiar. 1. For absolute clarity on the model being used in each figure please label all Figs with the cell and virus type eg VSV-G pseudotyped HIV-1 in HeLa. Gp160 pseudotyped HIV-1 in CD4 cells. This is important because, as discussed in previous reviews, VSV-G has a well-defined endosome dependent pH sensitive entry route whereas the HIV-1 env can fuse at the surface making the whole infection process less dependent on endosomes.

Author's response

We have labelled each figure and/or panel with the cells used and viruses applied. This makes our manuscript easier to understand. In addition, we discussed the entry of the virus into cells, particularly the one responsible for productive infection, in the light of our new data presented below on infection methods. Once again, CD4 dependence of the viral entry in HeLa cells is discussed below.

We agree with the reviewer that VSV-G-pseudotyped HIV-1 enters cells by endocytosis, which is why we used them to confirm that the novel nuclear pathway used by EVs is shared with membrane viruses to deliver their cargoes to the nucleoplasm. Regarding HIV-1 pseudotyped with native Env proteins, we do not exclude that they enter CD4⁺ cells (e.g., primary T cells and CD4-transfected HeLa cells) by direct fusion with the plasma membrane, but our data showed that HIV-1 integrase (IN) colocalized with Rab7⁺ late endosomes in a pole of the nucleus and in nuclear envelope invaginations (NEIs). A correlation between

NEI formation and productive infection was observed, with both events blocked by PRR851 drugs or in quiescent CD4⁺ T cells. Protein interactions of the VOR complex were also impaired in both conditions. These possibilities are described in the Discussion section:

“It should be noted that our data do not exclude that fusion of HIV-1 with the plasma membrane or early endosomes occurs. Whether HIV-1 capsids released at an early stage into the cytoplasmic compartment followed the same microtubule-dependent pathway as those endocytosed in late endosomes remains to be investigated. Distinct, but complementary, nuclear import pathways can coexist and collectively contribute to productive infection³². However, interference with translocation of late endosomes into the NR, or formation of type II NEIs, prevents productive infection as demonstrated in CD4⁺ T cells.”

Reviewer’s comment

If the goal is to say that HIV is much more endosome dependent than we thought, and dependent on this new endosome dependent pathway of infection, involving VAP-A and Orp3, then the reader must be focused on the experiments using the CD4/gp160 dependent pathway. Its meaningless to characterise VSV-G mediated entry as relevant to HIV. So, the key experiments should be labelled appropriately and should be in the main text, not in the supplementary data where nobody looks at them.

Author’s response

As indicated above, we have reorganized the manuscript accordingly, added new experiments with genuine HIV-1 Env protein and human CD4⁺ T cells as recipients and moved all these experiments into the main text. Figures 5 to 9 concern CD4⁺ T cells. Data relating to heterologous CD4⁺ HeLa cells have been moved into the Supplementary Materials as they are not essential to the understanding of our study.

For the pseudotyped HIV-1 VSV-G (Figures 1 to 4), as explained above, our intention is to highlight the features used by EVs to transport their contents into the nucleus of host cells that are shared with a virus that uses endocytosis to infect cells. We thus demonstrated that two distinct extracellular entities (EVs and membrane viruses) use the same machinery, namely the VOR complex, to deliver their contents into the nucleoplasmic reticulum, a mechanism inhibited by the new drugs. Indeed, this new nuclear pathway could be a general route used by distinct extracellular entities to deliver their contents to host cell nuclei, as explained in Discussion:

“Finally, the parallel nature of retroviruses and exosomes, based on shared physical and chemical characteristics and biogenesis pathways, is the rational for the Trojan exosome hypothesis: that retroviruses exploit cell-encoded pathways of vesicular traffic^{42, 44, 102}. Our data have demonstrated that nuclear transfer of extracellular materials associated with viruses (this study) and EVs^{34, 37, 38} after endocytosis has a shared mechanism, and that both increase the formation of type II NEIs (this study and Ref.³⁴), adding to the general knowledge about the action of extracellular entities on their target cells. Besides natural viruses and EVs, it has been proposed that exogenous lipid-protamine-DNA nanoparticles exploit tubular invaginations of nuclear envelope as entry-points towards the nucleoplasm¹⁰³. Thus, the inclusion of foreign materials in the type II NEI could be an alternative pathway to nuclear envelope breakdown for their nuclear transfer into non-dividing cells.”

Again, we labelled all the experiments in each figure to facilitate understanding.

Reviewer’s comment

2. It is unexpected, to say the least, that HeLa cells can be “infected” with HIV pseudotyped with gp160 even when they don’t express CD4. This makes me very nervous about the infection assay being used. Do the authors have any idea why this is.

Author’s response

We fully understand the importance of this topic and are aware that the productive infection in cells that do not have CD4 on their surface should be absent. The reason for the (low-level) EGFP positivity after infection of CD4-negative HeLa cell is due to the fact that we have used **RetroNectin**, a recombinant fibronectin fragment, to increase the level of infection. This technique was previously used to overcome the limitation associated with CD34⁺ hematopoietic stem/progenitor cell resistance to HIV infection in vitro (Tsukamoto T and Okada S, 2017 J Virol Methods 248;234). These CD34⁺ cells have a low level of CD4 expression. In another system, we have previously used this technique for the transduction of mouse neural stem cells (Rappa G et al. 2004 Neuroscience 124;8230).

To demonstrate that viral particles entry into CD4-negative cells is due to the technique used, i.e. RetroNectin, we compared RetroNectin-based infection with spinoculation, which concentrates viral particles on the cell surface through gentle centrifugation in the cold of the cells of interest with the viral particles.

As shown in Figure R1 (new Supplementary Figure 12a), repeated experiments with RetroNectin versus spinoculation showed that $6.24 \pm 3.94\%$ of infected CD4-negative HeLa cells expressed EGFP with the former technique, which significantly dropped to $3.12 \pm 2.35\%$ with the latter. The addition of the PRR851 drug, reduced the EGFP expression to $1.96 \pm 0.78\%$ when the RetroNectin was used, while no significant effect was observed with spinoculation, i.e. $1.28 \pm 0.98\%$, since the initial number of EGFP⁺ cells is already very low. In contrast, more than 20% of CD4-positive cells expressed EGFP, when spinoculation and RetroNectin were used. The addition of PRR851 has a significant impact on both. Thus, both techniques showed CD4 dependence for the productive infection, as did other studies, particularly when spinoculation was used.

Fig. R1. Efficacy of RetroNectin and spinoculation HIV-1 infection techniques. HeLa and CD4⁺ HeLa were pretreated with DMSO or 10 μM PRR851 for 30 minutes, then infected with HIV-89.6-EGFP (MOI = 2) using two different techniques: RetroNectin or spinoculation. In both cases, media were replaced prior to 48-hour incubation at

37°C in the presence of drugs. Trypsinized cells were analyzed by flow cytometry for EGFP expression. For the gating, the viral infection was omitted. Means \pm S.D. and individual values for each experiment (n = 9) are shown. Ns, not significant; *, $p < 0.05$; ***, $p < 0.001$.

This information is now described in the Supplementary Materials as it is not essential to the understanding of the study, but is nevertheless of interest as RetroNectin is used by some researchers to overcome CD4^{low} cell limitation.

Supplementary Materials:

“... This may be particularly true with the current method used to increase the level of infection. i.e. RetroNectin, which is based on a recombinant fibronectin fragment that binds to cells via its interaction with cell surface integrin receptors. Therefore, in the absence of CD4, a tiny fraction of the RetroNectin-virus complex present in the vicinity of the cell membrane can penetrate by other non-specific mechanisms of endocytosis, e.g., macropinocytosis, and thus produce a small, but present expression of the reporter gene EGFP. This technique has already been used to overcome limitations associated with the resistance of CD34⁺ hematopoietic stem/progenitor cells to HIV infection in vitro⁶. These CD34⁺ cells have a low level of CD4 expression. This prompted us to investigate this issue using an alternative method of infection, namely spinoculation⁷. This method concentrates the viral particles on the cell surface, among other notable biochemical/physical effects on the cells⁸, by a step of gentle centrifugation at 4°C of the cells of interest with the viral particles. This technique is widely used in the HIV field. Interestingly, infection of CD4⁺ HeLa cells with HIV-89.6-EGFP by spinoculation gave the same results as the RetroNectin technique, with the EGFP expression impacted by the presence of PRR851 (Supplementary Fig. 12a). In contrast, in spinoculation-infected native HeLa cells the number of EGFP⁺ cells was significantly reduced to $3.12 \pm 2.35\%$, compared with $6.24 \pm 3.94\%$ obtained with RetroNectin (Supplementary Fig. 12a). Because spinoculation yielded few EGFP⁺ cells, the addition of PRR851 did not significantly reduce their number compared with the control ($1.28 \pm 0.98\%$ vs. $3.12 \pm 2.35\%$, respectively), whereas this was the case when RetroNectin was used (Supplementary Fig. 12a). It remains to be determined whether and how the internalization of RetroNectin complex can promote the fusion of virus particles with the endosomal membrane in the absence of CD4, and hence, escape the degradation into the lysosomes. These phenomena did not occur with spinoculation”.

Furthermore, it is possible that the few CD4-negative HeLa cells expressing EGFP, notably those incubated with RetroNectin, could reflect the fact that this reporter gene in our construct is not under the control of the HIV-1-LTR promoter, but depends on the internal spleen focus-forming virus promoter. As noted by Carter and colleagues, we also observed a difference between the expression of EGFP and structural protein Gag, which was explained by the early expression of EGFP, i.e. before the expression of HIV-1- LTR-driven Nef, Vpu or Env (Carter CC et al. 2010, Nat Med 16; 446). Thus, Gag expression is more a direct measure of the productive infection as it is under the control of the HIV promoter.

Consequently, we have addressed this issue by monitoring the expression of Gag by flow cytometry using a monoclonal antibody against Gag proteins and compared its expression to EGFP. Again, we used both techniques of infection, i.e. RetroNectin versus spinoculation. As shown in Figure R2 ((new Supplementary Figure 12b and c), the number of double EGFP⁺Gag⁺ cells (i.e. $2.36 \pm 0.22\%$) dropped significantly compared to the overall EGFP expression ($6.85 \pm 0.31\%$) in RetroNectin-infected HeLa cells lacking CD4, indicating that productive infection of the gene under the HIV promoter is still occur, but at a very low level, with this technique (Figure R2a). This small amount of double-positive cells is nevertheless PRR851 drug sensitive since it is reduced to background level, i.e. less than 1%. As expected by the Reviewer, the double EGFP⁺Gag⁺ cells are almost absent/background level ($0.87 \pm 0.16\%$) in cells infected by spinoculation, and consequently not impacted by the drug. **Thus, the RetroNectin technique used to infect HeLa cells explains the presence, albeit very low, of productive infection in cells lacking CD4.** In

contrast, in CD4-expressing HeLa cells, both techniques showed PRR851-sensitive productive infection, as evidenced by double EGFP⁺Gag⁺ cells, validating the data presented in our manuscript.

Fig. R2. The productive infection as evaluated by double EGFP⁺Gag⁺ cells is absent in CD4-negative HeLa cells using spinoculation-based method of infection. (a, b) HeLa and CD4⁺ HeLa were pretreated with DMSO or 10 μ M PRR851 for 30 minutes then infected with HIV-89.6-EGFP (MOI = 2) using either RetroNectin (a) or spinoculation (b) technique. Media were replaced prior to 48-hour incubation at 37°C in the presence of drugs. Trypsinized cells were fixed, permeabilized, and immunolabeled for Gag. EGFP and Gag expression were then analyzed by flow cytometry. Total EGFP⁺ (or Gag⁺) cells versus double-positive cells are displayed. For the gating, the viral infection was omitted. The means \pm S.D. and individual values for each experiment (n = 3) are shown. Note that absence of EGFP⁺Gag⁺ cells in CD4-negative cells using spinoculation, while about 2% are detected with RetroNectin. *, $p < 0.05$; **, $p < 0.01$; ***, $p < 0.001$.

In the revised manuscript, we also monitored the expression of the Gag structural protein in primary HIV-1-infected T cells, in addition to the EGFP reporter, and demonstrated its expression over time and its dependence on the PRR851 drug (new Figures 5e and f). Spinoculation was also used with CD4⁺ T cells, confirming the data obtained with RetroNectin (new figure 5g).

Reviewer's comment

The notion that this is due to "other lectin-like cell surface factors" is unconvincing and these references are out of date and not relevant here. Attachment to cells does not equal infection and infection is expected to be fully receptor (CD4) dependent. Sensitivity of this "background" to the inhibitor is also worrisome. Its not necessarily viral components being transferred into the cell here, it might just be GFP.

Author's response

Our new experiments with CD4-negative HeLa cells described above (Figures R1 and R2) addressed these issues. Again, the EGFP background observed was due to the use of RetroNectin-based methods that indeed

stimulate the cell attachment. As highlighted by the Reviewer, taken alone, the expression of the *EGFP* reporter gene, which is not under the control of the HIV-1 promoter, is not a full indication of the productive infection by HIV-89.6-EGFP that expresses a *bona fide* gp160 env. Nonetheless, combined with Gag expression, we demonstrated that double positive cells are absent when spinoculation is applied, and very limited (i.e. $2.36 \pm 0.22\%$) with the RetroNectin method. The latter are drug-sensitive, indicating that the virus enters the cells, albeit in very limited quantities, in the absence of CD4 cells.

To explain this, it is important to note that the novel pathway we have described is based on endocytosis. As the reviewer knows, various mechanisms of endocytosis are described; clathrin-, lipid raft- and caveola-mediated endocytosis, as well as macropinocytosis and phagocytosis contributed to the internalization of external components. For these reasons, we have proposed that “other lectin-like cell surface factors” or “Besides VSV-G and env proteins, it remains to be determined whether viral internalization also involves nonspecific adhesion or aggregation of HIV-1 particles to the cell surface” could explain viral internalization in CD4-negative cells, even if it is very limited. Our new data comparing RetroNectin and spinoculation confirm this.

Indeed, to be effective, the recombinant fibronectin fragment (RetroNectin reagent) contains three functional domains: the cell-binding domain (C-domain), the heparin-binding domain (H-domain), and the CS-1 sequence. Virus particles bind to the H-domain, while target cells bind mainly through the interaction of cell surface integrin receptors VLA-5 and/or VLA-4 with the fibronectin C-domain and CS-1 sites, respectively. By enabling physical proximity, the RetroNectin reagent can enhance viral gene transfer to target cells expressing integrin receptors VLA-4 ($\alpha 4\beta 1$) and/or VLA-5 ($\alpha 5\beta 1$). HeLa cells strongly express $\alpha 5$ and $\beta 1$ integrins (Maginnis MS et al. 2006 J Virol 80;2760). Therefore, in the absence of CD4, a tiny fraction of the RetroNectin-virus complexes present in the vicinity of the cell membrane can penetrate by other mechanisms, e.g., macropinocytosis, and thus produce a small (about 2% of cells), but present productive infection. It remains to be determined whether the internalization of RetroNectin complex can promote the fusion of virus particles with the endosomal membrane in the absence of CD4, and hence, escape the degradation into the lysosomes. **These phenomena did not occur during spinoculation.**

Reviewer’s comment

I’m reminded of reports of “pseudotransduction” in which free GFP in high concentrations in a viral prep can make all of the target cells go slightly green and read out as positive in a flow experiment. Typically only a few cells become green enough to cross the line in the flow scatter plot but this can lead to a background of envelope independent infection. This is evidenced by the cells just crossing the line and being a lot less bright than cells actually expressing GFP. Is this is what is happening here? I must say flow plots in supp Fig 9a do not look like pseudotransduction so I’m not sure what’s going on.

Author’s response

We did not observe pseudotransduction. As with VSV-G pseudotyped HIV-1 (HIV-Gag-iGFP) (original Supplementary Figure S1), Gp160 pseudotyped HIV-1 particles (HIV-89.6-EGFP) are very faintly green (Figure R3b), leading to a weak background signal, which cannot explain the GFP⁺ cells observed in CD4-negative HeLa cells (see previous points). Nevertheless, GFP/EGFP signals in the viral particles can be artificially observed during image post-processing (see Figure R3, now Supplementary Figure 1).

Fig. R3. Gag-iGFP and EGFP signal in viral particles. a-c HIV-Gag-iGFP (a, a', top panel) and the corresponding “bald” virus, i.e. viral particles lacking the VSV-G envelope (a, a', bottom panel), HIV-89.6-EGFP (b) and CD9-GFP-positive EVs derived from CD9-GFP-expressing FEMX-I cells (c) were adhered to coverslips and observed by CLSM. Arrows point to very weak GFP and EGFP signals in HIV-Gag-iGFP (a) and HIV-89.6-EGFP (b) viral particles, respectively, especially compared to those of the CD9-GFP⁺ EVs (c). Gag-iGFP and EGFP signals could be observed after image post-processing using Fiji (↗, a', b'). Scale bars, 10 μm.

Reviewer's comment

When it comes to the association of NEIs with viral infection the presence of CD4 only doubles the number of NEI from 18 to 33 (line 292). Tripling the titre by expressing CD4 and doubling the virus induced NEI with CD4 undermines the whole message of the paper because infection and any processes on which infection depends are expected to be way more dependent on receptor than a 2-3 fold increase on receptor expression. These experiments are really important for the paper because they use an actual HIV envelope and therefore the relevant entry pathway and they must be compelling. Can the authors explain what's going on here.

Author's response

Firstly, it is important to note that native HeLa cells already harbour NEIs in the absence of infection (see Figure 1g and Supplementary Figure 3) in a similar proportion to HIV-1-infected cells containing *bona fide*

Env gp160, indicating that their number does not necessarily increase after infection, consistent with the absence of CD4. For clarity, we modified the text in the revised manuscript, see Supplementary Materials:

“The number of NEIs also increased in infected CD4⁺ HeLa cells (33.3 ± 4.2 NEIs per 50 cells, $n = 3$) compared to native HeLa cells (18.0 ± 3.0 NEIs per 50 cells), the latter being at the level of native condition, i.e. without infection (see Supplementary Fig. 3)”.

The limited amount of internalized viral components (IN-2 integrase or p24) may nevertheless reach the NEIs, probably the pre-existing NEIs, via the endosomal system. In a cell line (HeLa) that already has NEIs in the absence of viral exposure, it is difficult to parse the pre-existing from the newly formed NEIs. The situation becomes more obvious with activated CD4⁺ T cells, which are completely devoid of NEIs in the absence of infection (Figure 6b-e). HIV-1 induces NEIs, which are essential for productive infection in activated, but not in quiescent CD4⁺ T cells. The phosphorylation of ORP3 promoted by CD4⁺ T cell activation is a key step to allow VOR protein interaction, NEI formation and productive infection. For these reasons, in the revised manuscript, we focused on CD4⁺ T cells as mentioned above and moved all data with CD4⁺ HeLa cells in the Supplementary Materials.

Secondly, we cannot establish a direct link between cell internalization and nuclear transfer and/or NEI induction. This is again well illustrated with CD4⁺ T cells and HIV-1 containing a *bona fide* Env gp160 (Figure 6i and j). For example, doubling the amount of virus (MOI 2 to 4) did not increase the number of cells showing NEIs, although the relative amount of viral components (i.e. HIV-1 integrase) in the cytoplasm of infected cells increased (Supplementary Figure 14b and c). This is due to the translocation of virus-laden late endosomes to the MTOC where their interaction with the outer nuclear membrane via the VOR complex proteins will induce NEIs (Figure 7c). The polarization of late endosomes at the perinuclear region will not necessarily increase with increasing numbers of virus-laden late endosomes, as it occurs efficiently with small amounts of virus. In general, we observed only 1 (or 2) NEIs per cell (Supplementary Figure 3a) alongside the general deformation of the nuclear membrane observed after infection (Figure 6g and h). Although we haven't formally quantified the subnuclear localization of NEIs in CD4⁺ T cells, they seem to often appear close to the MTOC, where virus-laden late endosomes have accumulated (for example, see Figure 6k and l). In our previous study, we demonstrated that microtubules are essential for EV entry into NEIs (Santos et al., 2018 JBC 293;13834). Such information are summarized in the illustration presented in Figure 7c.

Reviewer's comment

The experiments with T cells simply use bald virus and this is more compelling. As I say above I wonder if the HeLa CD4 are just a terrible model for this. Line 563, I object to the statement “it remains to be determined whether viral internalization also involves nonspecific adhesion or aggregation of HIV-1 particles to the cell surface”98. The model that HIV infection is CD4 dependent is well established and reference 98, and others cited to explain the strange data are way out of date, this one 1998. Using cells in which infection is apparently not receptor dependent is not helping the case here.

Author's response

We are pleased to read that the reviewer appreciates the CD4⁺ T cell data and for these reasons we added additional data, including spinoculation-based infection and HIV-1-LTR-driven Gag expression on this more representative infection model. With regard to HeLa cells, we are not sure that this is a horrible model, although they contain NEIs endogenously, unlike CD4⁺ T cells, which made our conclusion more difficult to grasp. As mentioned above, the very low amount of productive infection (assessed by the presence of double EGFP⁺Gag⁺ cells) observed on CD4-negative HeLa cells is a consequence of the use of the RetroNectin complex to promote the internalization of viral particles, a phenomenon that is not observed with spinoculation. So, depending on the infection method, other cell surface proteins (e.g. integrins in the

case of RetroNectin) may contribute indirectly, albeit to a small extent, to the internalization of extracellular particles.

REVIEWERS' COMMENTS

Reviewer #3 (Remarks to the Author):

The authors have worked hard to reproduce their original findings made in HeLa cells in primary CD4+ T cells and have presented new figures in the main text.

I agree that this makes the manuscript a lot more compelling and I have no further comments.

REVIEWER COMMENTS

Reviewer #1 (Remarks to the Author):

Reviewer's comment:

The manuscript by Santos and colleagues reports their results obtained while investigating the entry of HIV into the nucleus of an infected cell. They demonstrate that nuclear entry of endocytosed HIV is associated with invaginations of the nuclear envelope (NEI), which involve a complex of VAP23A, hyperphosphorylated ORP3, and Rab7. They also show that drugs targeting the VAP23A-ORP3-Rab7 complex inhibit HIV nuclear entry and infection. This work follows on a series of studies by Melikyan's group, who suggested that HIV fuses with endosomes, rather than plasma membrane. This notion was confirmed in later studies, although not for all HIV-susceptible cell types. It appears to hold for CD4+ T cells, the major target of HIV, so results of the current study are highly relevant and important. Nevertheless, a number of key questions remain obscure, so the reader is left without a clear understanding of the process. The issues listed below need to be addressed experimentally, or at least discussed based on available evidence.

Author's response:

We are pleased to read that the reviewer appreciated our work and noted that it was important. We thank him/her for the constructive comments. We have experimentally addressed all of her/his concerns.

Reviewer's comment:

1. The authors show that NEI are induced by late endosomes carrying HIV. Late endosomes are commonly associated with the lysosomal degradation pathway. It is important to determine why late endosomes with HIV inside do not acidify, or if only a small proportion does not acidify what determines this protection.

Author's response:

We addressed this intriguing question using different markers associated with either late endosomes/multivesicular bodies or lysosomes. These data are presented in the new Figure 3. We now show that alongside Rab7, the exosomal marker CD63, typically associated with luminal vesicles in late endosomes/multivesicular bodies, is co-labeled with HIV-1 integrase (IN-2) in nuclear envelope invaginations (NEIs), whereas Lamp1, a lysosome marker, is excluded. Quantification of double-immunolabeled cells indicates that only Rab7⁺, but not Lamp1⁺ organelles, penetrate and/or induce type II NEIs. As indicated in the revised Discussion, it remains to be determined whether endocytosed viral particles impact the composition and/or maturation of late endosomes as well as their cytoplasmic transport into the NEI core. The mechanism by which a subset of late endosomes avoid acidification doesn't seem to be clear in the literature, assuming it is not simply based on probabilistic encounters. We hope that our findings will spur studies aimed at a deeper understanding of the decision-making process of late endosomes to acidify and its timing.

Reviewer's comment:

2. The time course is important to determine, as this has a big impact on the viral infection. Does virus-endosome fusion occur when endosomes dock to the MTOC? How is this timing regulated?

Author's response:

Our time course experiments (now shown in Figure 2) indicated that viral particles, as monitored by HIV-1 integrase (IN-2), move from early to late endosomes, where they accumulate at the MTOC. A fraction of them then end up in NEIs, as shown by colocalization of Rab7 and IN-2, suggesting that virus-endosome fusion occurs in the cytoplasmic core of NEIs, not at the MTOC. Of course, such fusion could also occur at the MTOC, as suggested by HeLa cells lacking VAP-A or ORP3, or those treated with itraconazole or PRR851, where accumulation of Rab7⁺ late endosomes containing IN-2 is restricted considering their absence in NEIs. In the latter cases, though, neither nuclear transfer of viral components nor productive infection was observed, suggesting that late endosome entry into the NEI and virus-endosome fusion therein are essential.

Reviewer's comment:

3. How do NEI help virus to pass through the nuclear pore? Are they just a by-product or a functional structure? ORP3 findings do not address this question, as the complex may be needed for nuclear pore function as well as for NEI generation.

Author's response:

First, NEIs are not just a byproduct, as their appearance is related to productive infection. In a new set of experiments, we correlated NEI formation and productive infection with the amount of virus applied (see new panel I in Figure 6). This issue is now introduced in the revised Results section:

“To evaluate the relationship between the formation of NEIs and productive infection, we infected PHA/IL-2-activated T cells with different doses of virus (MOI ranging from 0.2 to 4) and determined the percentage of cells with induced NEIs relative to the amount of EGFP⁺ cells. First, we observed a consistent increase in both events with increasing virus concentration (Fig. 6I). Second, regardless of the amount of virus applied, there was a tight correlation ($r = 0.88$) between the amount of cells showing NEIs and EGFP expression (Fig. 6I), confirming that NEI induction is a prerequisite for productive infection.”

Moreover, the relationship between NEIs and productive infection is also illustrated in infected quiescent T cells where an accumulation of virus-laden late endosomes at a nuclear pole is detected without producing NEIs and without generating productive infection (Figure 7a-g). In addition, in the revised version we employed Sotrastaurin, a protein kinase C (PKC) inhibitor, and found that it inhibits NEI formation, productive infection and, perhaps coincidentally, ORP3 hyperphosphorylation. Altogether, these observations suggest that NEIs are actively involved in the nuclear transfer of HIV-1 components. The exact mechanism remains to be dissected as well as the advantage of using NEIs for such process.

Second, we addressed these issues in the revised Discussion section. In particular, we pointed out that the cytoplasmic core of NEIs could create a protective environment for HIV-1 components where they could interact with host factors. Exclusion of Lamp1⁺ lysosomes from this core could also prevent fusion of late endosomes with them, and hence protect HIV-1 degradation. In addition, by penetrating deep into the nuclear compartment, often reaching the nucleolus, NEIs could facilitate HIV-1 transfer to specific nuclear sub-compartments. Finally, it is also possible that the extreme curvature of nuclear membrane in NEIs facilitates the passage of the intact ~60 nm HIV-1 capsid (Zila et al. 2021, PMID: 33571428), which is much larger than the previously proposed ~40 nm central channel diameter of the human nuclear pore

complexes. Of note, a recent publication by Bhargava et al. (2021, PMID: 34592156) has shown a relationship between nuclear membrane organization mediated by its structural components and HIV-1 infectivity has recently been demonstrated. Such information is now added in the revised Discussion.

Reviewer's comment:

4. Findings with the drugs targeting the VAP23A-ORP3-Rab7 complex are very important and require standard time-course analysis of the drugs' effect on viral replication, preferably with primary HIV isolates and CD4⁺ T cells.

Author's response:

We agree with the reviewer and this important issue was addressed using CD4⁺ T cells. The new data are presented in Figure 6f, g. In summary, we have shown that it is essential to add the drug (PRR851) before infection, not after, to prevent productive infection. In addition, we also investigated, using HeLa cells, the impact of PRR851 concentration on HIV-1 integrase accumulation in NEIs and productive infection (see new Supplementary Figure 7b).

Reviewer's comment:

5. Animal experiments are out of place here, as no HIV infection has been done in mice. Demonstration that mice do not lose weight is meaningless without knowing the effective dose, measuring PK and PD. Instead, the authors should do a more careful analysis of drug toxicity in primary T cells.

Author's response:

We have performed the MTS assay on CD4⁺ T cells as we did on HeLa cells. These data are presented in a new Supplementary Figure 8b. As suggested by the reviewer, we have removed the animal experiments.

Reviewer's comment:

6. Fig. 1: just one cell is shown at high resolution. Please show several cells at lower resolution.

Author's response:

First, it is important to mention that each NEI is located in a given section of the x-y plane of the nucleus, which varies from cell to cell. Therefore, the probability of having a cross-section of several cells with NEIs is very low. For these reasons, we have shown representative images throughout the manuscript. This issue is now introduced in the revised Results section:

“Of note, the entire nuclear compartment needs to be scanned because NEIs are restricted to certain optical x-y planes. Thus, IN-2 within the NEIs can appear at any z-level of the nucleus, a phenomenon that varies from cell to cell.”

Second, we have added as requested lower resolution images as well as all x-y optical sections through the nucleus and composite images of all optical sections. See the new Supplementary Figure 11.

Reviewer's comment:

7. There is no staining for endosomes, how do the authors know that endosomes are involved?

Author's response:

In addition to the use of RFP fusion proteins (Rab5 and Rab7), we have now provided immunostaining of Rab7 and its colocalization with HIV integrase (IN-2) in NEIs (see new Figure 3a), where cross-sections and longitudinal sections of NEIs are shown. Besides Rab7, we now show that CD63, a marker of intraluminal vesicles found in late endosomes/multivesicular bodies, also colocalizes with IN-2 in NEIs (Figure 3b), whereas LAMP1, a marker of lysosomes, does not (Figure 3c-e). In these experiments, NEIs were highlighted with VAP-A-GFP.

Reviewer's comment:

8. Fig. 5: there is no evidence that ORP3 hyperphosphorylation is essential. The only conclusion that can be made is that it associates with complex formation nuclear entry. To show necessity, mutagenesis experiments should be involved.

Author's response:

First, we and others have previously demonstrated that the phosphorylation of ORP3 is important for its interactions with VAP-A (Weber-Boyvat et al. 2015, PMID: 25447204; Santos et al. 2021, PMID: 34429859). For example, treatment of ORP3 with \square -phosphatase abolished its interactions with VAP-A and Rab7. The action of \square -phosphatase was demonstrated by a shift of ORP3 mobility on electrophoresis gel (Santos et al. 2021, PMID: 34429859), where the hyperphosphorylated form corresponds to the slower migrating band (Figure 5H, now Figure 7h). In the revised manuscript, we treated the activated T cells with \square -phosphatase and showed the same shift of the molecular weight of ORP3 (see new Figure 7j).

Second, given that ORP3 phosphorylation is very complex (i.e. multiple sites; see Weber-Boyvat et al. 2015, PMID: 25447204) and PKC has been suggested to be involved, among others (Lehto et al. 2008, PMID: 18270267), we incubated CD4⁺ T cells with Sotrastaurin, a PKC pan-inhibitor, and showed that such a treatment reduced the ORP3 expression and its phosphorylation (see new Figure 8). The latter phenomena mimic the situation observed in quiescent T cells. In both cases, quiescent T cells and those incubated with Sotrastaurin, neither NEI formation nor productive infection was observed. Even with these new data, we cannot exclude that other factors impacted by PKC inhibition are involved and therefore we have modified our statement that ORP3 hyperphosphorylation is essential (see our answers to other reviewers below).

Reviewer #2 (Remarks to the Author):**Reviewer's comment:**

Santos et al propose a very intriguing mechanism by which HIV-1 viruses in endosomal vesicles invaginate the nuclear envelope and enter the nuclear compartment. Although this is a very interesting hypothesis, the data presented here is not convincing and lack important experiments and controls. Imaging helps, but the authors need biochemical evidence, where all viruses are accounted, i.e., cellular fractionation into nuclear and cytosolic content for the same type of experiments. Isolation of these EVs by gradient-centrifugation etc. The authors showed that depletion of VAP-A and the use of ICS blocks HIV-1; however, the stage of the block is not studied (this should correlate with their model). MOIs are not provided. Overall, the data presented here does not support the conclusions.

Author's response:

We agree with the Reviewer's criticism and we have addressed each issue as answers to the specific points below raised by her or him. Throughout the manuscript, we used multiplicities of infection (MOI) of 2 unless otherwise noted. This is now indicated at the beginning of the revised results section as well as in the figure legends.

Reviewer's comment:**Major issues:**

Fig1A: bald virus should show IN in vesicles in the inside the cytosol since bald particles do not have VSV-G; however, they have all the other components(i.e. IN), and they are endocytosed. The bald infected control used by the author is not correct(after an hour, the authors should see vesicles and virus in the surface). Lack MOI information.

Author's response:

We thank the reviewer for raising this issue. We used "bald" virus as negative control because cell entry is impaired without VSV-G. Even though a small fraction of HIV-1 is endocytosed, we think it is still a valid control. We have now repeated our experiments and, as correctly anticipated by the reviewer, we show that most of HIV-1 integrase (IN-2) is located outside the cells, near their membrane (see new Figure 3g and Supplementary Figure 11c), as in cells treated with dynasore (Figure 2A, DNS). Moreover, as pointed out by the reviewer, we could observe a tiny fraction of IN-2 in the cytoplasmic compartment in few cells. We modified the text accordingly:

"Almost no IN-2 was observed in the cytoplasmic compartment of cells infected with "bald" virus for 1 hour (Fig. 3g, inset g1). In the latter case, an accumulation of IN-2 was observed outside the cells, similar to the DNS treatment. Only a minute fraction of IN-2 from "bald" virus was able to enter some cells in a non-specific manner (see below), which is consistent with other studies^{47, 48}." MOI is now indicated.

Reviewer's comment:

Fig1A and B: the authors need to show how many of these events they see (nuclear envelope invaginations), is this rare? What is the MOI. Statistics are important here. Showing images of single cells is not convincing.

Author's response:

We have now clarified the text (lines 117-124) to better show that we indeed quantified the number of NEIs (type I and II) per 100 HeLa cells (Figure 1g) and per single cell (Supplementary Figure 2A, now Supplementary Figure 3a), and the impact of viruses on them (Supplementary Figure 2B, now Supplementary Figure 3b). Thus, in HeLa cells, the findings suggest that NEIs are not rare events, but their number increases after infection. For CD4⁺ T cells, almost no NEI is observed in the absence of infection (Figure 4F, now Figure 6i). These data were also quantified.

In addition, we quantified the number of NEIs containing HIV-1 integrase (IN-2) after 1 hour of infection. These data were already presented in Figures 2B and 3E (now Figure 4b and Figure 5e).

As mentioned above, we used a MOI of 2 for these experiments unless indicated.

Reviewer's comment:

Fig1H: DNS should also accumulate particles in the surface, or near the surface, of the cell. Again showing one picture does not provide convincing evidence. One of the very well-known problems of imaging HIV-1 is that bald particles can be seen in the cytoplasm(in endocytic vesicles); Therefore the use of dynasore (DNS) should allow the investigator to see particles in the surface/near-the-surface of the cell.

Author's response:

We clarified the description of the data that showed the presence of VSV-G-pseudotyped viral particles (see IN-2 immunolabeling) outside the cells and near the surface after dynasore treatment (Fig. 1H, right panel, asterisks, now Figure 2a, right panel, green asterisk). This information was indicated in the legend of the corresponding figure, and we have now indicated such information in the main text (Results) of the revised manuscript. Fig. 1H was not related to the Bald virus, but we repeated the experiments with Bald virus (see new Figure 3g) as mentioned above, and we now show all x-y sections of a low power view in Supplementary Figure 11c. Therein, we observed that the vast majority of IN-2 is found outside the cells. Only few sporadic IN-2 signals appear inside few cells indicating that the non-specific uptake is minor under our experimental setting. We revised the Results section accordingly (see previous point).

Reviewer's comment:

Fig2A: the authors need a proof of knockdown expression for VAP-A, VAP-B and ORP3. This Figure needs an MOI.

Author's response:

In the revised manuscript, we have now added a new set of images showing the absence of IN-2 in NEIs of cells lacking VAP-A and ORP3, but not in scrambled control and shVAP-B. See new Figure 4a (top panels). In the original version of the manuscript, we demonstrated the silencing of VAP-A, VAP-B and ORP3 by immunoblotting and quantified their reduction (see original Supplementary Figure 4, now Supplementary Figure 5). Likewise, we showed that the lack of VAP-A and ORP3 impeded the productive infection (Figure 2E, now Figure 4e).

As indicated above, we have now indicated that we used a MOI of 2 for all experiments unless noted otherwise.

Reviewer's comment:

Fig. 2F: needs a curve of infection using different amounts of virus when depleting VAP-A, not only one point. In addition, the investigators need to figure out where the block is when depleting VAP-A (reverse transcription, nuclear import, integration etc..)

Author's response:

We performed the requested experiments using an MOI of 8 instead of 2, and the new data are described in the revised Results section as follows:

“Raising the MOI (e.g., 8 instead of 2) partially increased the number of GFP⁺ cells in VAP-A-deficient cells from $4.56 \pm 2.03\%$ to $8.07 \pm 1.71\%$, which is lower than the corresponding number in scrambled shRNA, $48.81 \pm 4.82\%$, but in line with the remaining amount of VAP-A (Supplementary Fig. 5a).”

The depletion of VAP-A results in a block of nuclear import of both HIV-1 (this study) and extracellular vesicles (Santos et al., 2021, PMID: 34429859) and corresponds to the effects of PRR851, that disrupts the VOR complex, preventing nuclear import. We trust that our study will spur further investigations on the larger issues of timing of capsid disassembly and reverse transcription in relation to the newly discovered nuclear pathway and its genetic or pharmacological inhibition. We added the following paragraph to the Discussion section to make clear the need for additional studies in this area:

“Further studies will be required to identify the site(s) of reverse transcription of the viral RNA genome into double-stranded DNA in relation to the VOR complex-mediated nuclear pathway. Because of the rarity of reverse transcription events in an infected cell, the sub-viral reverse transcription complex is currently poorly characterized^{28,86}. Our findings are consistent with reverse transcription occurring in intact capsids^{87, 88}, perhaps inside endosomes located outside and/or in NEIs. The possible link between reverse transcriptase activity and HIV-1 uncoating⁸⁹, would suggest, however that this event occurs after the release of HIV capsid in the core of NEIs and near NPCs^{30, 31}. Our model could also support the evidence that reverse transcription is completed after HIV-1 nuclear import inside the nucleus^{32, 33, 79}.”

Reviewer’s comment:

Fig3: the assay that quantifies SUN2+ NEI containing IN-2 should be validated biochemically by fractionating cells in to nuclear and cytosolic content, or isolation of these EVs by gradient-centrifugation etc. This assay is used all over the manuscript and is not clear to this reviewer how reliable it is since the bald viruses show no virus in the cell, which is unlikely.

Author’s response:

We agree with the reviewer. These biochemical experiments were performed and new data, which support the information in the original manuscript, are presented in new panel g of Figure 5 (see below). Again, both itraconazole and PRR851 blocked the nuclear transfer of HIV-1 integrase.

Figure 5g. Itraconazole and PRR851 block the nuclear transfer of HIV-1 integrase. Drug-treated HeLa cells as indicated were infected with HIV-Gag-iGFP for 3 hours, then fractionated into cytoplasmic and nuclear fractions and probed by immunoblotting for HIV IN. The 1/4 and total fractions were loaded for cytoplasmic and nuclear fractions, respectively. As controls, fractions were immuno-probed for GAPDH (cytoplasmic marker) and Lamin B1 (nuclear marker). Note the absence of IN-2 in nuclear fraction of ICZ and PRR851-treated cells.

The EVs used were characterized by nanoparticle tracking analysis using ZetaView (Particle Metrix GmbH, Meerbusch, Germany). Our EV preparations were also characterized by immunoblotting for the presence and absence of particular markers, three-dimensional direct stochastic optical reconstruction microscopy, and electron microscopy in our previous studies (Santos et al. 2021, PMID: 34429859; Rappa et al, 2013, PMID: 23318676) and this information was submitted to EV track (EV-TRACK, <https://evtrack.org/>; ID: EV210180, author: Santos). The details above are indicated in the revised Methods section.

As mentioned above, we repeated our experiments with Bald virus, and only sporadic and limited labeling of HIV-1 integrase (IN-2) was found in some cells, while the vast majority of IN-2 remained outside the cells and near their membrane (see new Figure 3g, g1 and Supplementary Figure 11c).

Reviewer's comment:

Fig 3: the authors show that ICZ blocks viral infection, they should provide the stage of the block (reverse transcription, nuclear import, integration etc..)

Author's response:

As monitored by the trafficking of HIV-1 integrase (IN-2), our data have shown that nuclear import is blocked, and consequently, all following steps, including the viral integration, are also blocked. The larger issue associated with when reverse transcription occurs relative to nuclear transport is more complex, as discussed above, and it requires significant study beyond what can be done in this already very long revised manuscript. We have addressed these issues with the appropriate references in the revised Discussion section (see answer to a previous comment).

As mentioned above, we trust that our study will spur further investigations on the timing of capsid disassembly and reverse transcription in relation to the newly discovered nuclear pathway and its genetic or pharmacological inhibition.

Reviewer's comment:

Fig4B: what is the stage at which HIV-1 is inhibited by these drugs(reverse transcription, nuclear import, uncoating, integration etc..)

Author's response:

We share the reviewer's eagerness to seek answers about the timing of capsid disassembly and reverse transcription (RT). As indicated above, a paragraph was added to the Discussion to point out that the mechanism for nuclear entry that is identified and supported in this manuscript does not exclude current theories regarding the timing of RT. Given the substantial work needed to shed light on this important and

unanswered question about the HIV life cycle and the size of the current manuscript, we feel that this question is a future direction for this work. See answer to the point above.

Reviewer's comment:

The paper is written with very little information in the text and figure legends. For a reviewer to assess the data, the authors need to provide important details, MOI, time of infection, how the quantification of IN+ is performed, etc. The paper is hard to read, please describe the experiments in the figure legends better.

Author's response:

We have significantly improved the writing of the manuscript by adding details about the experiments and the purposes of the experiments. We hope that our manuscript will not be too long, but we are willing to shorten it if necessary. We apologize for the lack of information. In particular, we have specified the MOI and time of infection for every experiment and described the quantification of HIV-1 integrase (IN-2) positivity in the Immunocytochemistry section of the Methods.

Reviewer's comment:

Line 53-59: references suggesting that nuclear import of the HIV-1 core precedes uncoating are missing.

Author's response:

We added the relevant references and discussed this issue in detail in the revised Discussion section, as mentioned above.

Reviewer #3 (Remarks to the Author):

Reviewer's comment:

Santos and colleagues have studied the role of the VOR complex comprising VAP-A, ORP3 and Rab7 in HIV infection. They provide data suggesting that it is an essential cofactor for efficient HIV infection and possibly a determinant of T cell infection that may contribute to explaining why activation of T cells is required to make them permissive, at least in vitro. It's an intriguing study and I enjoyed reading it. I can see that these authors have not worked previously in HIV and as such may not be familiar with the enormous body of literature relating to HIV traffic in cells via endosomes and the process of capsid nuclear entry. They also haven't discussed modern models of what dictates resting T-cell infection and I'm afraid that without incorporating, at least discussing, some of the key dogmas of HIV biology, this study is going to get a tough time in review and not be widely appreciated by the field.

Author's response:

We are delighted to hear that the reviewer has appreciated our effort. Her/his comments and suggestions were instrumental in improving our manuscript and discuss the current status of the vast literature on HIV-1 intracellular transport. For more details, see below.

Reviewer's comment:

I like it because many of the experiments are compelling and are clearly telling us something important that we didn't know about cofactor requirements for HIV infection. But at the moment the study is written

without taking into account some of the key HIV literature and in places the authors do not support the conclusions with data.

Author's response:

Again, we are pleased to read that the reviewer liked our study. Following her/his suggestions, we have significantly improved our manuscript by including new experiments, especially those related to dose-response relationships. We have revised the Introduction and Discussion sections in light of the current literature.

Reviewer's comment:

I'd like to suggest re-consideration of some key points.

Author's response:

We appreciate the reviewer's effort and her/his suggestions to improve our manuscript.

Reviewer's comment:

1. The HIV particles that actually infect cells are not thought to be in endosomes when they arrive at the nucleus. They are thought to be naked in the cytoplasm and able to interact with cytoplasmic and nuclear pore associated cofactors as a prerequisite to nuclear transport. As well as all the studies describing capsid binding cofactor dependence, there are the microscope studies eg (Zila et al Cell 2021), and also from Vinay Pathak, that confirm that HIV is not in endosomes at this point. Its important to understand that you can see HIV particles doing all kinds of things but the question has been, which ones are infectious, and what do they do that is different. The answer appears to be, arrive at the nuclear pore as a free capsid and interact directly with nuclear pores. In other words, if HIV-1 is located within endosomes that promote nuclear envelope invaginations for nuclear import, how do the authors explain interactions between cytoplasmic co-factors and capsid regulating infection and nuclear transport? How would these co-factors access the capsid within endosomes? The authors need to present a model in which their data stacks up with previous work.

Author's response:

Before interacting with the nuclear pores, the HIV-1 capsid will escape from the endosomal compartment upon fusion of the viral particles with the limiting membrane of the late endosomes. In the revised Discussion section, we have now pointed out these issues as well as the potential interaction of HIV-1 capsid with host proteins such as cyclophilin A and cleavage and polyadenylation specificity factor 6 and nuclear pore components, as follows:

“The endosome-based, NEI dependent pathway for HIV-1 infections suggests some hypotheses about the timing and location of interactions with cytoplasmic co-factors and components of the nuclear membrane. Nuclear import could be facilitated by the exit of viral components from the late endosomes, i.e. after HIV-1 fusion with the limiting membrane, and their concentration in the core of the NEIs, which could create a protective microenvironment and favor their interactions with cytosolic factors and NPC components (e.g., Nup358, Nup153) en route to nucleoplasm^{26, 63-65}. Likewise, the exclusion of Lamp1⁺ lysosomes from NEIs

would prevent late endosome-lysosome fusion. The segregation of HIV-1 particles and Lamp1⁺ lysosomes has been previously noted in CD4⁺ T cells⁶⁶. Determination of the binding of HIV-1 capsid to host proteins such as cyclophilin A and cleavage and polyadenylation specificity factor 6 in relation to the perinuclear region, particularly NEIs, will be of interest^{67, 68}. By penetrating deep into the nuclear compartment, often reaching the nucleolus^{34, 36}, NEIs may facilitate the transfer of HIV-1 to specific nuclear subcompartments⁶⁹⁻⁷². This is an important consideration because viral complexes can bypass intranuclear movement or diffusion to reach integration sites.

Our data do not contradict the fact that most of the endocytosed viruses end up in the lysosomal compartment, as observed by the colocalization of HIV-1 IN and Lamp1 in the cytoplasmic compartment, or that a fraction of them are released from endosomal compartment outside the NEIs, where interactions with host cell factors might determine the fate of viral particles and infectivity. Within the NEIs (Fig. 6q, right panel), it is also possible that the extreme curvature of nuclear membrane may facilitate the passage of the intact ~60 nm HIV-1 capsid²⁸, which is much larger than the previously proposed ~40 nm limit for the central channel diameter of the human NPCs^{73, 74}. These physical constraints could impact the structure and/or composition of NPCs in terms of nucleoporins, which is consistent with the heterogeneity of NPCs^{33, 75}.

To summarize a part of our findings, we described a model in Figure 6q (see below).

Figure 6q. Schematic representation of the induction of type II NEIs by virus-laden late endosomes, a process mediated by the interaction of VOR complex proteins, namely ONM-associated VAP-A, cytoplasmic ORP3 and late endosome-associated Rab7 (left panel). Release of viral components from late endosomes into the cytoplasmic core of induced NEIs at the vicinity of the nuclear pore would facilitate their transfer to the nucleoplasm (right panel). PRR851 inhibits the interaction of the VOR complex proteins, and hence the NEI formation.

Reviewer's comment:

2. I'm afraid the conclusion that infection of resting T cells is dictated by Orp3 phosphorylation is not supported by the data. There is no measurement of Orp3 phosphorylation. It is true they see 2 bands in the activated cells and one in the resting cells but we don't know this is phosphorylation, it could be another PTM. We also don't know that this is why the VAP complex interacts in activated but not resting cells on HIV infection or that this additional band, whatever it is, dictates HIV permissivity. To make these claims the authors need to provide some direct evidence, not indirect observation.

Author's response:

First, we performed a new experiment showing the shift of the upper band of ORP3 after γ -phosphatase treatment of detergent cell lysates prepared from PHA/IL-2 activated T cells (new Figure 7j). Thus, we were able to exclude other post-translational modifications responsible for this slower migratory ORP3 band. These data are in line with those we recently reported where the enzymatic dephosphorylation of ORP3 impeded its binding to VAP-A and Rab7 (Santos et al. 2021, PMID: 34429859).

Second, we are not claiming that the hyperphosphorylation of ORP3 is the direct cause of HIV permissivity; we simply observed a correlation between the status of ORP3 phosphorylation and the viral infection of CD4⁺ T cells. The latter is further supported by treatment of T cells with Sotrastaurin, a PKC pan-inhibitor, during their activation and infection, which prevents the formation of the VOR complex and HIV-1-induced NEIs. Productive infection is also inhibited under these conditions (see new Figure 8), as it is in HeLa cells lacking ORP3. We have now modified the Discussion accordingly, as follows:

“Thus, PKC-mediated hyperphosphorylation of ORP3 in PHA/IL-2-activated T cells could, among other factors, contribute to their permissiveness, while quiescent cells were refractory to productive infection. As PKC activity is required for T cell activation and proliferation⁵⁹⁻⁶¹, the fact that ORP3 status is similar between quiescent and Sotrastaurin-incubated cells during PHA/IL-2-induced activation could be more than coincidental. Little is known about ORP3 function(s), but an ORP3 knockout has been reported to impact the expansion of lymphoid progenitors and favor aneuploidy⁶². Further studies should investigate the impact of the absence of ORP3 on HIV-1 infection in vivo.”

Reviewer’s comment:

3. Also please bear in mind the previous literature. Many studies show viral DNA synthesis is poor in resting T cells, explaining poor infection. One study assigns this to SAMHD1 expression (PMID: 22972397). I think its OK to make these measurements and form hypothesis, but the current version feels like conclusion with very little evidence for the model and more or less complete disregard for an extensive literature.

Author’s response:

We have now been more careful to at least mention in the Discussion the vast HIV literature and substantial work required to fully integrate this new pathway into the existing models of HIV infection. We have now 104 publications in the Reference list. As mentioned above, we now provided additional data showing that the inhibition of PKC impedes the phosphorylation of ORP3, induction of nuclear membrane invagination and productive infection.

Reviewer’s comment:

4. I don’t like the title because I feel it trivialises the importance of the study’s findings. I appreciate that its true as stated but it doesn’t mention the new HIV cofactors discovered here, which I would expect to see in the title. Nuclear invaginations feel like a rather imprecise description and I imagine many things, including artefacts?, might induce nuclear invaginations. I would like to make the case that what the study has discovered is a really important role for the VOR complex in HIV nuclear entry and infection, which is important. I don’t want to insist on a title change but I expect that something like this would attract a greater

HIV readership. “The VOR complex regulates HIV nuclear entry and explains infection of activated T-cells” Of course, the authors should choose their title.

Author’s response:

We fully agree with the Reviewer’s suggestion. We have changed the title in: “HIV-1-induced nuclear invaginations mediated by VAP-A, ORP3, and Rab7 complex explain infection of activated T-cells: A novel targetable pathway”.

Reviewer’s comment:

5. I would really like to see a viral titration correlating the increase in NEI’s as titre increases with the level of infection. This would make this study much more compelling and really tightly link NEI induction to infection. I think that the percent infection where measured is slightly higher than the NEI count. Does this hold over a titration. Keen to see that, with some explanation/discussion, even if hypothetical.

Author’s response:

We agree with the reviewer and we have now experimentally addressed her/his comment using CD4⁺ T cells. The use of increasing amounts of HIV-89.6-EGFP MOI (0.2, 0.4, 1, 2, and 4) correlated with increasing numbers of type II NEIs and EGFP⁺ cells as a measure of productive infection. These new data are presented in Figure 6l. The text has been rewritten accordingly:

To evaluate the relationship between the formation of NEIs and productive infection, we infected PHA/IL-2-activated T cells with different doses of virus (MOI ranging from 0.2 to 4) and determined the percentage of cells with induced NEIs relative to the amount of EGFP⁺ cells. First, we observed a consistent increase in both events with increasing virus concentration (Fig. 6l). Second, regardless of the amount of virus applied, there was a tight correlation ($r = 0.88$) between the amount of cells showing NEIs and EGFP expression (Fig. 6l), adding further evidence that NEI induction is a prerequisite for productive infection.”

Reviewer’s comment:

6. Figure 2E, I’m not seeing puncta. Are there puncta? Can we see better images?

Author’s response:

We now indicated them, i.e. GFP signal at the plasma membrane, with green arrowheads (see new Figure 4e and Figure 5i). Such punctate and scattered Gag-GFP signals were previously observed by other investigators (Hübner et al. 2007, PMID: 17728233).

Reviewer’s comment:

7. Line 176 and throughout, I’ve never heard the word immunoisolation. I think, in the HIV field, we call this immunoprecipitation. If this is right, please change throughout. Its semantics but helpful for clarity.

Author’s response:

Our approach is based on immuno-affinity purification using super paramagnetic beads conjugated to protein G (Miltenyi Biotec). The immune complex is recovered using a column placed in the magnetic field of a separator (i.e. a magnet). Unlike conventional immunoprecipitation using Sepharose/agarose beads for

example, neither sedimentation nor precipitation (i.e. centrifugation) is required to isolate the protein of interest. For these reasons, we have used the term “immunoisolation”. To clarify this issue, we now indicated the paramagnetic bead-based system used in the revised Results section.

Reviewer’s comment:

8. Line 179 ORP3 is not obviously appearing as a double band here. Perhaps the authors can include a better example. This may be true but its not clear here. Resolve with a different gel density?

Author’s response:

Because we use the same blots to probe ORP3 with VAP-A or Rab7, we limit sample migration by gel electrophoresis. The double bands of ORP3 in HeLa cells are highlighted in the original Supplementary Figure 4 (now Supplementary Figure 5a) and the new Supplementary Figure 7a (see point #9, below). The hyperphosphorylation of the upper band was further demonstrated in PHA/IL-2-activated T cells by treating them with \square -phosphatase (see new panel j in Figure 7).

Reviewer’s comment:

9. Line 177, its not clear how the IC50s were calculated here. There is no titration. Please expand on how this was done and show data.

Author’s response:

This information is now added as a new paragraph in the revised Methods section. Moreover, we provided all data concerning this issue, i.e. immunoblot and linear regression approach used to estimate the IC50. These data are presented in a new Supplementary Figure 7a.

Supplementary Fig. 7. Impact of PRR851 concentration on Rab7 binding to ORP3, entry of HIV-1 IN in NEIs and productive infection.

See manuscript for the corresponding legend. Note the double bands of ORP3 (a).

Reviewer’s comment:

10. In figure 3, the drug concentrations used are very high and apparently way above what is required to inhibit HIV infection. Could the authors titrate the drug down and again, relate the level of infection to the number of NEI formed? Do they correlate over different levels of infection in a linear way. Such an experiment could really support the proposed mechanism. Without titration to change NEI count and infection we can’t easily quantitate how these 2 things relate quantitatively.

Author’s response:

To complement and strengthen the HeLa cell data presented in the original Figure 3 (now Figure 5), we evaluated the effects of lower concentrations of PRR851 on the presence of HIV-1 integrase in NEIs and

productive infection. These data are now presented in a new Supplementary Figure 7b. The text has been rewritten accordingly:

“The effects of PRR851 on the accumulation of HIV-1 IN in NEIs and productive infection were concentration-dependent (Supplementary Fig. 7b). IN-2⁺ NEIs and GFP expression also decreased in concert ($r = 0.91$) with increasing levels of PRR851.”

Reviewer’s comment:

11. Line 218, I don’t understand the connection between cells being stuck to the plate or not and the NEI infection connection. Does this relate to the fact that HeLa make NEI but they’re only seen on infection in T cells? Please explain.

Author’s response:

First, we have completely rewritten this section, better explaining the experiments performed, their rationale and the resulting data.

Second, we have expanded our ideas on the link between suspension growth versus adhesion and NEI formation. The text has been rewritten accordingly:

“Unlike adherent HeLa cells, CD4⁺ T cells grow in suspension, which has an impact on their morphology: the former are very spread out while the latter are rounded. These morphological alterations could influence the formation and/or induction of NEIs. This prompted us to re-examine the impact of viral infection on NEIs.”

Reviewer’s comment:

12. Line 223, this point would be much more compellingly made with titrations of virus or drug that allow NEI/infection to be related quantitatively over a range of infectiousness. I refer to point 5.

Author’s response:

Based on the new data presented (Figure 6I, see point 5 above), we repeated PRR851 toxicity using lower concentrations (2, 5, 10, and 30 μ M) and this was done on HeLa cells and CD4⁺T cells (see new Supplementary Figure 8a). The text has been rewritten accordingly:

“Up to 30 μ M, neither ICZ, PRR851, nor PRR846 had major effects on cell growth after 48 hours compared with DMSO. Only a \approx 10% reduction was observed, whereas H-ICZ caused \approx 60% inhibition (Supplementary Fig. 8a).”

Reviewer’s comment:

13. It would be nice to see quantitative data for 4J. 1 cell is not compelling.

Author’s response:

The sole purpose of this image is to show that another marker (Lamin B1), similar to SUN2, can highlight virus-induced nuclear deformation. We used SUN2 to quantify this phenomenon, which is blocked by the VOR complex inhibitor, PRR851 (see new Supplementary Figure 10b, c).

Reviewer's comment:

14. Line 243, please explain what the FRET data are that the data herein agree with.

Author's response:

We apologize for the lack of information, which is now introduced in the revised Results section:

“These observations are in agreement with our previous data using the fluorescence resonance energy transfer that showed a close contact of each protein pair (i.e. VAP-A/ORP3, ORP3/Rab7 and VAP-A/Rab7) at the nuclear membrane of NEIs¹⁹.”

Reviewer's comment:

15. Line 246 Figure 4L and S9A, please explain what nocodazole does, ie inhibits microtubules and thereby prevents virus reaching the nucleus. Readers may not get this without further detail.

Author's response:

Thank you for pointing out this issue. We have indicated this in the revised results section:

“An intact microtubule network, as demonstrated by the treatment with nocodazole – a microtubule depolymerizing agent – prior and during the infection, is important for the interaction of VOR complex proteins (Fig. 6p). These data implicate microtubules in the retrograde transport of virus-laden late endosomes from cell periphery toward perinuclear area where formation of the VOR complex occurs (Fig. 6q, see also Discussion). It remains to be demonstrated whether microtubules could be the driving force for NEI formation.”

Reviewer's comment:

16. End of page 13 the text drifts away from explaining the experiments. Please explain better. For example, the gel shown (5H) doesn't show any VAP-A interaction. Ie explain 5I in the text.

Author's response:

We apologize for the lack of information. We have added the techniques used, i.e., immunoisolation of ORP3 and probing of the bound fraction by immunoblotting. Also, we explain in more detail the data presented in the original Figure 5H, I (now Figure 7h, i).

Reviewer's comment:

17. There is a literature suggesting that the HIV-1 Vpr protein causes nuclear invagination, or at least nuclear envelope disruption. Of course, Vpr is in the particle. There's also a recent hint that Vpr helps HIV infection of resting T cells. See PMID: 35417711. The authors might read those studies and tell us what they think (if they make any sense) in the discussion. This feels relevant but it may not be and they should decide.

Author's response:

We agree with the Reviewer about the potential relevance of the Vpr findings. We have now mentioned the relevant literature on Vpr in the Discussion section (see below):

“In addition to, or in synergy with, the VOR complex, possible contributory role for HIV-1 Vpr protein in NEI induction cannot be excluded, based on the finding that Vpr induced transient, localized herniations (probably NEI) in the nuclear envelope, associated with defects in the nuclear lamina⁷⁷. Interestingly, a recent study attributed to Vpr in HIV-1 virions a reprogramming role of resting T cells into tissue-resident memory T cells⁷⁸.”

POINT-BY-POINT RESPONSE TO THE REVIEWERS' COMMENTS

Reviewer #1

Reviewer's comment:

The authors have addressed my previous concerns. They added new experiments and, most importantly, provided a model that puts their findings in the context of current understanding of HIV nuclear import. Now, the proposed model does not look like an attempt to discard previously established mechanisms, but adds to these mechanisms identifying new players and pathways.

Author's response:

Again, we are delighted to read that the reviewer appreciated our work and noted that the proposed pathway and associated molecular players added new information to existing mechanisms.

Reviewer #2

Reviewer's comment:

I appreciate the author efforts to improve the manuscript, but I am still very conflicted with the images that are not convincing and that the paper is missing so much HIV-1 basic knowledge. Particularly the bold particles (HIV-1 like particles without VSV-G) should be seen everywhere (surface and endocytic pathway) and they do not represent real infection; therefore, it is really hard to believe that this is an infection assay since the real virus will behave the same. The authors do not show the bald virus in real cells anymore since I think this will invalidate their assay.

Author's response:

Following the advice of the reviewer (as well as the third reviewer), we have repeated our experiments with viral particles (HIV-89.6-EGFP) pseudotyped with native env produced by co-transfection with the plasmids p89.6 ΔE ΔN SF-EGFP and HIV 89.6 Env. As cell targets, we used the CD4⁺ HeLa cell line or PHA/IL-2-activated primary CD4⁺ T cells as the "real" cells. The new data are presented in a new panel c of Supplementary Figure 4, in new Supplementary Figures 9, 10, and 12, and in the main text.

We observed that both HIV-1 IN and p24 capsid protein enter the endocytic pathway, as they co-localized with Rab7 in CD4⁺ HeLa cells (new Supplementary Figure 4c) and CD4⁺ T cells (Supplementary Figure 14a) and they are found in NEIs (new Supplementary Figures 9d, e and 12a). The latter process is blocked with PRR851 drug (Supplementary Figures 9d, e), as we previously demonstrated with VSV-G-pseudotyped virus (original Figure 5d, e).

Moreover, although viral particles pseudotyped with native env induced NEIs in primary T cells (original Figure 6i, j), the corresponding “bald” virus did not (new Supplementary Figure 12a, b). Indeed, “bald” virus showed very little HIV-1 IN inside the T cells after 3 hours infection and none inside the CD4⁺ HeLa cells after 1 hour infection compared to viral particles pseudotyped with native env. In both cases, the IN-2 signal remained at the cell periphery near the plasma membrane. As mentioned in the Methods section (Immunocytochemistry), it is important to note that all cells were washed with PBS before fixation, which removed the majority of free and unattached viral particles from the cell surface.

Reviewer’s comment:

The image of viruses treated with dinosaur(DNA) are not convincing, one cannot appreciate the virus in the surface of the cell(very faint green stain on the surface (Figure 2A)).

Author’s response:

We have repeated the experiments involving the cell-permeable dynamin inhibitor dynasore, with HIV-89.6-EGFP pseudotyped with native env on CD4-negative and positive HeLa cells and quantified the amount of HIV-1 IN (IN-2) within the cells after treatment (see new Supplementary Figure 10a-f). Again, we observed the absence (or strong reduction) of HIV-1 IN (IN-2) inside the cells and its presence outside the cells. As noted above, all cells were washed with PBS before fixation, which removed the majority of unattached viral particles from the cell surface.

Reviewer’s comment:

The authors provided proof of protein depletion for VAP-A, VAP-B and ORF3. Although it seems that knocking out/down VAP-A and ORP-3 affects HIV-1 infection, the infections are not convincing, they need to use an HIV-1 reporter virus (the GFP). Also I think, if these factors are really required for infection (which is not shown here), this is a strength in the paper and should be documented properly using an HIV-1 reporter. In addition, viral staging should be performed, meaning the investigators should determine where the block is(reverse transcription by measuring viral cDNA over time, nuclear import by measuring formation of 2-LTR circles, integration by alu-PCR to measure integration sites etc) upon protein knock down or the use of different drugs such as ICZ.

Author’s response:

Here, we are puzzled by the comments. Both viruses used in this study have GFP (or EGFP) reporter protein. Indeed, the very weak GFP signal inside the virus particles can be followed after processing the micrographs, as shown in Supplementary Figures 1 and 2, where GFP⁺ virus particles are found in the NEIs.

We share the reviewer’s eagerness to seek answers about the timing of capsid disassembly and reverse transcription (RT). In our previous revised version we added a paragraph to the Discussion to point out that the mechanism for nuclear entry that is identified and supported in this manuscript does not exclude current theories regarding the timing of RT. Clearly, the larger issue associated with when reverse transcription occurs relative to nuclear transport is more complex and it requires significant study beyond what can be done in this already very long revised manuscript. We trust that our study will spur

further investigations on the timing of capsid disassembly and reverse transcription in relation to the newly discovered nuclear pathway and its genetic or pharmacological inhibition.

Reviewer #3

Reviewer's comment:

The reviewers have addressed my points questions with new data and explanation.

Author's response:

We are pleased to read that the reviewer appreciated our efforts to address her/his initial key concerns.

Reviewer's comment:

However, like reviewer 2, I remain concerned about the use of VSV-G pseudotyped virus and the reduced amount of data using bald viruses.

Author's response:

As mentioned above, reviewer 2 raised valid concerns about the more extensive use of VSV-G pseudotyped viruses than those with native envelope components, particularly in CD4⁺ HeLa cells. These issues were addressed in the revised manuscript with additional experiments involving the bald viruses in other types of cells, including the use of the corresponding "bald" virus with CD4⁺ HeLa cells and PHA/IL-2-activated primary CD4⁺ T cells (see new Supplementary Figures 9, 10, and 12 and new panel c in Supplementary Figure 4).

Reviewer's comment:

I therefore have two further points to make:

1. The authors should show that a bald virus made without any envelope protein does not associate with NEI and particularly does not induce them. This is because bald viruses make an excellent control for things that happen independently of the envelope dependent infection event. This would be a really persuasive approach to show that the NEI induction is truly an infection dependent event. In the case that NEI are induced by bald virus, the conclusions of the paper can be changed to the observation that while NEI are induced, they are not induced by envelope dependent events and therefore may not be related to the infectious event, and might simply be induced by virus particles in endosomes.

Author's response:

In the first version of the manuscript, we demonstrated that "bald" virus did not induce type II NEI in HeLa cells (Supplementary Figure 3b). We have now shown in the revised manuscript that the incubation of primary PHA/IL-2 activated CD4⁺ T cells with "bald" virus also did not induce type II NEI (see new Supplementary Figure 12a, b). These data are consistent with the observation that only a small number of cells "infected" with the "bald" virus contain HIV-1 IN in their cytoplasm, and those that do have a very small amount (see new Supplementary Figure 12b, c).

Thus, induction of type II NEIs is directly related to cellular entry of virus particles, regardless of the nature of the env components (i.e. VSV-G or env gene products) present in the virus particles. As pointed out by

the reviewer, loading of the endosomal compartment with viral particles is an essential step, as is the hyperphosphorylation of ORP3, which occurs in PHA/IL-2-activated T cells, leading to the formation of NEIs.

Reviewer's comment:

2. Furthermore, the reviewers should use HIV that enters cells via a CD4 dependent event. That is, they should use some kind of HIV envelope-receptor dependent infection to show that NEI induction and VAP-A sensitivity is truly related to HIV infection and not something that is only true when a VSV-G envelope is used. I think that this is really important and will add hugely to making this work compelling. As it stands its very hard to be sure that the work is relevant to HIV because the VSV-G envelope is a very different, pH dependent endosomal route of entry that is somewhat different to the route normally used by HIV via CD4.

Author's response:

In the revised manuscript, we compared the impact of CD4 on uptake of viral particles pseudotyped with native env (see below for plasmids), induction of NEIs and nuclear entry using native versus CD4⁺ HeLa cells as targets. The effect of dynasore was also evaluated. These new data are presented in new Supplementary Figures 9 and 10 and within the main text. They show that the absence of CD4 in the host cells severely impaired uptake, induction of NEIs and nuclear entry, indicating the importance of CD4. In all cases, PRR851 blocked HIV-1 IN and p24 nuclear entry via NEIs in both cell lines (Supplementary Figure 9a, d). Moreover, while dynasore completely blocks viral particle uptake in native (i.e. CD4-negative) HeLa cells, as was monitored with HIV-1 IN, a small fraction of HIV-1 IN nonetheless escapes dynasore interference in CD4⁺ HeLa cells, particularly when a higher concentration of virus is applied (Supplementary Figure 10a-d). These new data are also commented in the revised Discussion (lines 579-582).

The impact of virus particles pseudotyped with native env on the induction of type II NEIs was originally shown in primary PHA/IL-2 activated T cells (original Figure 6i, j), and now in CD4⁺ HeLa cells (see main text, lines 293-295). Again, PRR851 blocks the induction of NEIs (original Figure 6j, k) and the latter is corroborated by the production of infection as monitored by EGFP expression (original Figure 6l). As mentioned above, the corresponding "bald" virus did not induce NEIs in native cells such as PHA/IL-2 activated primary T cells (new Supplementary Figure 12a, b).

Reviewer's comment:

It is true that VSV-G is used for convenience in many studies but it seems particularly important to use an HIV envelope in this particular study because the results are so unexpected and aiming to change the model by which HIV enters and infects cells.

Author's response:

We agree with the reviewer, for these reasons we have repeated our experiments using native HIV envelope (and corresponding "bald" virus) with CD4-negative versus -positive HeLa cells (see new Supplementary Figures 9 and 10) or PHA/IL-2-activated primary CD4⁺ T cells (see new Supplementary Figure 12).

Note that the first 5 figures in the main manuscript are for VSV-G pseudotyped viruses, while the next 3 are for viral particles pseudotyped with native env. Now, we have also 5 Supplementary figures showing data with viral particles pseudotyped with native env.

Reviewer's comment:

Ideally the authors should use replication competent wild type HIV-1. However, this isn't essential and delta Env HIV can be pseudotyped using an HIV-1 gp160 expression construct in the same way that the VSV-G construct is used. Of course, the target cells must make CD4 but there are many CD4 positive cell lines available for this experiment. I think these 2 experiments are very important for making this study completely compelling.

Author's response:

We have used an HIV delta Env (i.e., p89.6 ΔE ΔN SF-EGFP) and an autologous HIV 89.6 Env to generate viral particles. These plasmids were obtained from the NIH HIV Reagent Program. As target cells, we used CD4⁺ HeLa cells (original Figure 6a-c, new Supplementary Figures 9 and 10) and primary CD4⁺ T cells (original Figures 6d-p, 7 and 8; new Supplementary Figure 12). As mentioned above, we also used native (CD4 negative) HeLa cells and showed that viral particle uptake (as monitored by HIV-1 IN and p24), induction of NEIs and nuclear entry are severely impaired in the absence of CD4 in the host cells (new Supplementary Figure 9 and main text).

Reviewer #3 (Remarks to the Author):

I'm still bothered that the majority of the work characterising the NEI and effect of VAP-A and Orp3 on HIV infection is done in HeLa cells infected with VSV-G pseudotyped HIV-1 vector. I applaud the eventual use of HIV-1 with a bona fide gp160 Env and CD4 dependent entry but I think these experiments deserve to be highlighted in the study and not consigned to the supplementary data. The CD4 independence is problematic for me because infection is expected to be very much more dependent on CD4 than the data here suggest and so I don't know how the entry process is working, or whether its bona fide infection and therefore whether the experiments are really relevant to actual HIV infection or simply to situations where infection is dependent on pathways not normally used by HIV. The T cell experiments rescue the situation somewhat.

Author's response

In agreement with the reviewer's suggestion, we have refocused the main manuscript on native CD4⁺ T cells using HIV-89.6-EGFP, which expresses a genuine env gp160. We have added all the necessary experiments using these cells to reinforce our findings (see new Figures 5e-f and 6l). We have also moved data originally presented in Supplementary Materials into the main text.

Data with VSV-G-pseudotyped HIV-1 are initially presented to demonstrate that membrane viruses known to enter cells by endocytosis follow the same intracellular route used by endocytosed extracellular membrane vesicles (EVs) to deliver their materials into the nuclear compartment of recipient cells. This is very important because the similarity between enveloped viruses and EVs and the pathways of their formation and action are of high interest nowadays.

We moved all data with respect to the heterologous cell system, i.e. CD4⁺ HeLa cells, including the new data described below (see new Supplementary Figures 1b, 10b, c and 12a-c)) and the corresponding text, into the Supplementary Materials because they are not essential to our conclusions, but nevertheless address

the concern of the reviewer and further validate our data. They also showed that methods of infection (i.e. RetroNectin versus spinoculation) have an impact on productive infection on cells lacking CD4. Thus, the CD4 dependence in viral entry into HeLa cells is addressed in the revised manuscript in the light of new data (see below).

Reviewer's comment

I wonder if the HeLa are simply a terrible model for receptor dependent entry but I must say they are pretty well characterised cells and CD4 independent infection is peculiar. 1. For absolute clarity on the model being used in each figure please label all Figs with the cell and virus type eg VSV-G pseudotyped HIV-1 in HeLa. Gp160 pseudotyped HIV-1 in CD4 cells. This is important because, as discussed in previous reviews, VSV-G has a well-defined endosome dependent pH sensitive entry route whereas the HIV-1 env can fuse at the surface making the whole infection process less dependent on endosomes.

Author's response

We have labelled each figure and/or panel with the cells used and viruses applied. This make our manuscript easier to understand. In addition, we discussed the entry of the virus into cells, particularly the one responsible for productive infection, in the light of our new data presented below on infection methods. Once again, CD4 dependence of the viral entry in HeLa cells is discussed below.

We agree with the reviewer that VSV-G-pseudotyped HIV-1 enters cells by endocytosis, which is why we used them to confirm that the novel nuclear pathway used by EVs is shared with membrane viruses to deliver their cargoes to the nucleoplasm. Regarding HIV-1 pseudotyped with native Env proteins, we do not exclude that they enter CD4⁺ cells (e.g., primary T cells and CD4-transfected HeLa cells) by direct fusion with the plasma membrane, but our data showed that HIV-1 integrase (IN) colocalized with Rab7⁺ late endosomes in a pole of the nucleus and in nuclear envelope invaginations (NEIs). A correlation between NEI formation and productive infection was observed, with both events blocked by PRR851 drugs or in quiescent CD4⁺ T cells. Protein interactions of the VOR complex were also impaired in both conditions. These possibilities are described in the Discussion section:

“It should be noted that our data do not exclude that fusion of HIV-1 with the plasma membrane or early endosomes occurs. Whether HIV-1 capsids released at an early stage into the cytoplasmic compartment followed the same microtubule-dependent pathway as those endocytosed in late endosomes remains to be investigated. Distinct, but complementary, nuclear import pathways can coexist and collectively contribute to productive infection³². However, interference with translocation of late endosomes into the NR, or formation of type II NEIs, prevents productive infection as demonstrated in CD4⁺ T cells.”

Reviewer's comment

If the goal is to say that HIV is much more endosome dependent than we thought, and dependent on this new endosome dependent pathway of infection, involving VAP-A and Orp3, then the reader must be focused on the experiments using the CD4/gp160 dependent pathway. Its meaningless to characterise VSV-G mediated entry as relevant to HIV. So, the key experiments should be labelled appropriately and should be in the main text, not in the supplementary data where nobody looks at them.

Author's response

As indicated above, we have reorganized the manuscript accordingly, added new experiments with genuine HIV-1 Env protein and human CD4⁺ T cells as recipients and moved all these experiments into the main

text. Figures 5 to 9 concern CD4⁺ T cells. Data relating to heterologous CD4⁺ HeLa cells have been moved into the Supplementary Materials as they are not essential to the understanding of our study.

For the pseudotyped HIV-1 VSV-G (Figures 1 to 4), as explained above, our intention is to highlight the features used by EVs to transport their contents into the nucleus of host cells that are shared with a virus that uses endocytosis to infect cells. We thus demonstrated that two distinct extracellular entities (EVs and membrane viruses) use the same machinery, namely the VOR complex, to deliver their contents into the nucleoplasmic reticulum, a mechanism inhibited by the new drugs. Indeed, this new nuclear pathway could be a general route used by distinct extracellular entities to deliver their contents to host cell nuclei, as explained in Discussion:

“Finally, the parallel nature of retroviruses and exosomes, based on shared physical and chemical characteristics and biogenesis pathways, is the rationale for the Trojan exosome hypothesis: that retroviruses exploit cell-encoded pathways of vesicular traffic^{42, 44, 102}. Our data have demonstrated that nuclear transfer of extracellular materials associated with viruses (this study) and EVs^{34, 37, 38} after endocytosis has a shared mechanism, and that both increase the formation of type II NEIs (this study and Ref.³⁴), adding to the general knowledge about the action of extracellular entities on their target cells. Besides natural viruses and EVs, it has been proposed that exogenous lipid-protamine-DNA nanoparticles exploit tubular invaginations of nuclear envelope as entry-points towards the nucleoplasm¹⁰³. Thus, the inclusion of foreign materials in the type II NEI could be an alternative pathway to nuclear envelope breakdown for their nuclear transfer into non-dividing cells.”

Again, we labelled all the experiments in each figure to facilitate understanding.

Reviewer’s comment

2. It is unexpected, to say the least, that HeLa cells can be “infected” with HIV pseudotyped with gp160 even when they don’t express CD4. This makes me very nervous about the infection assay being used. Do the authors have any idea why this is.

Author’s response

We fully understand the importance of this topic and are aware that the productive infection in cells that do not have CD4 on their surface should be absent. The reason for the (low-level) EGFP positivity after infection of CD4-negative HeLa cell is due to the fact that we have used **RetroNectin**, a recombinant fibronectin fragment, to increase the level of infection. This technique was previously used to overcome the limitation associated with CD34⁺ hematopoietic stem/progenitor cell resistance to HIV infection in vitro (Tsukamoto T and Okada S, 2017 J Virol Methods 248;234). These CD34⁺ cells have a low level of CD4 expression. In another system, we have previously used this technique for the transduction of mouse neural stem cells (Rappa G et al. 2004 Neuroscience 124;8230).

To demonstrate that viral particles entry into CD4-negative cells is due to the technique used, i.e. RetroNectin, we compared RetroNectin-based infection with spinoculation, which concentrates viral

particles on the cell surface through gentle centrifugation in the cold of the cells of interest with the viral particles.

As shown in Figure R1 (new Supplementary Figure 12a), repeated experiments with RetroNectin versus spinoculation showed that $6.24 \pm 3.94\%$ of infected CD4-negative HeLa cells expressed EGFP with the former technique, which significantly dropped to $3.12 \pm 2.35\%$ with the latter. The addition of the PRR851 drug, reduced the EGFP expression to $1.96 \pm 0.78\%$ when the RetroNectin was used, while no significant effect was observed with spinoculation, i.e. $1.28 \pm 0.98\%$, since the initial number of EGFP⁺ cells is already very low. In contrast, more than 20% of CD4-positive cells expressed EGFP, when spinoculation and RetroNectin were used. The addition of PRR851 has a significant impact on both. Thus, both techniques showed CD4 dependence for the productive infection, as did other studies, particularly when spinoculation was used.

Fig. R1. Efficacy of RetroNectin and spinoculation HIV-1 infection techniques. HeLa and CD4⁺ HeLa were pretreated with DMSO or 10 μ M PRR851 for 30 minutes, then infected with HIV-89.6-EGFP (MOI = 2) using two different techniques: RetroNectin or spinoculation. In both cases, media were replaced prior to 48-hour incubation at 37°C in the presence of drugs. Trypsinized cells were analyzed by flow cytometry for EGFP expression. For the gating, the viral infection was omitted. Means \pm S.D. and individual values for each experiment (n = 9) are shown. Ns, not significant; *, $p < 0.05$; ***, $p < 0.001$.

This information is now described in the Supplementary Materials as it is not essential to the understanding of the study, but is nevertheless of interest as RetroNectin is used by some researchers to overcome CD4^{low} cell limitation.

Supplementary Materials:

“... This may be particularly true with the current method used to increase the level of infection. i.e. RetroNectin, which is based on a recombinant fibronectin fragment that binds to cells via its interaction

with cell surface integrin receptors. Therefore, in the absence of CD4, a tiny fraction of the RetroNectin-virus complex present in the vicinity of the cell membrane can penetrate by other non-specific mechanisms of endocytosis, e.g., macropinocytosis, and thus produce a small, but present expression of the reporter gene EGFP. This technique has already been used to overcome limitations associated with the resistance of CD34⁺ hematopoietic stem/progenitor cells to HIV infection in vitro⁶. These CD34⁺ cells have a low level of CD4 expression. This prompted us to investigate this issue using an alternative method of infection, namely spinoculation⁷. This method concentrates the viral particles on the cell surface, among other notable biochemical/physical effects on the cells⁸, by a step of gentle centrifugation at 4°C of the cells of interest with the viral particles. This technique is widely used in the HIV field. Interestingly, infection of CD4⁺ HeLa cells with HIV-89.6-EGFP by spinoculation gave the same results as the RetroNectin technique, with the EGFP expression impacted by the presence of PRR851 (Supplementary Fig. 12a). In contrast, in spinoculation-infected native HeLa cells the number of EGFP⁺ cells was significantly reduced to $3.12 \pm 2.35\%$, compared with $6.24 \pm 3.94\%$ obtained with RetroNectin (Supplementary Fig. 12a). Because spinoculation yielded few EGFP⁺ cells, the addition of PRR851 did not significantly reduce their number compared with the control ($1.28 \pm 0.98\%$ vs. $3.12 \pm 2.35\%$, respectively), whereas this was the case when RetroNectin was used (Supplementary Fig. 12a). It remains to be determined whether and how the internalization of RetroNectin complex can promote the fusion of virus particles with the endosomal membrane in the absence of CD4, and hence, escape the degradation into the lysosomes. These phenomena did not occur with spinoculation”.

Furthermore, it is possible that the few CD4-negative HeLa cells expressing EGFP, notably those incubated with RetroNectin, could reflect the fact that this reporter gene in our construct is not under the control of the HIV-1-LTR promoter, but depends on the internal spleen focus-forming virus promoter. As noted by Carter and colleagues, we also observed a difference between the expression of EGFP and structural protein Gag, which was explained by the early expression of EGFP, i.e. before the expression of HIV-1- LTR-driven Nef, Vpu or Env (Carter CC et al. 2010, Nat Med 16; 446). Thus, Gag expression is more a direct measure of the productive infection as it is under the control of the HIV promoter.

Consequently, we have addressed this issue by monitoring the expression of Gag by flow cytometry using a monoclonal antibody against Gag proteins and compared its expression to EGFP. Again, we used both techniques of infection, i.e. RetroNectin versus spinoculation. As shown in Figure R2 ((new Supplementary Figure 12b and c), the number of double EGFP⁺Gag⁺ cells (i.e. $2.36 \pm 0.22\%$) dropped significantly compared to the overall EGFP expression ($6.85 \pm 0.31\%$) in RetroNectin-infected HeLa cells lacking CD4, indicating that productive infection of the gene under the HIV promoter is still occur, but at a very low level, with this technique (Figure R2a). This small amount of double-positive cells is nevertheless PRR851 drug sensitive since it is reduced to background level, i.e. less than 1%. As expected by the Reviewer, the double EGFP⁺Gag⁺ cells are almost absent/background level ($0.87 \pm 0.16\%$) in cells infected by spinoculation, and consequently not impacted by the drug. **Thus, the RetroNectin technique used to infect HeLa cells explains the presence, albeit very low, of productive infection in cells lacking CD4.** In contrast, in CD4-expressing HeLa cells, both techniques showed PRR851-sensitive productive infection, as evidenced by double EGFP⁺Gag⁺ cells, validating the data presented in our manuscript.

Fig. R2. The productive infection as evaluated by double EGFP⁺Gag⁺ cells is absent in CD4-negative HeLa cells using spinoculation-based method of infection. (a, b) HeLa and CD4⁺ HeLa were pretreated with DMSO or 10 μ M PRR851 for 30 minutes then infected with HIV-89.6-EGFP (MOI = 2) using either RetroNectin (a) or spinoculation (b) technique. Media were replaced prior to 48-hour incubation at 37°C in the presence of drugs. Trypsinized cells were fixed, permeabilized, and immunolabeled for Gag. EGFP and Gag expression were then analyzed by flow cytometry. Total EGFP⁺ (or Gag⁺) cells versus double-positive cells are displayed. For the gating, the viral infection was omitted. The means \pm S.D. and individual values for each experiment (n = 3) are shown. Note that absence of EGFP⁺Gag⁺ cells in CD4-negative cells using spinoculation, while about 2% are detected with RetroNectin. *, $p < 0.05$; **, $p < 0.01$; ***, $p < 0.001$.

In the revised manuscript, we also monitored the expression of the Gag structural protein in primary HIV-1-infected T cells, in addition to the EGFP reporter, and demonstrated its expression over time and its dependence on the PRR851 drug (new Figures 5e and f). Spinoculation was also used with CD4⁺ T cells, confirming the data obtained with RetroNectin (new figure 5g).

Reviewer's comment

The notion that this is due to "other lectin-like cell surface factors" is unconvincing and these references are out of date and not relevant here. Attachment to cells does not equal infection and infection is expected to be fully receptor (CD4) dependent. Sensitivity of this "background" to the inhibitor is also worrisome. Its not necessarily viral components being transferred into the cell here, it might just be GFP.

Author's response

Our new experiments with CD4-negative HeLa cells described above (Figures R1 and R2) addressed these issues. Again, the EGFP background observed was due to the use of RetroNectin-based methods that indeed stimulate the cell attachment. As highlighted by the Reviewer, taken alone, the expression of the *EGFP* reporter gene, which is not under the control of the HIV-1 promoter, is not a full indication of the productive infection by HIV-89.6-EGFP that expresses a *bona fide* gp160 env. Nonetheless, combined with Gag expression, we demonstrated that double positive cells are absent when spinoculation is applied, and very limited (i.e. $2.36 \pm 0.22\%$) with the RetroNectin method. The latter are drug-sensitive, indicating that the virus enters the cells, albeit in very limited quantities, in the absence of CD4 cells.

To explain this, it is important to note that the novel pathway we have described is based on endocytosis. As the reviewer knows, various mechanisms of endocytosis are described; clathrin-, lipid raft- and caveola-mediated endocytosis, as well as macropinocytosis and phagocytosis contributed to the internalization of external components. For these reasons, we have proposed that “*other lectin-like cell surface factors*” or “*Besides VSV-G and env proteins, it remains to be determined whether viral internalization also involves nonspecific adhesion or aggregation of HIV-1 particles to the cell surface*” could explain viral internalization in CD4-negative cells, even if it is very limited. Our new data comparing RetroNectin and spinoculation confirm this.

Indeed, to be effective, the recombinant fibronectin fragment (RetroNectin reagent) contains three functional domains: the cell-binding domain (C-domain), the heparin-binding domain (H-domain), and the CS-1 sequence. Virus particles bind to the H-domain, while target cells bind mainly through the interaction of cell surface integrin receptors VLA-5 and/or VLA-4 with the fibronectin C-domain and CS-1 sites, respectively. By enabling physical proximity, the RetroNectin reagent can enhance viral gene transfer to target cells expressing integrin receptors VLA-4 ($\alpha 4\beta 1$) and/or VLA-5 ($\alpha 5\beta 1$). HeLa cells strongly express $\alpha 5$ and $\beta 1$ integrins (Maginnis MS et al. 2006 J Virol 80;2760). Therefore, in the absence of CD4, a tiny fraction of the RetroNectin-virus complexes present in the vicinity of the cell membrane can penetrate by other mechanisms, e.g., macropinocytosis, and thus produce a small (about 2% of cells), but present productive infection. It remains to be determined whether the internalization of RetroNectin complex can promote the fusion of virus particles with the endosomal membrane in the absence of CD4, and hence, escape the degradation into the lysosomes. **These phenomena did not occur during spinoculation.**

Reviewer's comment

I'm reminded of reports of “pseudotransduction” in which free GFP in high concentrations in a viral prep can make all of the target cells go slightly green and read out as positive in a flow experiment. Typically only a few cells become green enough to cross the line in the flow scatter plot but this can lead to a background of envelope independent infection. This is evidenced by the cells just crossing the line and being a lot less bright than cells actually expressing GFP. Is this is what is happening here? I must say flow plots in supp Fig 9a do not look like pseudotransduction so I'm not sure what's going on.

Author's response

We did not observe pseudotransduction. As with VSV-G pseudotyped HIV-1 (HIV-Gag-iGFP) (original Supplementary Figure S1), Gp160 pseudotyped HIV-1 particles (HIV-89.6-EGFP) are very faintly green (Figure R3b), leading to a weak background signal, which cannot explain the GFP⁺ cells observed in CD4-negative HeLa cells (see previous points). Nevertheless, GFP/EGFP signals in the viral particles can be artificially observed during image post-processing (see Figure R3, now Supplementary Figure 1).

Fig. R3. Gag-iGFP and EGFP signal in viral particles. a-c HIV-Gag-iGFP (a, a', top panel) and the corresponding “bald” virus, i.e. viral particles lacking the VSV-G envelope (a, a', bottom panel), HIV-89.6-EGFP (b) and CD9-GFP-positive EVs derived from CD9-GFP-expressing FEMX-I cells (c) were adhered to coverslips and observed by CLSM. Arrows point to very weak GFP and EGFP signals in HIV-Gag-iGFP (a) and HIV-89.9-EGFP (b) viral particles, respectively, especially compared to those of the CD9-GFP⁺ EVs (c). Gag-iGFP and EGFP signals could be observed after image post-processing using Fiji (↗, a', b').

Scale bars, 10 μm.

Reviewer's comment

When it comes to the association of NEIs with viral infection the presence of CD4 only doubles the number of NEI from 18 to 33 (line 292). Tripling the titre by expressing CD4 and doubling the virus induced NEI with CD4 undermines the whole message of the paper because infection and any processes on which infection depends are expected to be way more dependent on receptor than a 2-3 fold increase on receptor

expression. These experiments are really important for the paper because they use an actual HIV envelope and therefore the relevant entry pathway and they must be compelling. Can the authors explain what's going on here.

Author's response

Firstly, it is important to note that native HeLa cells already harbour NEIs in the absence of infection (see Figure 1g and Supplementary Figure 3) in a similar proportion to HIV-1-infected cells containing *bona fide* Env gp160, indicating that their number does not necessarily increase after infection, consistent with the absence of CD4. For clarity, we modified the text in the revised manuscript, see Supplementary Materials:

“The number of NEIs also increased in infected CD4⁺ HeLa cells (33.3 ± 4.2 NEIs per 50 cells, n = 3) compared to native HeLa cells (18.0 ± 3.0 NEIs per 50 cells), the latter being at the level of native condition, i.e. without infection (see Supplementary Fig. 3)”.

The limited amount of internalized viral components (IN-2 integrase or p24) may nevertheless reach the NEIs, probably the pre-existing NEIs, via the endosomal system. In a cell line (HeLa) that already has NEIs in the absence of viral exposure, it is difficult to parse the pre-existing from the newly formed NEIs. The situation becomes more obvious with activated CD4⁺ T cells, which are completely devoid of NEIs in the absence of infection (Figure 6b-e). HIV-1 induces NEIs, which are essential for productive infection in activated, but not in quiescent CD4⁺ T cells. The phosphorylation of ORP3 promoted by CD4⁺ T cell activation is a key step to allow VOR protein interaction, NEI formation and productive infection. For these reasons, in the revised manuscript, we focused on CD4⁺ T cells as mentioned above and moved all data with CD4⁺ HeLa cells in the Supplementary Materials.

Secondly, we cannot establish a direct link between cell internalization and nuclear transfer and/or NEI induction. This is again well illustrated with CD4⁺ T cells and HIV-1 containing a *bona fide* Env gp160 (Figure 6i and j). For example, doubling the amount of virus (MOI 2 to 4) did not increase the number of cells showing NEIs, although the relative amount of viral components (i.e. HIV-1 integrase) in the cytoplasm of infected cells increased (Supplementary Figure 14b and c). This is due to the translocation of virus-laden late endosomes to the MTOC where their interaction with the outer nuclear membrane via the VOR complex proteins will induce NEIs (Figure 7c). The polarization of late endosomes at the perinuclear region will not necessarily increase with increasing numbers of virus-laden late endosomes, as it occurs efficiently with small amounts of virus. In general, we observed only 1 (or 2) NEIs per cell (Supplementary Figure 3a) alongside the general deformation of the nuclear membrane observed after infection (Figure 6g and h). Although we haven't formally quantified the subnuclear localization of NEIs in CD4⁺ T cells, they seem to often appear close to the MTOC, where virus-laden late endosomes have accumulated (for example, see Figure 6k and l). In our previous study, we demonstrated that microtubules are essential for EV entry into NEIs (Santos et al., 2018 JBC 293;13834). Such information are summarized in the illustration presented in Figure 7c.

Reviewer's comment

The experiments with T cells simply use bald virus and this is more compelling. As I say above I wonder if the HeLa CD4 are just a terrible model for this. Line 563, I object to the statement “it remains to be determined whether viral internalization also involves nonspecific adhesion or aggregation of HIV-1 particles to the cell surface”⁹⁸. The model that HIV infection is CD4 dependent is well established and reference 98, and others cited to explain the strange data are way out of date, this one 1998. Using cells in which infection is apparently not receptor dependent is not helping the case here.

Author’s response

We are pleased to read that the reviewer appreciates the CD4⁺ T cell data and for these reasons we added additional data, including spinoculation-based infection and HIV-1-LTR-driven Gag expression on this more representative infection model. With regard to HeLa cells, we are not sure that this is a horrible model, although they contain NEIs endogenously, unlike CD4⁺ T cells, which made our conclusion more difficult to grasp. As mentioned above, the very low amount of productive infection (assessed by the presence of double EGFP⁺Gag⁺ cells) observed on CD4-negative HeLa cells is a consequence of the use of the RetroNectin complex to promote the internalization of viral particles, a phenomenon that is not observed with spinoculation. So, depending on the infection method, other cell surface proteins (e.g. integrins in the case of RetroNectin) may contribute indirectly, albeit to a small extent, to the internalization of extracellular particles.

REVIEWERS' COMMENTS

Reviewer #3 (Remarks to the Author):

The authors have worked hard to reproduce their original findings made in HeLa cells in primary CD4⁺ T cells and have presented new figures in the main text.

I agree that this makes the manuscript a lot more compelling and I have no further comments.

Author’s response

Thanks